# Heterogeneous, temporally consistent, and plastic brain development after preterm birth

Melissa Thalhammer [1,2] ✉, Jakob Seidlitz [3,4,5,6], Antonia Neubauer [1,2,7], Aurore Menegaux[1,2], Benita Schmitz-Koep[1,2], Maria A. Di Biase[8,9], Julia Schulz[1,2], Lena Dorfschmidt [3], Richard A. I. Bethlehem [10], Aaron Alexander-Bloch [3], Chris Adamson[11], Gareth Ball [11,12], Joana Sa de Almeida[11,13], Richard Beare[11], Claus Zimmer[1,2], Marcel Daamen [14,15,16], Henning Boecker [14,15], Peter Bartmann [16], Dieter Wolke [17,18], Dennis M. Hedderich [1,2] & Christian Sorg[1,2,19]

The current view of neurodevelopment after preterm birth presents a strong paradox: diverse neurocognitive outcomes suggest heterogeneous neurodevelopment, yet numerous brain imaging studies focusing on average dysmaturation imply largely uniform aberrations across individuals. Here we show both, spatially heterogeneous individual brain abnormality patterns but with consistent underlying biological mechanisms of injury and plasticity. Using cross-sectional structural magnetic resonance imaging data from preterm neonates and longitudinal data from preterm children and adults in a normative reference framework, we demonstrate that brain development after preterm birth is highly heterogeneous in both severity and patterns of deviations. Individual brain abnormality patterns are also consistent for their extent and location along the life course, associated with glial cell underpinnings, and plastic for influences of the early social environment. Our findings extend conventional views of preterm neurodevelopment, revealing a nuanced landscape of individual variation, with consistent commonalities between subjects. This integrated perspective implies more targeted theranostic intervention strategies, specifically integrating brain charts and imaging at birth, as well as social interventions during early development.

In humans, preterm birth is defined as birth before 37 weeks of gestation[1]. With a worldwide prevalence of about 11%[2], it is the leading cause of perinatal mortality and long-term motor, cognitive, and behavioral impairments[3–5]. Impairments are typically more severe with decreasing gestational age (GA) and lower quality of postnatal care[6,7]. Preterm birth impacts brain development through initial brain injuries that result in dysmaturation patterns affecting multiple tissue types and brain regions[8,9]. Initial injuries like hypoxic-ischemic events damage vulnerable cell populations, mainly the pre-oligodendrocyte

(OL) cell line and the subplate, with subsequent local inflammation mediated by reactive astrocytes and activated microglia[9–15]. Resulting brain aberrations after preterm birth have been shown to affect most parts of the brain, i.e., widespread gray matter areas, including cortical regions as well as basal ganglia, amygdala, thalamus, and hypothalamus nuclei[16–24], and white matter areas[25–30], not only in infants and children, but also in adolescents and adults. For example, cortical thickness (CTh) alterations might reflect various microscopic processes affected by prematurity, such as reduced synaptic density,

dendritic arborization, and axon length, potentially linked to aberrant white matter fiber maturation[28,31,32]. Group average focused brain magnetic resonance imaging (MRI) studies suggest that CTh is persistently altered in preterm-born subjects compared to full-term controls, with widespread CTh increases in infancy, a faster thinning rate in adolescence, and widespread decreases in adulthood[30,33–36]. Similarly, surface area (SA), thought to reflect the number of radial cortical columns[37], shows little or no alterations at term-equivalent age after birth[30,36], followed by a more pronounced and widespread decrease in childhood and adulthood[36,38,39]. These findings set the basis for a model in which preterm birth induces mean aberrations from typical maturation at different stages of development that are largely shared between individuals[8,40,41]. We term these widespread and consistent aberrations suggested by group average MRI studies average dysmaturation outcomes.

Whereas this model adequately describes the central tendency of brain aberrations of prematurity, it disregards, however, potential individual heterogeneity of altered brain development. Heterogeneity following preterm birth is evident for several outcomes[5,42–46], including postmortem neonatal brain aberrations as well as neurocognitive functioning. In particular, the average IQ after very preterm birth (i.e., before 32 weeks of gestation) is about 11 points lower than that of full-term peers in childhood and adulthood, but individual IQ varies considerably between subjects[42,47–49]. Furthermore, preterm birth is caused by a wide range of events, from spontaneous to induced, with various underlying pathogenic processes, each potentially associated with heterogeneous developmental trajectories. This neurocognitive and etiological heterogeneity is widely assumed to be reflected in related heterogeneity of neurodevelopmental processes[8,50,51]. In preterm neonates, Dimitrova and colleagues have already presented considerable heterogeneity of cortical outcomes with little spatial overlap[30]. Moving beyond the group-average paradigm to characterize individualized patterns of neurodevelopment may enable substantial conceptual as well as clinical progress in understanding and ultimately treating the consequences of preterm birth. Conceptually, the heterogeneity of the impact of preterm birth on the brain might be recognized as an essential feature of preterm birth that needs to be considered in our understanding of prematurity effects. If so, the diagnosis of such individual brain abnormality patterns might allow for individualized prognosis and treatment of risks for aberrant neurocognitive development.

Thus, considering vast heterogeneity in etiological underpinnings and neurocognitive outcomes alongside persistent and widespread average brain aberrations, we hypothesized substantial spatial heterogeneity of individual brain abnormality patterns (IBAPs) after preterm birth across development. Despite suspected heterogeneity, we expected consistency in specific aspects of IBAPs, particularly concerning features related to injury-induced dysmaturation and developmental plasticity. Regarding injury-induced dysmaturation, we hypothesized a consistent impact of initial injury on IBAPs in three domains: extent, anatomical location, and cellular underpinnings. Specifically, we hypothesized that (i) IBAP extent depends on GA, with earlier birth associated with larger IBAPs, (ii) anatomical locations of IBAPs remain temporally constant along individual development, and (iii) cellular underpinnings of IBAPs involve glial cells. Regarding developmental plasticity, we additionally expected IBAPs to be modified by the child's early social environment, supported by previous findings linking social environment to neurocognitive outcomes after preterm birth[52–54]. Finally, inspired by neurocognitive heterogeneity, we proposed that IBAP heterogeneity would underpin the cognitive performance variability among preterm born individuals.

In this work, we focus on CTh and SA as paradigmatic outcomes of brain development. Since CTh and SA capture distinct aspects of cortical development[55], both cortical measures implicated after preterm birth are investigated in parallel. First, we employ normative

modeling to describe regional CTh and SA developmental trajectories[56] and to assess spatial heterogeneity of IBAPs across three cohorts, encompassing preterm and full-term neonates, children aged 10 and 12 years, and adults aged 26 and 38 years (for an overview of the study hypotheses, data, and methods, see Fig. 1). Second, we test the consistency of initial injury using longitudinal IBAPs of preterm children and adults and integrate cell density maps derived from the Allen Human Brain Atlas (AHBA) to trace cellular underpinnings of IBAPs back to preterm birth. Third, we examine plasticity of IBAPs in response to the environment by linking adult IBAPs with social environmental features of early development. Finally, we test whether adult IBAPs were associated with neurocognitive performance scores after preterm birth. Taken together, we propose a temporally consistent effect of variable initial injury-induced dysmaturation at the macroscopic (i.e., extent and location) and microscopic (i.e., cellular) scale, which is, however, sensitive to early social-environmental influences, leading to individually heterogeneous abnormality patterns after preterm birth.

## Results
### Preterm brain development is heterogeneous
Instead of focusing on average dysmaturation outcomes, we sought to demonstrate spatial heterogeneity in brain development after preterm birth by using brain MRI data from three cohorts: (i) neonates scanned at term-equivalent age (131 preterm, 564 full-term) from the developing Human Connectome Project (dHCP), (ii) children scanned longitudinally at ages 10 and 12 years (191 preterm, 5762 full-term) from the Adolescent Brain Cognitive Development Study (ABCD-10 and ABCD-12), and (iii) adults scanned at age 26 years (96 preterm, 107 full-term) with a subgroup scanned again at age 38 years (52 preterm, 53 full-term) from the ongoing Bavarian Longitudinal Study (BLS-26 and BLS-38; Supplementary Data 1). Subjects born at different stages of the preterm period, namely extremely preterm (GA ≤ 28 weeks), very preterm (28 < GA ≤32 weeks), and moderate to late preterm (32 < GA < 37 weeks) were included in this study and are collectively referred to as preterm throughout this work (see Methods for the detailed inclusion criteria for each cohort). To replicate previous models of average dysmaturation, we used linear regression models correcting for age and sex to analyze mean CTh and SA differences between preterm and full-term individuals for each cohort for 34 cortical regions of the Desikan-Killiany parcellation[57]. Consistent with previous findings[30,35,41,58], regional mean CTh was increased in most areas except for the rostral middle frontal and inferior parietal areas in preterm neonates. In contrast, it was restrictedly decreased in frontal and temporal regions in children. In preterm adults, CTh was widely decreased in lateral associative and primary cortices and increased in cingulate areas (Fig. 2a: BLS-26; Fig. 3: other cohorts, Supplementary Tables S2). SA decreases extended to frontal, temporal, and parietal areas across the lifespan. Only the temporal pole showed increased SA in preterm children and adults compared to controls (Fig. 4, Supplementary Tables S2). For a control analysis excluding subjects with severe perinatal brain injury, refer to Supplementary Fig. S1a and Supplementary Methods and Results S2.1.

Next, to investigate spatial heterogeneity of brain aberrations, we analyzed IBAPs for CTh measured as CTh norm deviations based on the BrainChart normative reference framework. This framework has previously been established based on approximately 100,000 subjects to capture typical regional CTh development across the human lifespan[56]. We adopted these reference charts to our datasets by independently estimating random effects of study based on term-born individuals. Normative ranges were operationalized as the range between the 5th and 95th percentiles for a given age and sex. For each individual and each brain region, we calculated deviation scores that quantify how much an individual's CTh deviates from the normative range (Fig. 2b). A deviation profile refers to the regional distribution of deviation

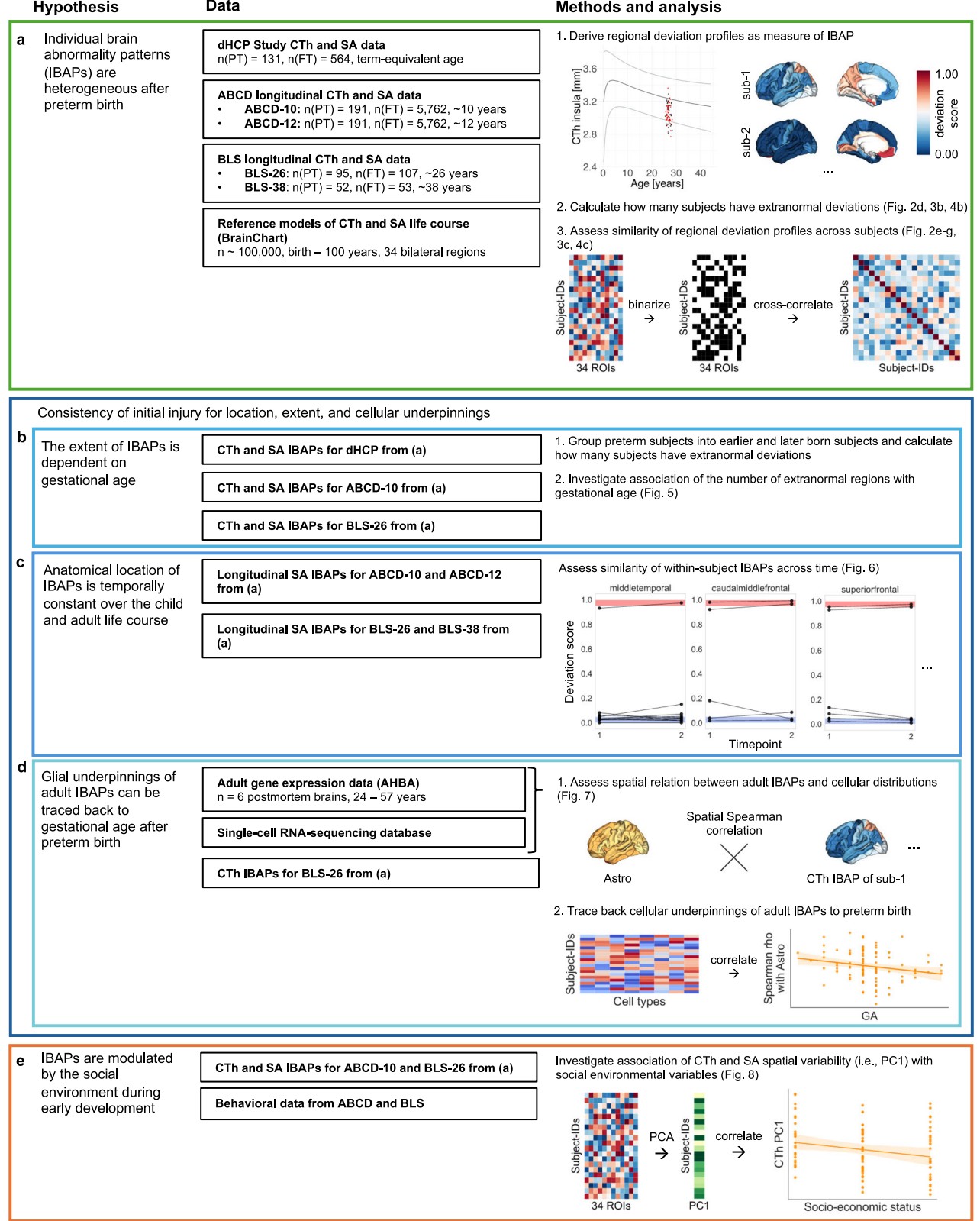

scores across all 34 investigated brain regions. Adapted normative models accurately captured typical development since more than 85% of full-term individuals resided within the normative range for any given region. To examine whether deviation profiles are heterogeneous across preterm individuals, we first focused on the BLS-26 cohort. Visual comparison demonstrated that the severity and pattern of deviations varied substantially, not only with respect to average

dysmaturation outcome (Fig. 2a) but also between individuals, even among those born with similar GA and birth weight (BW; Fig. 2c). Although the majority of individuals (84% of full-term and 97% of preterm subjects) showed at least one extranormal deviation, the locations of these deviations were markedly heterogeneous among individuals. No more than 27% of preterm adults of the BLS-26 cohort shared extranormal deviations in the same cortical region (Fig. 2d).

**Fig. 1 | Study overview.** Schematic of the study workflow, from hypotheses (left) to data sources (middle) to analysis steps (right). **a** Cross-sectional and longitudinal cortical thickness (CTh) and surface area (SA) data were obtained from three developmental cohorts, including preterm (PT) and full-term (FT) participants. Data were parcellated into 34 bilateral cortical regions. Population CTh and SA life course trajectories were extracted from a normative model. For each participant, individual regional deviations from population life courses were classified as infranormal (i.e., <5th percentile) or supranormal (i.e., >95th percentile) for each cortical region. The regional deviation profile of a certain modality is defined as the measure of the individual brain abnormality pattern (IBAP) for a given participant. IBAP heterogeneity was assessed in two ways: by quantifying the number of subjects with extranormal deviations in each region, and by measuring spatial similarity of regional deviation profiles across subjects. For the latter, binarized regional deviation profiles were cross-correlated to determine the average correlation between each subject's IBAP with all others. **b** Consistency in initial brain injury

following preterm birth was examined in terms of extent, location, and cellular underpinnings. Extent consistency was examined by relating a subject's number of extra-normal regional deviations to their gestational age (GA). **c** Location consistency was exami ned by within-subject longitudinal comparison of IBAPs along childhood and adulthood. **d** Cellular underpinning consistency was examined by linking gestational age with the spatial correlation of eight brain cell type distributions with adult IBAPs, respectively. Investigated cell types were astrocytes (Astro), endothelial cells (Endo), microglia (Micro), excitatory neurons (Neuro-Ex), inhibitory neurons (Neuro-In), oligodendrocytes (Oligo), oligodendrocyte precursors (OPC), and pericytes (Per). **e** To investigate the developmental plasticity of IBAPs in children and adults, we captured spatial variability of IBAPs across 34 cortical regions using Principal Component Analysis (PCA) as the main axis of deviation across regions (i.e., PC1) and linked it with social environmental factors during early life.

This result was independent from the normative model used to estimate regional CTh development in the population (Supplementary Methods and Results S2.2, Supplementary Fig. S2– S4, Supplementary Tables S3)[56,59], suggesting both the robustness of our finding regarding the applied model and distinct patterns of cortical dysmaturation after preterm birth. To investigate potential spatial patterns of extranormal deviations in more detail, we created binarized deviation profiles (i.e., infra- and supranormal deviations are designated with 1, all others with 0; Fig. 2e) and cross-correlated them across preterm subjects to estimate deviation profile similarity (Fig. 2f). Although extranormal deviation profiles show similarities between some preterm subjects (Spearman rho > |0.4 | ), the average correlation coefficient between one individual's pattern and all others was very weak (ranging from −0.07 to 0.09), suggesting individually heterogeneous patterns of CTh deviations across cortical regions. Severe perinatal brain injury did not drive these findings (Supplementary Fig. S1b, c). Control analyses for non-binarized deviations (Supplementary Fig. S5– S6) as well as the similarity of regional deviation patterns across individuals (Supplementary Methods and Results S2.3, Supplementary Fig. S7– S8) are provided in the Supplement.

To investigate whether this heterogeneity is observable across development, we analyzed deviation profiles in neonates, children, and adults aged 38 years (Fig. 3). The overall prevalence of at least one extranormal deviation varied with age from only less than 65% in preterm neonates to over 85% of preterm children and adults. Location patterns remained heterogeneous across all developmental stages, with no more than 27% of preterm individuals sharing deviations in any given region in the BLS-38 cohort as well, suggesting that CTh is heterogeneously altered across development after preterm birth.

To determine whether individual heterogeneity of structural abnormalities after preterm birth also applies to other brain features, we examined regional SA (Fig. 4) as well as cerebral tissue volume measures of white matter volume (WMV), gray matter volume (GMV), and subcortical GMV (sGMV; Supplementary Fig. S9). Regional percentile SA trajectories for the entorhinal cortex were not available due to known issues of lower data quality or cortical surface reconstruction for this region[56,60]. Most brain measures showed similar heterogeneity compared to CTh, with only less than 30% of preterm individuals sharing abnormalities in the same location for regional SA, GMV, and WMV. This suggests that the concept of individual heterogeneity of regional brain measures after preterm birth generally holds for cortical development. In contrast, we observed that up to 40% of preterm adults deviated in sGMV, which might point to a different mechanism for subcortical development. These findings indicate that brain development after preterm birth is characterized by persistent individual heterogeneity across developmental stages, affecting multiple structural measures of brain development.

## Consistency of preterm brain development deviations

Despite demonstrating that brain development after preterm birth is substantially heterogeneous between individuals, we hypothesized that biological mechanisms of injury-induced brain development deviations would be consistent across preterm subjects. Therefore, we examined macroanatomical (i.e., extent and location) and microanatomical (i.e., cellular underpinnings) consistency of deviations across development (Fig. 1b–d, respectively).

To investigate whether the extent of extranormal deviations depends on the severity of prematurity, we first divided the BLS-26 preterm adults into two groups of earlier (GA of ≤ 30 weeks, $n = 53$) and later birth (GA of >30 weeks, $n = 43$). In the earlier born group, up to 36% of preterm subjects shared a deviation in the same location, while only 19% of subjects did in the later born group. Similarly, up to 37% of preterm neonates born before 30 weeks GA ($n = 27$) shared the location of supranormal deviations, whereas this was the case for only 18% of subjects born later in the preterm period ($n = 65$; Supplementary Fig. S10). Repeating this analysis with partitioning subjects into groups more commonly used in clinical practice, i.e., extreme preterm (GA of ≤28 weeks), very preterm (28 < GA ≤32 weeks), and late preterm (GA > 32 weeks) resulted in similar results (Supplementary Fig. S11) but led to less balanced group sizes. In general, these results indicate that earlier birth increases the percentage of individuals that share extranormal structural alterations in overlapping regions. To further support this relationship, we examined whether the number of extranormal brain regions scales with GA. In both preterm neonates (supranormal: Spearman rho(129) = −0.255, $p = 0.003$, $p_{FDR} = 0.005$, confidence interval (CI) = [−0.409, −0.087], two-tailed) and adults (infranormal: Spearman rho(94) = −0.313, $p = 0.002$, $p_{FDR} = 0.005$, CI = [−0.484, −0.120], two-tailed), earlier birth was associated with a higher number of extranormal regions (Fig. 5a), demonstrating that the timing of preterm birth influences the extent of extranormal CTh deviations. In children, the association was not significant (infranormal: Spearman rho(380) = −0.098, $p = 0.055$, $p_{FDR} = 0.055$, CI = [−0.197, 0.002], two-tailed), probably because in youth, deviations shift from supra- to infranormal[36]. As a control analysis, we examined two additional measures of the severity of prematurity in the BLS-26 cohort, namely BW and the Duration of Neonatal Treatment Index (DNTI). DNTI provides an estimate for the severity of medical complications after birth. In contrast to BW (Spearman rho(94) = −0.162, $p = 0.116$, $p_{FDR} = 0.116$, CI = [−0.351, 0.040], two-tailed), DNTI showed a significant association with the number of infranormal regions (Spearman rho(94) = 0.242, $p = 0.018$; $p_{FDR} = 0.036$, CI = [0.043, 0.422], two-tailed, Supplementary Fig. S10c). These results demonstrate that GA might contribute substantially to the extent of CTh deviations, with earlier birth leading to more widespread patterns of deviations, also later in life.

To investigate whether locations of extranormal deviations remain stable over time within the same individual, we compared

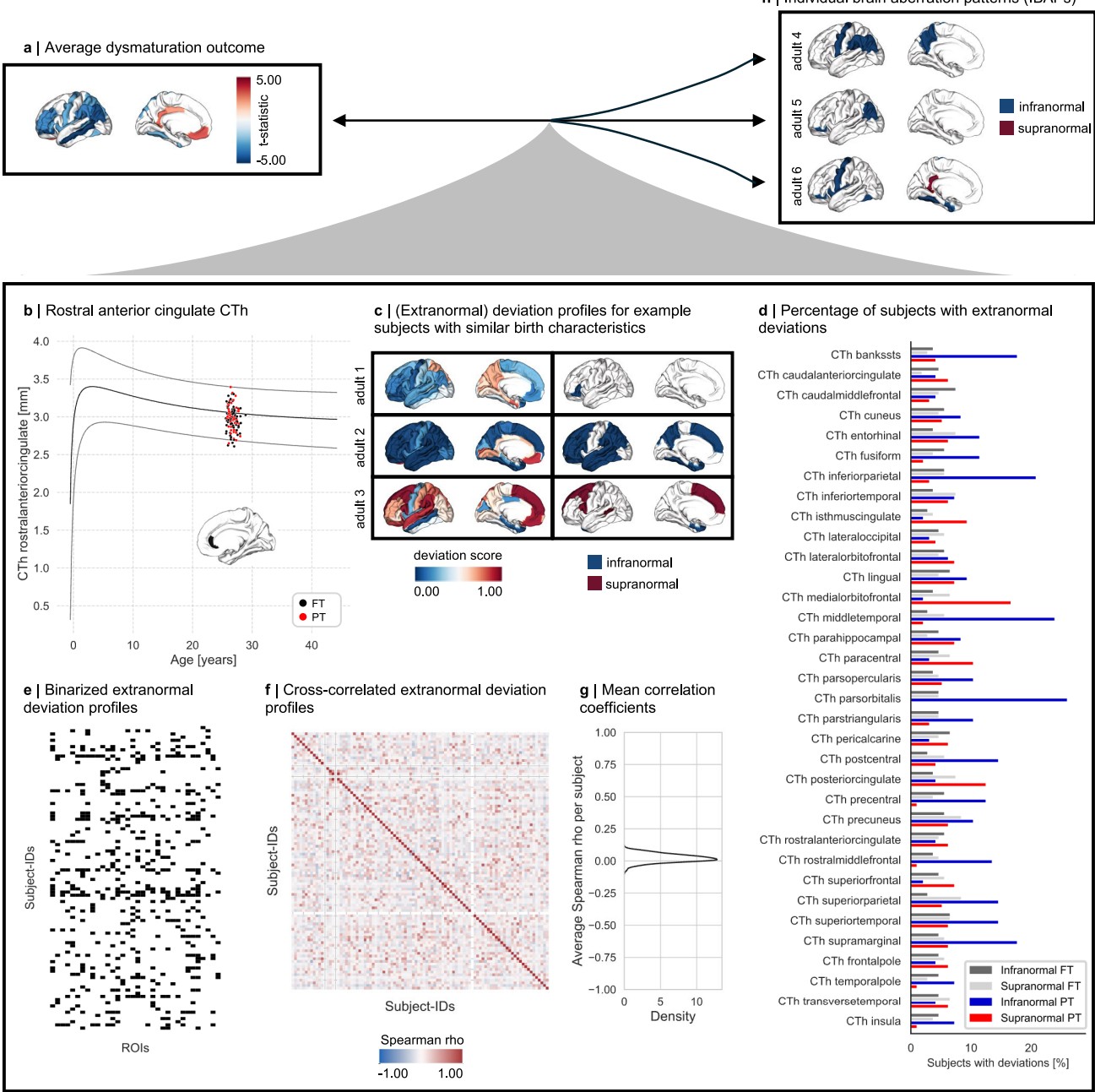

**Fig. 2 | Brain development after preterm birth is individually heterogeneous. a** Cortical thickness (CTh) average dysmaturation outcome for 26-year-old adults after preterm birth estimated by linear regression models correcting for age and sex of between-group differences ($p_{FDR}$ <0.05, see Supplementary Table S2j for exact $p$ values) suggests abnormalities shared between preterm subjects. **b** Individual regional deviations were estimated from a previously published normative framework[56] and classified as infranormal (i.e., <5th percentile) or supranormal (i.e., >95th percentile) for a given cortical region. The lifespan trajectory of the rostral anterior cingulate CTh in males is shown as an example region, with the rostral anterior cingulate CTh of male preterm (PT, red) and full-term (FT, black) subjects plotted on top. **c** Individual deviation score profiles of example subjects with similar gestational age (GA) and birth weight (BW). Left column: Deviation scores of all 34 cortical regions, right column: only infra- (i.e., <5th percentile, dark blue) and supranormal (i.e., >95th percentile, dark red) deviations. Adult 1: GA = 30 weeks, BW = 1450 g; adult 2: GA = 30 weeks, BW = 1460 g; adult 3: GA = 30 weeks, BW = 1465 g. **d** The percentage of subjects sharing an extranormal deviation in any given region is less than 30%, illustrating individual heterogeneity of CTh after preterm birth. **e** Binarized representation of extranormal deviations for each subject. **f** Spearman correlation matrix of binarized extranormal deviations across subjects. **g** Distribution of averaged correlation coefficients for each subject with all others. **h** Further example individual deviation score profiles. Left and right columns as in panel (**c**). See Supplementary Table S1 for region abbreviations. Source data are provided as a Source Data file.

regional SA and CTh deviation scores longitudinally in preterm children (from 10 to 12 years) and in adults (from 26 to 38 years). We focused on SA rather than CTh due to previous studies demonstrating that FreeSurfer estimates of raw regional CTh have substantially higher overall variability compared to SA[61]. Results for longitudinal consistency of CTh can be found in the Supplement (Supplementary Figs. S12– S13). We observed that preterm subjects who exhibited an extranormal deviation for SA in a particular region at the first timepoint were likely to exhibit a similar deviation in that same region at the second timepoint (Fig. 6 for selected regions of ABCD and BLS due

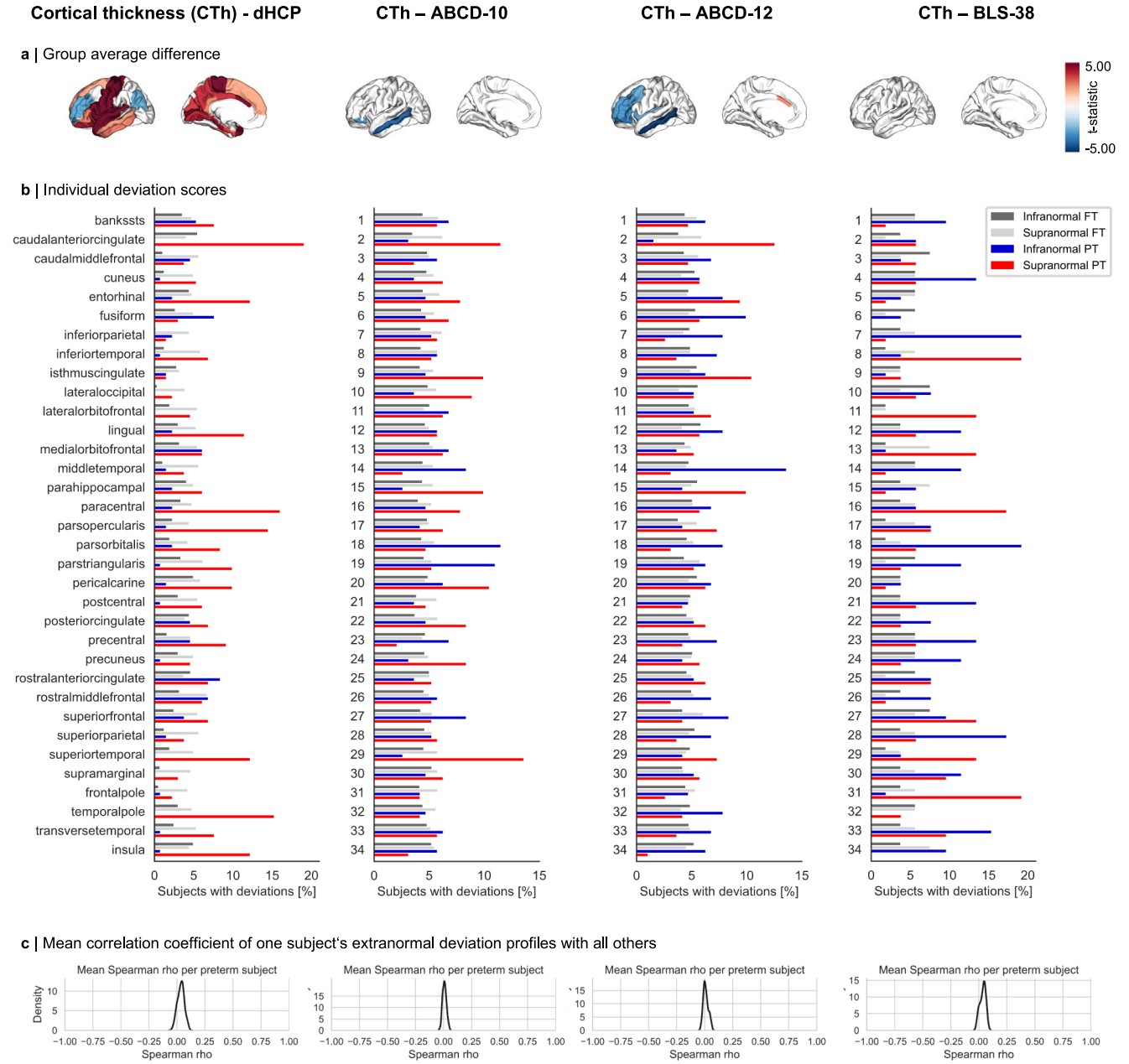

**Fig. 3 | Individual heterogeneity of cortical thickness after preterm birth is lasting across age groups. a** Cortical thickness (CTh) average dysmaturation outcome after preterm birth estimated by linear regression models correcting for age and sex ($p_{FDR}$ <0.05, see Supplementary Tables S2 for exact $p$-values). Results are shown for neonates (dHCP), children (ABCD-10 and ABCD-12), and adults (BLS-38). **b** Bar plots representing the percentage of preterm (PT) and full-term (FT) subjects sharing an extranormal deviation in any given region. An overlap of less than 20% suggests substantial heterogeneity between subjects. **c** Distribution of averaged correlation coefficients of binarized extranormal deviation profiles for each subject with all others. See Supplementary Table S1 for region abbreviations and number encoding. Source data are provided as a Source Data file.

to space limitations, Supplementary Fig. S14 – S15 for all cortical regions of both datasets). For infranormal deviations, 77% of children and 74% of adults maintained their patterns between scans. Supranormal deviations showed even higher consistency, with 71% stability in children and 87% in adults. To quantitatively evaluate whether IBAPs of preterm individuals remain constant over time, we computed the intraclass correlation coefficient (ICC) for each brain region. For all regions except the frontal and temporal poles, which are known to be less reliably estimated by FreeSurfer[57,60], ICC was higher than 0.837 for children and higher than 0.803 for adults (Supplementary Tables S4a, b), indicating longitudinal stability of extranormal deviations. In contrast, CTh IBAPs were less stable across childhood and adulthood, with

ICCs ranging from 0.394 to 0.876 in children and from 0.426 to 0.844 in adults (with the temporal pole being an extreme outlier, ICC = −0.093; Supplementary Tables S4c, d; Supplementary Fig. S16 for regional ICC distributions). While some regions showed higher variability, possibly due to measurement stability issues for these regions[61] or by non-linear developmental trajectories[56,62,63], individual patterns generally remained stable throughout childhood and adulthood.

To investigate whether there are consistent cellular underpinnings of IBAPs across development, we assessed whether the locations of some subjects' deviations follow the regional distribution of the same glial cells shown to be vulnerable to prematurity-induced

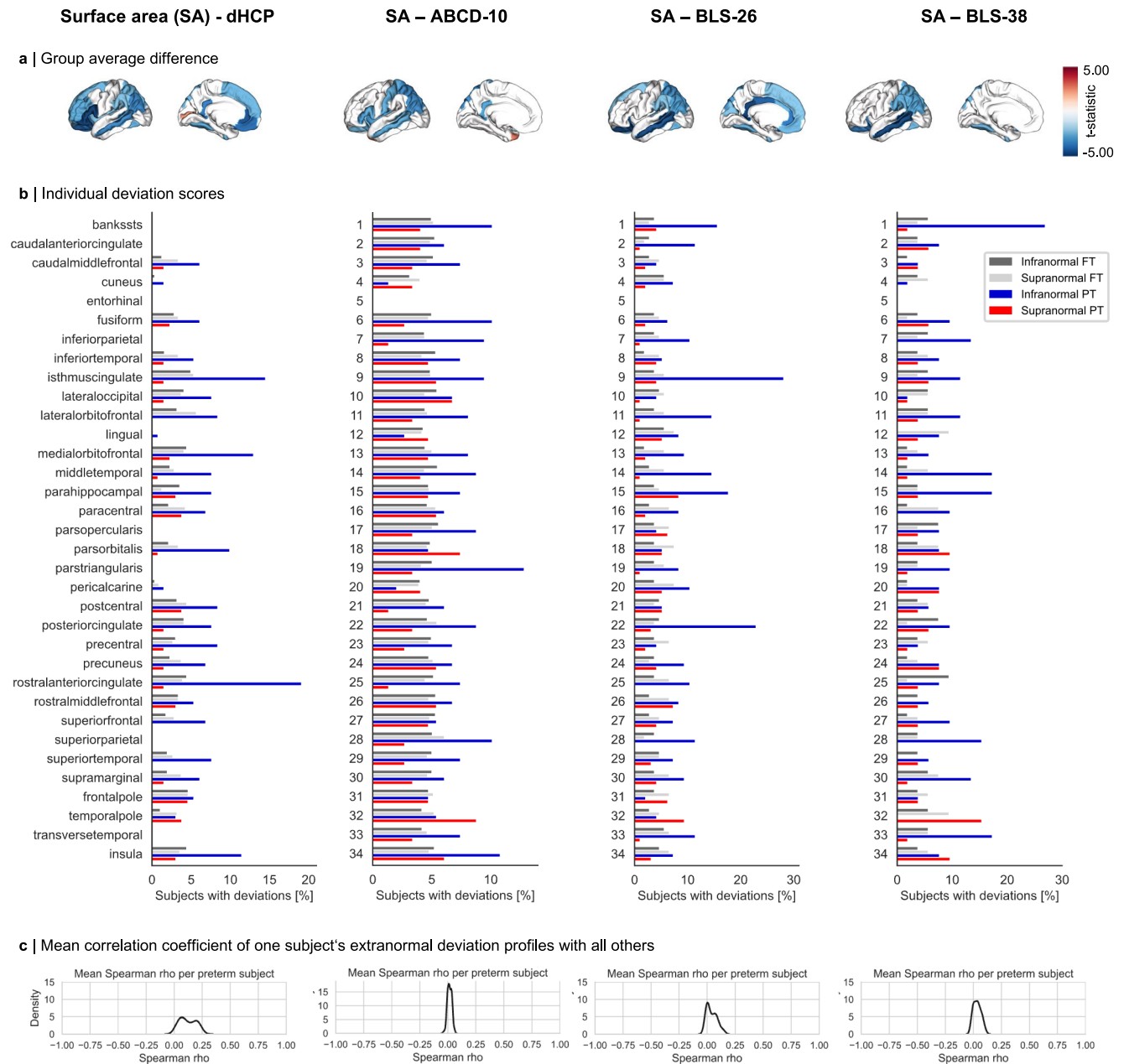

**Fig. 4 | Individual heterogeneity of regional surface area after preterm birth across cohorts. a** Surface area (SA) average dysmaturation outcome after preterm birth estimated by linear regression models correcting for age and sex ($p_{FDR}$ <0.05). Results are shown for neonates (dHCP), children (ABCD-10), and adults (BLS-26 and BLS-38). **b** The percentage of preterm (PT) and full-term (FT) subjects sharing an extranormal deviation in any cortical region. Not more than 30% of subjects overlap in any region, demonstrating that spatial heterogeneity between individuals is also

evident for regional SA development after preterm birth. As regional percentile SA trajectories for the entorhinal cortex were not available from the normative model, no subjects of any cohort shows extranormal deviations in that region.
**c** Distribution of averaged correlation coefficients of binarized extranormal deviation profiles for each subject with all others. See Supplementary Table S1 for region abbreviations and number encoding. Source data are provided as a Source Data file.

injury. Given that CTh variability has been demonstrated to be sensitive to various cellular underpinnings[31,32,64–66], we focused on CTh deviations in the following analyses. First, we estimated the regional abundance of eight different cell types across 34 cortical regions using gene sets from single-cell gene expression studies of the adult human cortex ("Methods", for similar methodologies see refs. [67,68]) and regional gene expression information from the AHBA[69,70] (Fig. 7a; Methods). The expression level of cell-specific marker genes has previously been established as an indirect indicator of the relative abundance and/or function of this cell type[71,72]. To bridge macro- and microscale by mapping cell-type marker gene expression patterns to

regional CTh deviation scores across the cortical sheet, we conducted regional Spearman correlation analysis across the 34 brain regions for each subject. Thereby, significance was assessed using spatial null models (spin test, see "Methods")[73,74] (Fig. 7a, filtered for significant associations only in Supplementary Fig. S17). We observed a significant association of subjects' regional CTh deviation patterns to some cellular distributions ($p_{spin}$ <0.05; refer to Supplementary Methods and Results S2.4 for a different spatial null model). Of note, across subjects, we found that the strength of the spatial relationship between cellular distributions and deviation profiles varied as a function of GA only for certain cell types (Fig. 7b, Supplementary Table S5). Specifically, earlier

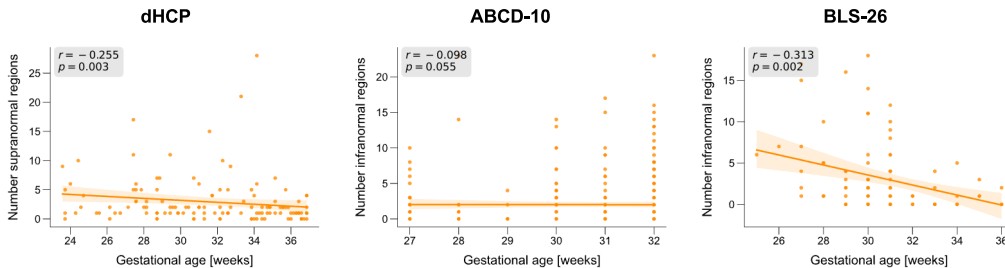

**Fig. 5 | Consistency of IBAP extent.** The number of supranormal or infranormal CTh deviations per subject was correlated with gestational age for dHCP, ABCD-10, and BLS-26, respectively (dHCP: Spearman rho(129) = −0.255, p = 0.003, p_FDR = 0.005, CI = [−0.409, −0.087], two-tailed; ABCD-10: Spearman rho(380) = −0.098, p = 0.055, p_FDR = 0.055, CI = [−0.197,0.002], two-tailed; BLS-26:

Spearman rho(94) = −0.313, p = 0.002, p_FDR = 0.005, CI = [−0.484, −0.120], two-tailed). Orange lines and shades represent the linear regression line with 95% confidence intervals based on 10,000 bootstrap resamples. Source data are provided as a Source Data file.

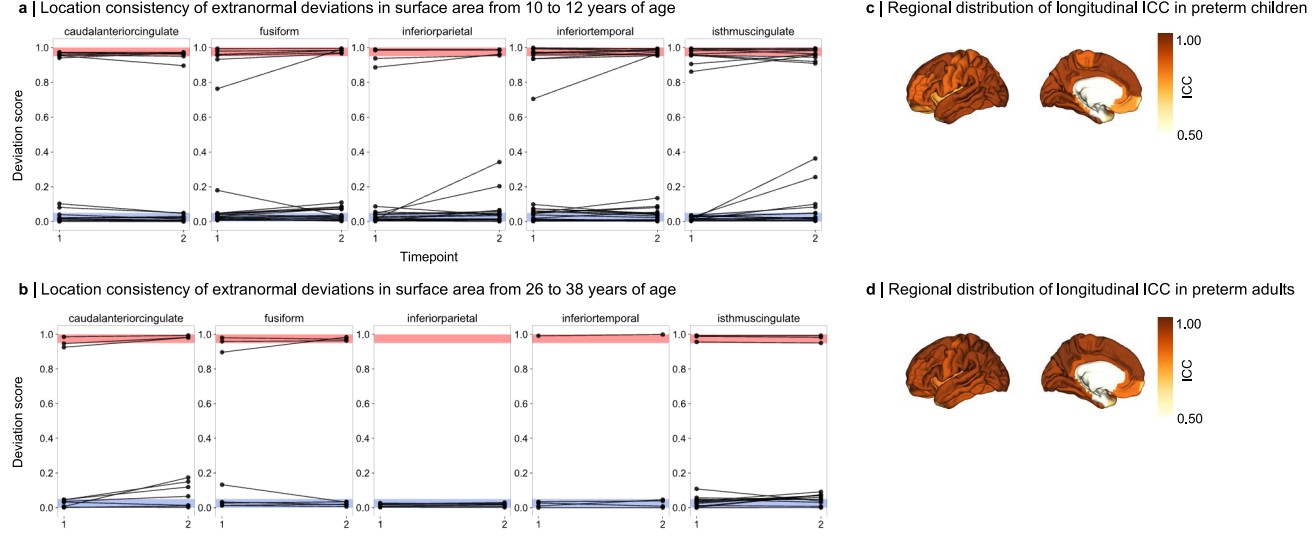

**Fig. 6 | Temporal consistency of IBAP anatomical location.** Deviation scores for regional surface area (SA) were computed in preterm individuals with available longitudinal data using the bilateral Desikan-Killiany parcellation: **a** for preterm children from the ABCD cohort (n = 296) at ages 10 and 12 years and **b** for preterm adults from the BLS cohort (n = 46) at ages 26 and 38 years. Five representative regions are shown here; a complete representation can be found in Supplementary Figs. S14–S15. Only individuals who exhibited an extranormal deviation (i.e., infranormal: <5th percentile, supranormal: >95th percentile) at either timepoint are depicted. Within-subject comparisons demonstrate that the anatomical location of infranormal (blue) and supranormal (red) deviations remain largely consistent across time. Intraclass correlation coefficients (ICCs) were used to quantify temporal stability of regional deviations. Regional distributions of ICCs are shown on left hemispheric cortical surfaces for (**c**) preterm children and **d** preterm adults. See Supplementary Table S1 for region abbreviations. Source data and exact p-values are provided as a Source Data file (panels **a**, **b**) and in Supplementary Tables S4a, b (panels **c**, **d**).

preterm birth was linked to a stronger association between regional CTh deviations and the regional distributions of astrocytes, endothelial cells, and oligodendrocyte progenitor cells (OPCs), and to a weaker association with mature oligodendrocytes. This suggests that the locations of deviations in earlier prematurity depends largely on the same glial cells that are known to be vulnerable to initial injury. We interpret this result as a glial consistency of cellular underpinnings of IBAPs.

Altogether, these findings point towards lasting, individually heterogeneous brain deviations, consistent in both macroanatomical extent and location as well as microanatomical association through glial cells.

### Impact of early social environment on IBAPs

In addition to consistency in injury-induced dysmaturation processes, we also expected a consistent plastic effect of the early environment on individual structural deviations after preterm birth (Fig. 1e). We addressed this question in children (ABCD-10) and adults (BLS-26). To summarize CTh and SA deviations across the cortex, we derived a first principal component (PC1) across all 34 regional CTh or SA deviation scores, respectively, for preterm individuals of each investigated cohort. Loadings for PC1 were positive across all 34 regions and both cohorts (Supplementary Fig. S18), indicating that this component represents a global trend in deviations across the cortex, with lower PC1 scores reflecting lower deviation scores relative to population norms. As representatives of the early developmental environment, we used family socio-economic status (SES) as a general, widely used measure of the economic and sociological environment components of a developing child[75], and the Parent-Infant-Relationship Index (PIRI) as a measure of parental social environment specifically (only available for BLS-26)[76].

In children of the ABCD-10 cohort, SES was estimated based on parental education level retrieved from parental interviews. In adults of the lifelong BLS study, SES was assessed neonatally as a composite score of family profession and parental education based on parent interviews and was classified into low, middle, and high[77]. PIRI describes attachment-related parental concerns and the mother's current and anticipated relationship problems[76]. A sum score from 0

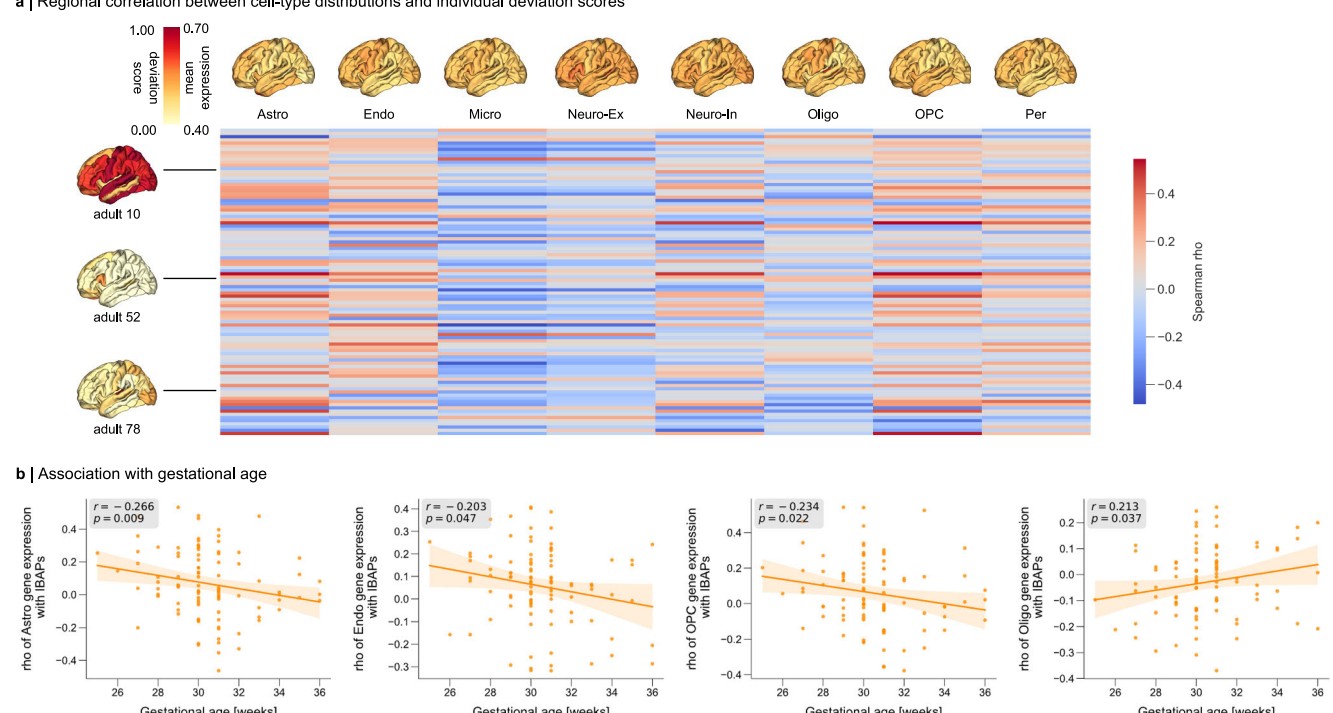

**a | Regional correlation between cell-type distributions and individual deviation scores**

**b | Association with gestational age**

**Fig. 7 | Temporal consistency of IBAP cellular underpinnings. a** Regional Spearman correlation matrix of cortical thickness (CTh) individual brain abnormality patterns (IBAPs) from preterm adults aged 26 years (vertical) with mean expression profile of eight brain cell types (horizontal). CTh IBAPs of three example subjects are plotted for illustration purposes. Investigated cell types were astrocytes (Astro), endothelial cells (Endo), microglia (Micro), excitatory neurons (Neuro-Ex), inhibitory neurons (Neuro-In), oligodendrocytes (Oligo), oligodendrocyte precursors (OPC), and pericytes (Per). **b** Spearman correlation coefficients for the association between astrocytes, endothelial cells, oligodendrocytes, and OPCs, with individual CTh deviations were significantly related to gestational age in preterm adults ($p < 0.05$, two-sided). Orange lines and shades represent the linear regression line with 95% confidence intervals based on 10,000 bootstrap resamples. See Supplementary Table S5 and Source Data files for Source Data and exact $p$ values.

(good parent-infant relationship) to 8 (poor parent-infant relationship) was calculated based on nurse observations neonatally and an interview at five months after birth. This allowed us to examine whether an individual's social environment during childhood might contribute to heterogeneous deviation profiles in adulthood.

To analyze whether a worse early social environment negatively impacts on CTh and SA heterogeneity after preterm birth, we correlated the respective variable with CTh and SA PC1 scores, respectively. In children, SES was significantly associated with SA PC1 (Spearman rho(294) = 0.269, $p = 0.001$, $p_{FDR} = 0.006$, CI = [0.160,0.371], two-sided), but not with CTh PC1 (Spearman rho(380) = −0.002, $p = 0.977$, $p_{FDR} = 0.977$, CI = [−0.102,0.098], two-sided; Fig. 8a). In adults, CTh PC1 scores and both SES (Spearman rho(94) = −0.217, $p = 0.033$, $p_{FDR} = 0.088$, CI = [−0.400, −0.018], two-sided) and PIRI (Spearman rho(89) = −0.211, $p = 0.044$, $p_{FDR} = 0.088$, CI = [−0.400, −0.006], two-sided) were significantly associated (Fig. 8b), indicating that a more favorable social and economic environment links to higher CTh deviations across the cortex. SA PC1 scores were not correlated with social environmental scores (SES: Spearman rho(94) = −0.006, $p = 0.952$, $p_{FDR} = 0.977$, CI = [−0.207,0.194], two-sided; PIRI: Spearman rho(89) = −0.076, $p = 0.474$, $p_{FDR} = 0.711$, CI = [−0.278,0.132], two-sided). As control measures for PCA-based integrated individual CTh deviations across the cortex, we used the number of infranormal regions as well as the deviation scores of mean CTh across the whole cortex (Supplementary Fig. S19). In preterm adults, we observed that a lower SES was correlated with a lower deviation score of mean CTh (Spearman rho(94) = −0.223, $p = 0.029$, $p_{FDR} = 0.051$, CI = [−0.405, −0.032], two-sided) and a higher number of infranormal regions in CTh per subject (Spearman rho(94) = 0.200, $p = 0.051$, $p_{FDR} = 0.051$, CI = [0.000,0.385], two-sided), substantiating the

association between early social environment and CTh deviations. Based on previous literature suggesting that a lower SES modifies the relationship between preterm birth and adverse neurodevelopmental outcomes[52–54] in later life, we tested whether a lower SES also adversely modifies the association between preterm birth and adult IBAPs. Moderation analysis revealed that the association between GA and PC1 scores (Spearman rho(94) = 0.264, $p = 0.009$, CI = [0.068,0.433], two-sided) was significantly weakened among individuals with middle and low SES (interaction term SES x GA: β = 0.104, $p = 0.035$, SE = 0.057, CI = [−0.0090, 0.217], one-sided; conditional effects of SES on GA: high SES: effect = 0.039, $p = 0.303$; middle SES: effect = 0.143, $p = 0.001$; low SES: effect = 0.246, $p = 0.35 \times 10 \times 10^{-4}$, one-sided; Fig. 8c), indicating that an adverse early social environment modifies IBAPs after preterm birth.

## Impacts on cognitive outcome variability

Finally, to test whether heterogeneity in IBAPs underlies cognitive performance variability among preterm individuals across the lifespan, we associated CTh and SA PC1 scores with measures of cognitive outcomes in the dHCP, ABCD-10, and BLS-26 cohorts.

In the dHCP cohort, neurocognitive testing was performed at 18 months using the Bayley Scales of Infant and Toddler Development, Third Edition (Bayley-III)[78,79]. There was no significant correlation between IBAPs at term-equivalent age and cognitive abilities at 18 months (CTh PC1: Spearman rho(129) = −0.153, $p = 0.127$, $p_{FDR} = 0.152$, CI = [−0.316,0.019], two-sided; SA PC1: Spearman rho(129) = 0.157, $p = 0.118$, $p_{FDR} = 0.152$, CI = [−0.015,0.320], two-sided; Fig. 9a), presumably because of the age gap between imaging and neurocognitive assessments as previously reported by others[30]. In the ABCD-10 cohort, the NIH Toolbox Cognition Battery composite scores

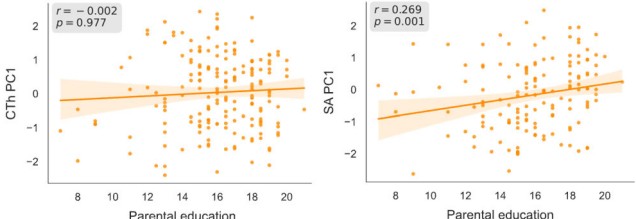

**a | ABCD-10: early social environment impacts CTh and SA after preterm birth**

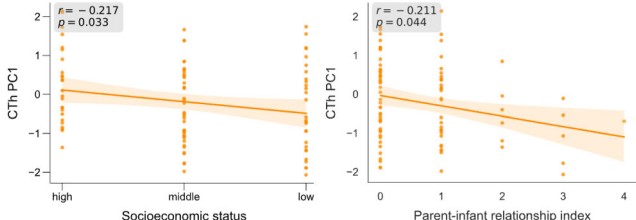

**b | BLS-26: CTh and SA after preterm birth are associated with early social environment**

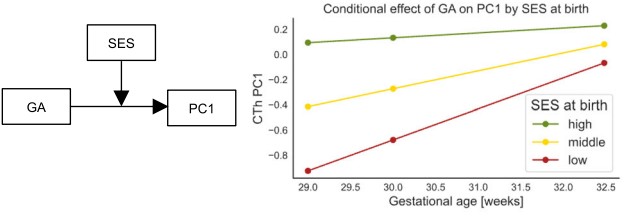

**c | BLS-26: early social environment moderated preterm birth impact on CTh**

**Fig. 8 | Deviations of children and adults are linked to the social environment during early development.** To capture regional variation of deviation scores across 34 cortical regions, we applied Principal Component Analysis (PCA) to individual deviations (Fig. 1e). Next, we assessed the association between the first principal component (PC1) of regional deviations and early developmental social factors, including socio-economic status (SES) and quality of mother-infant relationship (Parent-Infant Relationship Index, PIRI; BLS-26 only) using Spearman rank correlation. **a** In preterm children of the ABCD-10 cohort, the association between SES with the first principal component (PC1) of regional cortical thickness (CTh) (left, Spearman rho(380) = −0.002, $p$ = 0.977, $p_{FDR}$ = 0.977, CI = [−0.102,0.098], two-sided) and of regional surface area (SA, right, Spearman rho(294) = 0.269, $p$ = 0.001, $p_{FDR}$ = 0.006, CI = [0.160,0.371], two-sided) is shown. **b** In preterm adults of the BLS-26 cohort, the association between CTh PC1 with SES (left, Spearman rho(94) = −0.217, $p$ = 0.033, $p_{FDR}$ = 0.088, CI = [−0.400, −0.018], two-sided) and with PIRI (Spearman rho(89) = −0.211, $p$ = 0.044, $p_{FDR}$ = 0.088, CI = [−0.400, −0.006], two-sided) is illustrated. Orange lines and shades in panels (**a**, **b**) represent the linear regression line with 95% confidence intervals based on 10,000 bootstrap resamples. **c** In adults, the association between gestational age (GA) and CTh PC1 scores was moderated by SES (interaction term SES x GA in linear regression model: β = 0.104, $p$ = 0.035, SE = 0.057, CI = [−0.0090,0.217], one-sided). Conditional effects of SES on GA are shown on the right. A low (red, effect = 0.246, $p$ = 0.35 × 10⁻⁴, one-sided) or middle (yellow, effect = 0.143, $p$ = 0.001, one-sided) SES significantly moderates the relationship between SES and GA, whereas a high SES (green, effect = 0.039, $p$ = 0.303, one-sided) does not. Source data are provided as a Source Data file.

were used to estimate intelligence[80], which has previously been validated to provide a reliable measure of IQ[81]. Cognition total composite scores were significantly associated with PC1 of SA IBAPs (Spearman rho(146) = 0.238, $p$ = 0.005, $p_{FDR}$ = 0.010, CI = [0.080,0.385], two-sided; Fig. 9b) but not with PC1 of CTh IBAPs (Spearman rho(189) = −0.054, $p$ = 0.472, $p_{FDR}$ = 0.472, CI = [−0.195,0.088], two-sided) in 10-year-olds. Finally, in the BLS-26 cohort, we observed a significant correlation between full-scale IQ and SA PC (Spearman rho(94) = 0.270, $p$ = 0.009, $p_{FDR}$ = 0.027, CI = [0.073,0.446], two-sided) but not with CTh PC1 (Spearman rho(94) = 0.198, $p$ = 0.059,

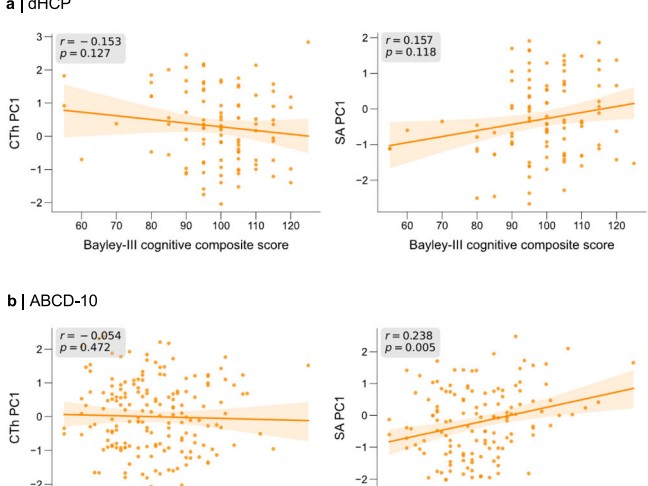

**a | dHCP**

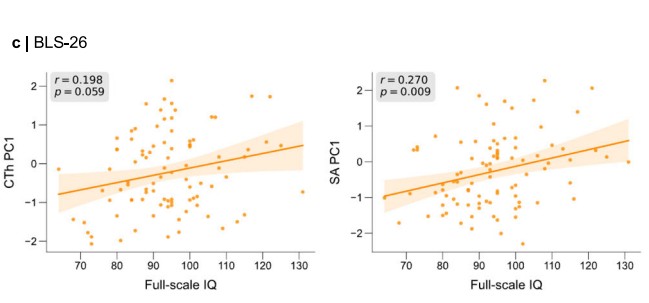

**b | ABCD-10**

**c | BLS-26**

**Fig. 9 | Individual brain deviations after preterm birth are associated with cognitive outcome variability in later life.** Spearman correlations between the first principal component (PC1) of regional cortical thickness (CTh, left) or of regional surface area (SA, right) with measures of cognition in preterm neonates (**a**), children aged 10 years (**b**), and adults aged 26 years (**c**). Orange lines and shades represent the linear regression line with 95% confidence intervals based on 10,000 bootstrap resamples. All statistical tests were two-sided, and uncorrected $p$ values are provided for illustration purposes. Source data are provided as a Source Data file.

$p_{FDR}$ = 0.118, CI = [−0.003,0.383], two-sided; Fig. 9c). The control analyses associating the number of infranormal SA regions with IQ confirmed this relationship in children (Spearman rho(294) = −0.232, $p$ = 0.006, $p_{FDR}$ = 0.012, CI = [−0.337, −0.121], two-sided) and adults (Spearman rho(94) = −0.214, $p$ = 0.041, $p_{FDR}$ = 0.041, CI = [−0.397, −0.014], two-sided; Supplementary Fig. S20). In brief, results indicate that individual structural deviations in SA contribute, at least partly, to the variability of general cognitive performance in adulthood after preterm birth.

## Discussion

This study provides evidence for lasting and heterogeneous brain aberrations after preterm birth, shaped by consistent effects of variable initial injury-induced impaired brain development at macroscopic (i.e., extent and location) and microscopic (i.e., cellular) scales, and mediated by the early social environment. This individual heterogeneity of cortical development, combined with biological mechanistic consistency for injuries and plasticity, integrates two opposing suggestions on preterm brain development: highly heterogeneous neurodevelopment due to varying neurocognitive outcomes versus regular aberrant neurodevelopment based on average dysmaturation outcomes of brain imaging studies. Thus, the concept of broadly altered brain development with widespread, persistent, average-based brain aberrations after preterm birth needs to incorporate the

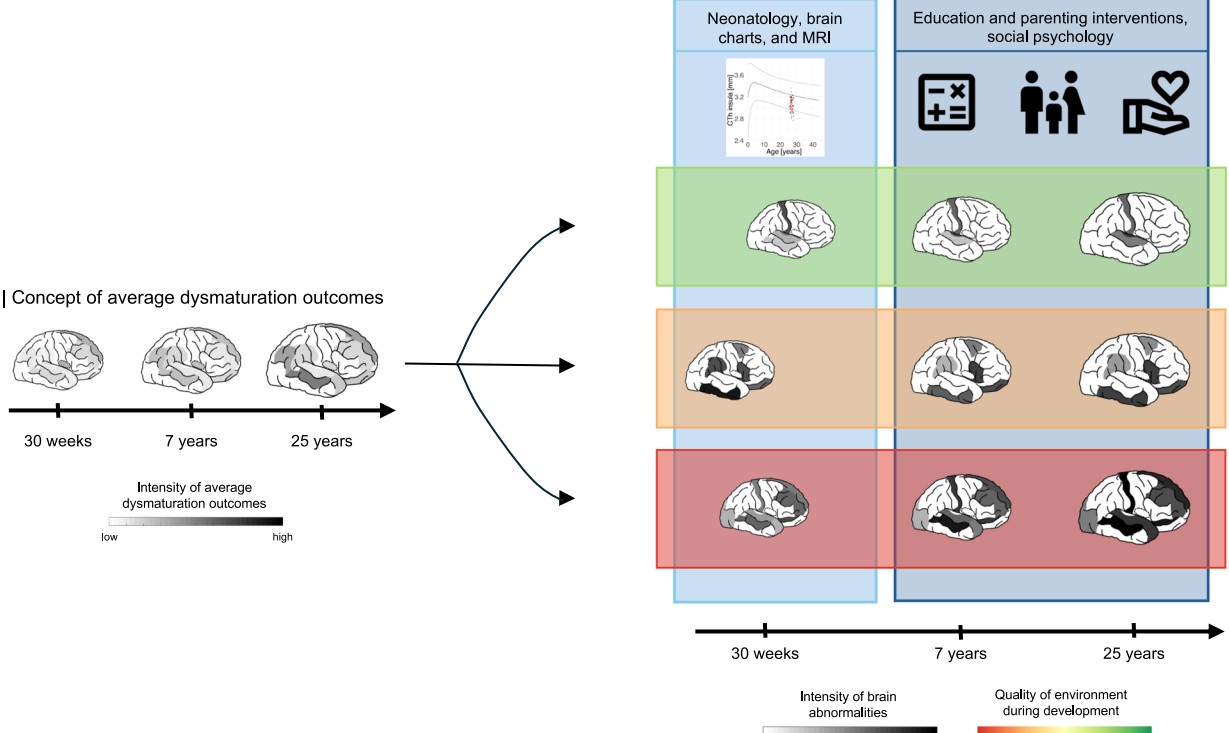

**b | Concept of individual brain abnormality patterns and theranostic approaches**

**a | Concept of average dysmaturation outcomes**

**Fig. 10 | Conceptual transition from average dysmaturation outcome-based view to individual brain abnormality pattern centered model of prematurity.** Schematic representation of the two contrasted concepts. **a** The prevailing view of brain aberrations based on group-level brain imaging studies after preterm birth suggests injury-induced average dysmaturation outcomes, disregarding individual variability. **b** The extended concept presented here emphasizes the heterogeneity of individual brain abnormality patterns (IBAPs) among preterm subjects, as a result of individual initial injury patterns (indicated by varying extent and location of abnormalities across three arbitrary subject cases), subsequent dysmaturation (indicated by a brain trajectory for each case), and plastic individual changes due to social environment influences (indicated by horizontal boxes around distinctively developing brain trajectories). Critically, the model of heterogeneous and plastic IBAPs suggests specific theranostic approaches on prematurity (blue shaded vertical boxes): on the one hand, neonatology and chart-based brain imaging approaches to identify at-risk infants, and on the other hand, social psychology approaches to social environment to modify developmental trajectories. Thus, heterogenous IBAPs seem to have the potential to frame and integrate approaches of basic neuroscience, neonatology, brain imaging, and social psychology on human prematurity. Source data are provided as a Source Data file. Icons used from Google's Material Icons (https://fonts.google.com/icons; licensed under Apache License 2.0).

substantial component of individual heterogeneity following temporally consistent but environmentally sensitive trajectories of dysmaturation (Fig. 10).

Leveraging normative modeling approaches, we identified distinct individual brain abnormality patterns (IBAPs) in preterm individuals across life stages, varying in extent and spatial distribution. While group-level analyses suggest spatially consistent alterations across subjects, individual effects vary substantially (Fig. 2a, Fig. 3a, Fig. 4a). Only about a quarter of preterm subjects exhibited an overlapping deviation in the same cortical region, indicating that the degree and location of brain abnormalities in an individual cannot be inferred from average dysmaturation outcomes derived from group-level analyses. This heterogeneity persisted across all examined cohorts, extended beyond CTh to other structural measures, and was robust across different normative reference models (Supplementary Fig. S2), suggesting individual developmental trajectories of brain structural features after preterm birth. These findings substantially advance the concept of individually heterogeneous brain development after preterm birth, previously proposed by postmortem and imaging studies, which presented high spatial variability of glial cell-mediated white matter lesions between subjects[82–84].

Consistent with our results, previous normative modeling studies have also presented substantial heterogeneity of brain measures in preterm neonates, including CTh and SA[30,50,51]. The work by Dimitrova and colleagues suggests that no more than 10% of preterm neonates share extranormal deviations in several structural brain features, which is similar to our findings in the same cohort of a maximum of about 20% spatial overlap of deviations. Small discrepancies might stem from methodological differences, especially different normative models and cortical parcellations used. Although overlapping for neonates, our study advances the field by additionally investigating lifelong heterogeneity as well as consistency of underpinning biological mechanisms including plasticity. In doing so, we advance the understanding of brain development after preterm birth by providing a more comprehensive and life-long perspective.

Despite the considerable heterogeneity of brain aberrations, injury mechanisms regarding the extent, location, and cellular underpinnings of individual abnormality patterns seem to share commonalities (Figs. 5–7). Earlier gestational age was correlated with more extranormal deviations in both neonates and adults, suggesting a consistent extent of deviations from birth into adulthood. Considering that previous studies have shown that earlier born preterm individuals are more likely to have brain injuries, extreme individual brain aberrations, and subsequent neurodevelopmental impairments[30,85–87], these findings are not surprising. Moreover, structural deviations outside the normative range demonstrated remarkable macro-anatomical stability throughout development, with some regional exceptions potentially due to signal or segmentation quality issues[57,60] or non-linear maturation trajectories[56,62,63]. Studies that analyzed group average longitudinal trajectories of raw CTh and SA from

childhood to early teenage years[36] and from adolescence to young adulthood[88] after preterm birth also reported a consistent aberration pattern across individuals in later life. In conclusion, individual cortical aberrations are dependent on GA and seem to persist at the same locations into middle adulthood.

Investigating potential cellular-molecular underpinnings, we found that the cell types mediating initial injury effects of preterm birth on individual CTh aberrations are similar across the first half of the lifespan, i.e., involve certain glial cell types. As expected, individually heterogeneous regional CTh deviation patterns were significantly linked to several different cell type profiles, again stressing that CTh aberrations are different across preterm individuals. Critically, however, adults born earlier in the preterm period exhibit deviation patterns more closely aligned with the regional distributions of astrocytes, endothelial cells, and OPCs and less aligned with those of oligodendrocytes. This suggests consistent cellular underpinnings between neonates and adults, at least for adults born earlier in the preterm period. Particularly, glial cells prone to hypoxia-ischemia injury may be damaged around time of birth[9-15,89-94], with extranormal deviations still being present at the locations where most astrocytes, endothelial cells, and OPCs reside in adulthood. OPCs are highly vulnerable to hypoxia-ischemia and are replenished after damage but fail to differentiate to mature oligodendrocytes in the following, resulting in hypomyelination[10]. This could potentially explain the overlap of IBAPs with OPC but the anti-correlation with oligodendrocyte cell distributions. As cortical thinning is largely dependent on intracortical myelination[31,32], areas with higher OPC abundance might be more prone to extreme cortical thinning after preterm birth. This interpretation assumes that regional cellular abundance in the adult and infant brain overlap. Yet, previous evidence suggests that transcriptional profiling of cells alters across development, but the regions where cells reside once at place remain mainly conserved after birth[95,96], substantiating our interpretation. Collectively, these findings reveal that the individual heterogeneity in regional CTh observed in adults born earlier in the preterm period is significantly associated with the regional distributions of glial cells implicated in neonatal brain injury.

Furthermore, we present evidence that IBAPs after preterm birth are modifiable by the early social environment and relevant for cognitive outcome variability, demonstrating an interplay between environment, brain development, and eventually long-term cognitive outcomes. By applying PCA, we captured global CTh and SA deviation as PC1. Individual child and adult CTh PC1 scores were significantly influenced by a subject's social environment during early development (Fig. 8), with medium or low SES moderating the association between GA and CTh PC1 scores in adults. Importantly, this suggests that a more favorable early social environment could mitigate the adverse effects of preterm birth on the brain. As social influences continue throughout childhood including parenting, peer and neighborhood influences[97,98], further investigation of childhood social influences is warranted. Taken together, we show that individual heterogeneous brain aberrations after preterm birth are not only modulated by common perinatal and social factors but also underpin cognitive outcome variability.

Overall, our findings suggest that brain development following preterm birth is both highly heterogeneous and temporally consistent, yet remains considerably plastic, which opens important theranostic implications (Fig. 10). As changes in neonatal care have increased survival rates but not cognitive, mental health, or health-related quality of life[99-102], interventions post-discharge across childhood may alter brain variability and cognitive development[103]. On the one hand, our findings stress the necessity of individualized diagnostics, for example using normative approaches of brain imaging measures to better identify at-risk infants. On the one hand, in addition to pharmacological interventions for neonates, comprehensive therapeutic strategies need to incorporate social and developmental interventions that focus on the social and learning environment such as parenting interventions and improved support in early education and schooling.

Clearly, the present work should be considered alongside some important methodological limitations. First, the study combined data from three cohorts with differing inclusion criteria for preterm birth, resulting in the investigation of individuals ranging from extreme preterm (GA ≤ 28 weeks) to late preterm (32 < GA < 37 weeks). Importantly, the consequences of preterm birth are influenced by the degree of prematurity, and findings from very preterm individuals may not generalize to those born moderately or late preterm. Second, for preterm neonates, we rely on cross-sectional data only. Although fetal scans were available for a small group of participants, normative modeling relies on reference data from term-born subjects at each timepoint to disentangle effects of age from diagnosis. Due to lack of these data, longitudinal analysis in the dHCP cohort was not possible. Furthermore, apart from the BrainChart reference charts[56], robust models for early age ranges between birth and 2 years of age are lacking, hindering validation of neonatal results. Since substantial heterogeneity of regional CTh in preterm neonates has been shown previously[30], however, there is further evidence based on different methodologies for individual heterogeneity of structural brain aberrations after preterm birth also in infancy. Future studies would benefit from longitudinal data, both for reference chart estimations[104] and for assessing individual preterm brain developmental trajectories. Third, investigation of cellular underpinnings relies on the assumption that the spatial organization of adult neurobiology is reflected in CTh estimates and conserved between humans. Although several pieces of evidence support this notion[64,65,69], non-invasive tools to investigate this assumption are lacking. Furthermore, the relationship between cell-type specific marker-gene expression and cell abundance varies for different cell types[105], and cell types are actually more diverse than simplified here[106].

In summary, we demonstrate that lasting and widespread altered developmental brain aberrations after preterm birth are individually heterogeneous in their extent and distribution across the brain. Still, the underlying initial injury mechanisms with respect to the extent, location, and cellular underpinnings of individual brain abnormalities patterns seem to be related to gestational age and are temporally consistent from birth until adulthood. Importantly, these abnormalities are modifiable by the early social environment, opening theranostic opportunities. Our work uncovers an integrative perspective on brain alterations after preterm birth by complementing broadly and widespread altered brain development after preterm birth with highly individual heterogeneous brain alterations.

## Methods
### Ethics
No new human data were acquired for this study. The Technical University of Munich, Germany, approved the usage of human demographic, behavioral, and neuroimaging data. Specific approval for collection and sharing of all data used in this study were provided by local ethics committees: research for the BrainChart model was reviewed by the Cambridge Psychology Research Ethics Committee (PRE.2020.104) and The Children's Hospital of Philadelphia's Institutional Review Board (IRB 20-017874) and deemed not to require PRE or IRB oversight as it consists of secondary analysis of de-identified primary datasets[56]. The dHCP study was approved by the United Kingdom Health Research Authority (Research Ethics Committee reference number: 14/LO/1169) and written parental consent was obtained in every case for imaging and open data release of the anonymized data[78]. The ABCD study was reviewed and approved by the institutional review board of the University of California, San Diego (IRB no. 160091). Additionally, the institutional review boards of each of the 21 data collection sites approved the study. Informed consent was obtained from all parents, and informed assent was obtained from

participants[107,108]. The BLS study was carried out in accordance with the Declaration of Helsinki and was approved by the local ethics committee of the Klinikum rechts der Isar, Technische Universität München, and the University Hospital Bonn. All study participants gave written informed consent. They received travel expenses and a small payment for participation. The responsible BLS investigators are D. Wolke, P. Bartmann, and C. Sorg.

## Software

If not stated otherwise, analyses were carried out in a Python 3.11.9 environment. Details on used software packages can be found in Supplementary Methods S2.5.

## Neonatal brain measure estimation from dHCP data

Infant brain measures were obtained from the developing Human Connectome Project[109] (approved by the National Research Ethics Committee; REC: 14/Lo/1169; 3rd data release). Prior to imaging, informed written parental consent was obtained. Term-born and preterm-born babies were scanned at term-equivalent age (37–45 weeks postmenstrual age) during natural unsedated sleep at the Evelina London Children's Hospital. T2-weighted (T2w) scans with the following parameters were acquired: TR = 12 s, TE = 156 ms, SENSE = 2.11/2.58 (both axial/sagittal) with in-plane resolution of $0.8 \times 0.8$ mm (1.6 mm slice thickness, 0.8 mm overlap). A 3D motion corrected sensitivity encoding reconstruction was applied, resulting in 0.5 mm isotropic resolution images[110]. Infant preparation and imaging have been previously described in detail[109]. Term-born subjects were excluded if admitted to neonatal intensive care unit or if scans revealed a significant intracranial abnormality (e.g., acute infarction and parenchymal hemorrhage but not punctate white matter lesions, small subependymal cysts/hemorrhages in the caudothalamic notch, mildly prominent ventricles, or widening of the extra-axial CSF). Preterm subjects were only excluded if they had major congenital malformations or diagnosed genetic disorders. Fetal age was estimated from the mother's last menstrual period and confirmed by ultrasound where possible. Gestational age at birth ranged from 24.29 to 36.86 weeks, i.e., the cohort includes late preterm as well as extreme preterm born neonates. Supplementary Data 1 shows demographic data of the final study sample of 564 term-born (300/264 male/female, age at scan: 37.43–44.86 weeks, mean ± SD: 41.40 ± 1.72 weeks) and 131 preterm-born (72/59 male/female, age at scan: 37.00–45.14, mean ± SD: 40.87 ± 2.09 weeks) individuals. Thickness and surface area maps were retrieved from automatically processed T2w images[111] and were parcellated into 34 bilateral cortical regions using the Desikan-Killiany-compatible M-CRIB-S(DK) atlas[57,112,113] and averaged across hemispheres.

## Child brain measure estimation from ABCD data

Brain measures for children were obtained directly from 4th release of the Adolescent Brain Cognitive Development (ABCD) Study (https://doi.org/10.15154/1523041)[114]. T1-weighted (T1w) and T2w MRI scans were acquired at different 3 T scanners at 21 sites, with the parameters that are listed on the study website (https://abcdstudy.org/images/Protocol_Imaging_Sequences.pdf). MRI data from baseline (acquired at around age 10 years) and the 2-year-follow-up (acquired at around age 12 years) were preprocessed using FreeSurfer 7.1.1 by the ABCD study team. Details about image acquisition and processing can be found in the original manuscript by Casey et al. [114] and in the release manuscript (https://doi.org/10.15154/1523041). Following quality control procedures previously established in normative modeling studies[56,59], we used FreeSurfer's Euler Index (EI), an automated, quantitative measure of cortical reconstruction quality[115,116], to estimate surface defects in cortical surface reconstructions. Following Lotter et al. [64]., we excluded subjects that had at least one missing value for the investigated brain measure (i.e., CTh or SA), exceeded a threshold of Q3 + IQRx1.5

calculated in each sample across timepoints[115,117], or failed the manual quality ratings provided in the ABCD dataset. Gestational age was derived from a parent questionnaire administered approximately 10 years after childbirth, asking "About how many weeks premature was the child when they were born?" (included in file abcd_devhxss01.txt). Given the retrospective and potentially imprecise nature of this estimate, we applied a conservative approach to define our cohorts. We determined gestational age as 40 weeks minus the reported number of weeks premature and included only those participants whose estimated GA was ≤32 weeks in the preterm group. This threshold corresponds to very preterm birth and was chosen to minimize misclassification due to vague recall of GA at birth. Therefore, subjects born between gestational weeks 32 and 37 were excluded. Full-term birth was defined as GA ≥ 37 weeks. Moreover, only subjects with longitudinal assessments were included. Regional CTh estimates were extracted for 191 preterm (102/89 male/female, age at first scan: 9.00–10.92 years, mean ± SD: 9.99 ± 0.60 years; age at second scan: 10.83–13.67 years, mean ± SD: 12.01 ± 0.64 years) and 5762 full-term (3051/2711 male/female, age at first scan: 8.92–11.08 years, mean ± SD: 9.90 ± 0.62 years; age at second scan: 10.58–13.75 years, mean ± SD: 11.94 ± 0.65 years) participants (Supplementary Table S1), regional SA estimates for 148 preterm (77/71 male/female, age at first scan: 9.00–10.92 years, mean ± SD: 9.97 ± 0.61 years; age at second scan: 10.83–13.67 years, mean ± SD: 12.00 ± 0.65 years) and 4929 full-term (2600/2329 male/female, age at first scan: 8.92–11.08 years, mean ± SD: 9.90 ± 0.62 years; age at second scan: 10.58–13.75 years, mean ± SD: 11.94 ± 0.65 years) participants (i.e., due to a higher number of missing values for SA). Estimates were averaged across hemispheres.

## Adult brain measure estimation from BLS data

The Bavarian Longitudinal Study (BLS) is a geographically defined, whole-population sample of individuals born very preterm (VP; <32 weeks of gestation) and/or with very low birth weight (VLBW; <1500 g) and full-term (FT; >37 weeks of gestation) controls that were followed from birth between January 1985 and March 1986 until adulthood[118,119]. Therefore, the cohort included subjects born very preterm but also subjects born later in the preterm period that had VLBW. 682 infants were born VP and/or with VLBW. Informed written consent from a parent and/or legal guardian was obtained. From the initial 916 FT born infants born at the same obstetric hospitals who were alive at 6 years, 350 were randomly selected as control subjects within the stratification variables of sex and family socioeconomic status in order to be comparable with the VP/VLBW sample. Of these, 411 VP/VLBW individuals and 308 controls were eligible for the 26-year follow-up assessment. 260 of the VP/VLBW group and 229 controls participated in psychological assessments[120]. All subjects were screened for MR-related exclusion criteria, including (self-reported): claustrophobia, inability to lie still for >30 min, unstable medical conditions (e.g., severe asthma), epilepsy, tinnitus, pregnancy, non-removable MRI-incompatible metal implants, and a history of severe CNS trauma or disease that would impair further analysis of the data. However, the most frequent reason not to perform the MRI exam was that subjects declined to participate. Finally, 101 VP/VLBW subjects and 111 FT controls underwent MRI at 26 years of age. The MRI examinations took place at two sites: The Department of Neuroradiology, Klinikum Rechts der Isar, Technische Universität München, (n = 145) and the Department of Radiology, University Hospital of Bonn (n = 67). At both sites, MRI data acquisition was performed on Philips Achieva 3 T TX systems or Philips Ingenia 3 T systems using an 8-channel SENSE head coil. A high-resolution T1w 3D-MPRAGE sequence (TI = 1300 ms, TR = 7.7 ms, TE = 3.9 ms, flip angle = 15°, field of view = 256 mm × 256 mm, reconstruction matrix = 256 × 256, reconstructed isotropic voxel size = 1 mm³) was acquired. Further details about the dataset are described elsewhere[17,20,25,26]. At the age of 38 years, the subjects deemed eligible for the 26-year MRI follow-up assessment were

contacted again and screened for MR-related exclusion criteria mentioned above. At the time of this study, a second T1w image was acquired for a subset of participants at the age of 38 on a Philips Ingenia Elition X 3 T scanner at the Klinikum Rechts der Isar, Technical University of Munich. A 3D-MPRAGE sequence with the following parameters was acquired: TI = 692 ms, TR = 8.7 ms, TE = 3.8 ms, flip angle = 8°, field of view = 256 mm × 240 mm × 161 × 161 mm, reconstruction matrix = 256 × 256, compressed-SENSE factor = 3, reconstructed isotropic voxel size = 1 mm³. For both acquisition timepoints, surface-based parcellation to extract regional CTh and SA in the Desikan-Killiany atlas[57] was conducted using FreeSurfer (version 7.3.2; http://surfer.nmr.mgh.harvard.edu/). Global measures of cerebrum tissue volume were also retrieved from FreeSurfer. For quality control, FreeSurfer's EI was extracted. Scans that had EI greater than 4 median absolute deviations above the median EI (n = 8) were excluded. One additional subject was excluded due to missing information on gestational age. The final samples of the BLS study have the following demographics: 96 preterm (53/42 male/female, age at scan: 25.71–28.34, mean ± SD: 26.75 ± 0.61) and 107 full-term (62/45 male/female, age at scan: 25.49–28.90, mean ± SD: 26.82 ± 0.72) subjects were analyzed as cohort BLS-26. At the time of this study, a subset of 52 preterm (30/22 male/female, age at scan: 37.36–39.07, mean ± SD: 38.10 ± 0.46) and 53 full-term (30/23 male/female, age at scan: 37.41–38.91, mean ± SD: 38.21 ± 0.40) subjects underwent a second scan, referred to as BLS-38 here. For the assessment of location consistency (see section Anatomical location consistency along development), longitudinal data were available for 46 preterm subjects (25/21 male/female, age at first scan: 25.77–28.34 years, mean ± SD: 26.72 ± 0.61 years). Further information on demographics is provided in Supplementary Data 1. Regional CTh and SA were averaged across hemispheres to match the normative modeling framework of BrainChart, resulting in 34 bilateral cortical ROIs. GA and BW were obtained from obstetric records.

## Data harmonization

As the ABCD-10, ABCD-12, and BLS-26 datasets were acquired on different scanners, bilateral regional CTh and SA as well as GMV, sGMV, and WMV values were retrospectively harmonized to remove variance related to different hardware and scanner sites. Since the scanners used for data acquisition of BLS-26 and BLS-38 were differing and BLS-38 was acquired with a single scanner, only the cross-sectional data of BLS-26 were harmonized with NeuroCombat (version 0.2.12 in Python; https://github.com/Jfortin1/neuroCombat)[121]. NeuroCombat has been repeatedly shown to successfully remove scanner variation in diffusion tensor imaging[122], structural[121], and functional[123] MRI data while preserving biological variance. In this algorithm, an empirical Bayesian framework is applied to estimate additive and multiplicative scanner site effects using parametric empirical priors. To adjust for both effects, the data is then subtracted by the additive and divided by the multiplicative effect parameters[121,124]. In our application, we preserved biological variation attributed to sex, prematurity, and age while controlling for the effect of four scanners used for data acquisition.

For ABCD-10 and ABCD-12, an extension of NeuroCombat for longitudinal brain imaging data, longCombat (version 0.0.0.9 in R 4.4.1; https://github.com/jcbeer/longCombat), was used, which has been shown to be more powerful in detecting longitudinal change than cross-sectional NeuroCombat[125]. One of 21 sites stopped data acquisition after baseline and was thus not included in our dataset. We adjusted site effects for 20 sites, while preserving biological variation attributed to sex, prematurity, and age.

The application of longCombat requires that the hardware used is the same at both acquisition times, which was not the case in BLS. The dHCP and BLS-38 years datasets were not harmonized for scanner effects since these data were acquired at the same scanner.

## Assessment of group average differences (Figs. 2a, 3a, 4a)

To estimate average dysmaturation outcomes for bilateral regional CTh and SA, a linear regression model correcting for age and sex was fit for each cohort. Results were adjusted for the False Discovery Rate (FDR)[126].

## Obtaining individual brain abnormality patterns (IBAPs; Fig. 2b)

Normative reference charts for bilateral regional CTh, SA, as well as for global measures (i.e., GMV, sGMV, and WMV) were obtained from the BrainChart project[56], in which GAMLSS-based normative models were computed over the human lifespan as a function of age and sex. The original model was trained on a large, population-level reference dataset of more than 100,000 subjects that also included individuals from the dHCP and ABCD studies. Importantly, for original model training, participants were selected based on general health criteria and were not stratified by gestational age, i.e., both term- and preterm-born individuals were included to the reference cohort as cognitively normal. Furthermore, only chronological age (not gestational or postnatal age) was included as a covariate into the model.

To adapt the reference charts to the datasets used for this study, random effects of study were calculated only based on full-term individuals and by treating dHCP and ABCD as a new site (i.e., "dHCP_new", "ABCD_new"). This step ensures that any potential differences due to imaging site or preprocessing pipeline are accounted for without including preterm data into the model fitting. By this means, the application data of interest in the present study (i.e., preterm individuals) are not part of the refitting process. Since Bethlehem et al. demonstrated in their original work that leave-one-study-out models yielded nearly identical model parameters compared to the full model[56], we argue that the overall model shape is only minimally influenced by inclusion or exclusion of individual cohorts and that refitting of the original model to term dHCP or ABCD data as well as its adaptation to preterm data is not influenced by the fact that dHCP and ABCD were used in the original training process.

Normative ranges were operationalized as the range between the 5th and 95th percentile. Based on the adapted normative models for each brain measure, a deviation score between 0 and 1 was then calculated for each individual and each cortical region. This deviation score represents the percentile of this person's CTh or SA value for the respective cortical region in relation to the normative range. Subjects falling outside the limits defined as the normal range were classified as infranormal (i.e., below the 5th percentile) or supranormal (i.e., above the 95th percentile) for a given region.

## Number of extranormal deviations (Figs. 2d, 3b, 4b)

To assess the number of subjects per group overlapping in their extreme deviation per region, the percentage of subjects within a group exhibiting an extranormal (i.e., infranormal or supranormal) deviation in a single region was calculated for each cohort.

## Assessing spatial heterogeneity of IBAPs (Figs. 2e–g, 3c, 4c)

To evaluate whether brain aberrations after preterm birth are spatially heterogeneous, we binarized IBAPs of preterm subjects for each cohort, so that deviations below the 5th or above the 95th percentile for any given region were designated with 1 and all others with 0. Next, Spearman correlation was used to cross-correlate binarized IBAPs across subjects. To estimate the similarity of a preterm individual with all others, the average correlation coefficient for a subject to all others was computed.

## Extent consistency (Fig. 5)

To determine whether the number of extranormal deviations depends on gestational age (GA), preterm individuals of the dHCP and BLS-26 cohorts were grouped into earlier birth (i.e., GA of ≤30 weeks) or later

birth (i.e., GA of >30 weeks) before calculating the percentage of subjects with extranormal deviations (Supplementary Fig. S10). Furthermore, Spearman's rank correlation was used to associate the number of extranormal regions with gestational age for preterm subjects of the dHCP, ABCD-10, and BLS-26 cohort (Fig. 5).

### Anatomical location consistency along development (Fig. 6)

To determine whether anatomical locations of extranormal deviations (i.e., below 5th or above 95th percentile) are temporally constant along development, we calculated deviation scores for the longitudinal subsamples of ABCD (i.e., 191 PT for CTh and 148 PT for SA, see section Child brain measure estimation from ABCD data), acquired at around ages 10 and 12, and BLS (i.e., 46 PT, see section Adult brain measure estimation from BLS data), acquired at around ages 26 and 38. To focus on clinically relevant deviations, only subjects who showed an extranormal deviation at a minimum of one of the two timepoints were selected. We compared whether extranormal deviations remained extranormal over time. To quantify the consistency of brain region deviation scores across timepoints, we calculated the Intraclass Correlation Coefficient (ICC) for each brain region using Python's pingouin package[127]. Specifically, the ICC(3,1) model was used, which is appropriate for repeated measures where the timepoints are treated as fixed effects and subjects as random effects.

### Microarray gene expression

Regional microarray expression data were obtained from 6 postmortem brains provided by the AHBA (https://human.brain-map.org/)[70]. The abagen toolbox (version 0.1.3; https://github.com/rmarkello/abagen)[128] was used to correctly match AHBA gene expression data with native donor space MRI images following published recommendations[129] as follows:

- Probe aggregation: Microarray probes were aggregated to genes using data provided by Arnatkevičiūtė et al. [129]. Probes not matched to a valid Entrez ID were discarded. When multiple probes indexed the expression of the same gene, we selected the probe with the highest mean correlation across donor pairs (i.e., differential stability[69])
- Intensity-based filtering: Probes were filtered based on their expression intensity relative to background noise[130]. Probes with intensity less than the background in at least 50% of the samples across donors were discarded.
- Distance threshold: Samples were assigned to brain regions, using Montreal Neurological Institute (MNI) coordinates generated via nonlinear registrations (https://github.com/chrisfilo/alleninf), by minimizing the Euclidean distance between the MNI coordinates of each sample and the nearest surface vertex. Samples where the Euclidean distance to the nearest vertex was more than 2 standard deviations above the mean distance for all samples belonging to that donor were excluded. To reduce the potential for misassignment, sample-to-region matching was constrained by hemisphere and gross structural divisions (i.e., cortex, subcortex/brainstem, and cerebellum, such that, e.g., a sample in the left cortex could only be assigned to an atlas parcel in the left cortex)[129]. All tissue samples not assigned to a brain region in the provided atlas were discarded.
- Sample normalization: To address inter-subject variation, tissue sample expression values were normalized across genes using a scaled robust sigmoid function[131].
- Gene normalization: Genes were normalized across tissue samples, again using a scaled robust sigmoid.
- Regional gene expression matrix: Samples assigned to the same brain region were averaged separately for each donor and then across donors, yielding a regional expression matrix in the 68-region surface-based Desikan-Killiany parcellation[57].

### Cell type-specific gene expression maps

As there is currently no available means to measure cellular abundance in vivo, data from multiple single-cell studies of the adult human cortex[132–136] were compiled previously[67] to generate a set of cell type-specific gene markers in the adult human brain for the following cell-types: astrocytes, endothelial cells, microglia, oligodendrocyte progenitor cells, oligodendrocytes, excitatory neurons, and inhibitory neurons. Cell type-specific maps representing cellular abundance across the cortex were generated by calculating the mean regional expression score for each gene set in the preprocessed AHBA bulk microarray dataset. Expression was averaged across hemispheres to match the bilateral 34 region setting of regional CTh deviation scores.

### Investigating cellular correlates of CTh heterogeneity (Fig. 7a)

To explore the cellular correlates of individual CTh heterogeneity in preterm 26-year-old adults, we assessed the spatial correspondence between cell type-specific gene expression maps and individual deviation score patterns using spatial Spearman correlation across 34 cortical regions. Whereas correlation coefficients for analyses of two spatial maps are meaningful, standard parametric or non-parametric methods for statistical inference are not as they are influenced by the spatial autocorrelation in spatial systems such as the brain. This potentially leads to inflated p-values and subsequently to increased familywise error rates across analyses[73]. To circumvent this problem, we applied spatial autocorrelation preserving spatial null models (spin tests) to assess the statistical significance of these map-to-map correspondences[73,74]. In brief, one of the investigated brain maps is projected to a sphere, random angular rotations are applied to this spherical projection, and the rotated map is then projected back to the brain surface. By repeating this procedure 1000 times and calculating the correspondence between the rotated maps and the second brain map of interest, a null distribution of correlation coefficients between the two brain maps is generated. The original correlation coefficient is then compared to this null distribution. This method maintains the existing spatial autocorrelation by applying the same random rotations to each parcel, thereby preserving the spatial structure of the brain map[73,74]. The analyses were implemented in Python using the ENIGMA toolbox[137] (https://github.com/MICA-MNI/ENIGMA). To evaluate the relationship between the spatial covariation of deviation scores and cell type abundance with GA, we used Spearman's rank correlation of Spearman correlation coefficients between individual deviation scores and cell type-specific gene expression with GA.

### Dimension reduction of IBAPs across cortical regions

To capture regional variability of deviation scores across 34 cortical regions, dimension reduction using Principal Component Analysis (PCA) was performed with the scikit-learn package (version 1.5.2) in Python. PC1 loadings as well as explained variances for each component were extracted from the calculations.

### Perinatal, social environmental, and behavioral variables

For the dHCP cohort, sex was derived from clinical or enrollment records. Available variables for social environment were limited to questionnaires about the age at which the mother and father left full-time education, which does not automatically translate to the quality of education. Neurocognitive assessments were conducted at the age of about 18 months using the Bayley Scales of Infant and Toddler Development, Third Edition (Bayley-III)[78,79]. We used the age-normed composite scores for cognitive development (mean = 100, SD = 15)[78]. This measure was available for a subset of 101 preterm participants only (56/45 males/females, age at scan: 37.00–45.14 weeks, mean ± SD = 40.83 ± 2.10 weeks).

For the ABCD cohort, sex was derived from demographic interview with a parent or guardian. Parental education based on responses to questionnaires ("What is the highest grade or level of school you

have completed or the highest degree you have received?") was used as an estimate for early social environment. SES estimates were available for 187 preterm subjects of the ABCD-10 cohort that also had CTh data available (101/86 males/females, age at scan: 9.00–10.92 years, mean ± SD = 10.00 ± 0.60 years) and for 145 preterm subjects that also had SA data available (76/69 males/females, age at scan: 9.00– 10.92 years, mean ± SD = 9.99 ± 0.61 years). Neurocognitive development was assessed with the NIH Toolbox Cognition Battery at the age of 10 years[80], which is highly (r = 0.89) correlated with IQ scores measured with the Wechsler Adult Intelligence Scale, fourth edition (WAIS-IV)[81]. For this study, the fully corrected T-scores of the Total Score Composite (considering certain demographic characteristics such as gender, education, and race/ethnicity; mean = 50, SD = 10) were used[80]. Cognitive scores were available for 177 preterm subjects of the ABCD-10 cohort that also had CTh data available (95/82 males/females, age at scan: 9.00–10.92 years, mean ± SD = 10.01 ± 0.60 years) and for 136 preterm subjects that also had SA data available (71/65 males/females, age at scan: 9.00– 10.92 years, mean ± SD = 10.00 ± 0.61 years). For the BLS-26 cohort, self-reported sex information was retrieved. Several variables were used to investigate the influences on and consequences of IBAPs in more detail. The duration of neonatal treatment indices (DNTI) was used to quantify the duration of intensive care treatment after birth, providing an estimate of medical complications after birth[76,138]. For this, care level, respiratory support, feeding dependency, and neurological status (mobility, muscle tone, and neurological excitability) were assessed daily and rated on a 4-point scale (0-3)[139]. DNTI is considered the duration of neonatal treatment in days. DNTI measures were available for 95 preterm subjects (52/43 males/females, age at scan: 25.71–28.34 years, mean ± SD: 26.75 ± 0.61 years). To estimate the social environment during early development, family socioeconomic status (SES) at birth was measured via parental interviews within 10 days after childbirth and computed as a weighted composite score considering the profession of the self-identified head of the family and the highest education held by either parent[77]. Based on this evaluation, SES was classified into high, middle, and low. SES estimates were available for all preterm participants of the BLS-26 cohort. The Parent-Infant Relationship Index (PIRI) between mother and child was used to quantify attachment-related parental concerns and feelings as well as the parent's current and anticipated relationship problems[76]. A score ranging from 0 (good) to 8 (poor parent-infant relationship) was calculated based on nurse observations neonatally and an interview at five months after birth. PIRI scores were available for 94 preterm subjects (52/42 male/female, age at scan: 25.71–28.34 years, mean ± SD: 26.75 ± 0.61 years). For adults, global cognitive performance at the age of 26 years was assessed with the "Wechsler Intelligenztest für Erwachsene", the German adaptation of the WAIS-III[140]. Ninety preterm subjects (51/39 male/female, age at scan: 25.71–28.34 years, mean ± SD: 26.71 ± 0.60 years) underwent this assessment. Results were used to derive full-scale IQ[119,120].

### Association of IBAPs with social environment and behavior (Figs. 8 and 9)
To assess the relations between CTh deviation variability (i.e., PC1) and social environmental or behavioral variables, a Spearman's rank correlation was performed (scipy, version 1.12.0).

### Moderation analysis of SES on the relation between GA and PC1 (Fig. 8)
To test whether social environment during development mediates the influence of premature birth on the principal component of individual CTh deviations (PC1), a moderation analysis was performed using the PROCESS toolbox (version 4.3)[141] for R. GA was treated as the independent variable, PC1 as the dependent variable, and SES at birth as the moderator. A test for X by M interaction was performed, and code for visualization was automatically generated. Variables were not mean-

centered. One-sided p-values of p < 0.05 were considered significant for the primary interaction effect. Bootstrapping (n = 5,000) was used to estimate 95% confidence intervals.

### Plots
All plots were generated with Python. Cortical renderings were generated using the ENIGMA toolbox[137] (https://github.com/MICA-MNI/ENIGMA). Other plots were created using matplotlib (version 3.9.2) and seaborn (version 0.13.2). Confidence intervals were estimated with bootstrap resampling (n = 10,000). For Fig. 10, brain plots were created with the Simple Brain Plot toolbox (https://github.com/dutchconnectomelab/Simple-Brain-Plot) in MATLAB 2023b (The MathWorks Inc., https://www.mathworks.com).

### Reporting summary
Further information on research design is available in the Nature Portfolio Reporting Summary linked to this article.

## Data availability
Neonatal imaging data were collected for the developing Human Connectome Project (dHCP) and are available under restricted access for data privacy reasons and by regulations of the original investigators. Access can be obtained by application to the National Institute of Mental Health (NIMH) Data Archive (https://nda.nih.gov/edit_collection.html?id=3955). Further information on how to gain access can be found on the dHCP study website (https://biomedia.github.io/dHCP-release-notes/). Children imaging data were collected for the Adolescent Brain Cognitive Development (ABCD) Study and are also available under restricted access for data privacy reasons and by regulations of the original investigators. Access can be obtained by application to the NIMH Data Archive (https://nda.nih.gov/abcd/request-access). Adult imaging data were collected in-house. Raw data are protected, are not available due to data privacy laws, and cannot be publicly shared. Access to processed data can be requested on reasonable grounds; please contact the corresponding author for details. Normative reference charts for regional CTh and SA from the BrainChart project[56] are publicly available on Github (https://github.com/brainchart/Lifespan). Charts for cerebral volume measures are also accessible via shinyapps (https://brainchart.shinyapps.io/brainchart/). Source Data files corresponding to each main and Supplementary Fig. are provided with this paper. Source data are provided with this paper.

## Code availability
All code used to perform the analyses presented here can be found at https://github.com/Melissa1909/preterm-brain-heterogeneity (https://doi.org/10.5281/zenodo.16875920)[142]. Code to re-fit pretrained normative BrainChart models are available at https://github.com/brainchart/Lifespan.

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

## Acknowledgements

M.T. and J. Schulz are financially supported by the German Academic Scholarship Foundation ("Studienstiftung des deutschen Volkes"). R.A.I.B. was supported by a British Academy Postdoctoral fellowship and by the Autism Research Trust. J. Seidlitz was supported by NIMH T32MH019112-29 and K08MH120564. Data used in the preparation of this article were obtained from the Adolescent Brain Cognitive Development (ABCD) Study (https://abcdstudy.org/), held in the NIMH Data Archive (NDA). This is a multi-site, longitudinal study designed to recruit more than 10,000 children age 9-10 and follow them over 10 years into early adulthood. The ABCD Study is supported by the National Institutes of Health and additional federal partners under award numbers U01DA041048, U01DA050989, U01DA051016, U01DA041022, U01DA051018, U01DA051037, U01DA050987, U01DA041174, U01DA041106, U01DA041117, U01DA041028, U01DA041134, U01DA050988, U01DA051039, U01DA041156, U01DA041025, U01DA041120, U01DA051038, U01DA041148, U01DA041093, U01DA041089, U24DA041123, U24DA041147. A full list of supporters is available at https://abcdstudy.org/about/federal-partners/. A listing of participating sites and a complete listing of the study investigators can be found at https://abcdstudy.org/consortium_members/. ABCD consortium investigators designed and implemented the study and/or provided data but did not necessarily participate in the analysis or writing of this report. This manuscript reflects the views of the authors and may not reflect the opinions or views of the NIH or ABCD consortium investigators. The ABCD data repository grows and changes over time. The ABCD data used in this report came from https://nda.nih.gov/study.html?id=1299. Furthermore, data were obtained from the developing Human Connectome (dHCP) Project (https://www.developingconnectome.org/), also held in the NDA. Aiming to create a 4-dimensional connectome of early life, longitudinal, fetal and neonatal data were acquired in a multi-site design. The dHCP project was funded through a Synergy Grant by the European Research Council (ERC) under the European Union's Seventh Framework Program (FP/2007-2013) / ERC Grant Agreement no. 319456. The work was also supported by the NIHR Biomedical Research Centers at Guys and St Thomas NHS Trust. We are grateful to the families who generously supported this trial. We would like to acknowledge core support for data acquisition was provided by the Wellcome/EPSRC Center for Medical Engineering [WT 203148/Z/16/Z]. We are also thankful to the WU-Minn-Oxford Human Connectome Project consortium (1U54MH091657-01) for access to their computing resources. The Bavarian Longitudinal Study (BLS) is an ongoing longitudinal, multi-site study that aims at the characterization of brain development and aging after preterm birth led by D. Wolke, P. Bartmann, and C. Sorg. Data collection of the BLS was supported by grants from the German Federal Ministry of Education and Science (BMBF): PKE24, JUG14, 01EP9504, and 01ER0801. D. Wolke and data collection of BLS at age 38 years are supported by the UK Research and Innovation (UKRI) Research Frontier Grant (EP/X023206/1) under the UK governments Horizon Europe funding guarantee for an ERC Advanced Grant and a UK Innovation Service grant (UKRI: 10037942) as an associated partner to the Horizon Europe: Risk and Resilience in Developmental Diversity and Mental Health (R2D2-NH) project grant. We thank all current and former members of the BLS Group who contributed to general study organization, recruitment, data collection, and management as well as subsequent analyses. We thank the staff of the Department of Neuroradiology in Munich and the Department of Radiology in Bonn for their help in data collection. Most importantly, we thank all study participants and their families for their efforts to take part in this still ongoing study. The BrainChart project is based on a large number of datasets. We acknowledge the invaluable contribution to this effort made by several openly shared MRI datasets. The Allen Human Brain Atlas provides detailed maps of regional gene expression in the human brain. It was supported by the following grants of the National Institute of Health (NIH): R33 DA027644/DA/NIDA NIH HHS/United States, U54 MH091657/MH/NIMH NIH HHS/United States, 1U54MH091657/MH/NIMH NIH HHS/United States, 4R33DA027644/DA/NIDA NIH HHS/United States. Furthermore, we thank Clara Kretschmer for providing help with figure generation using Inkscape.

## Author contributions

Conception and design: M.T., J. Seidlitz, M.A.D.B., D.M.H., C.S. Data acquisition and provision: J. Seidlitz, A.M., B.S.-K., M.A.D.B., J. Schulz, L.D., R.A.I.B., A.A.-B., C.A., G.B., J.S.D.A., R.B., C.Z., M.D., H.B., P.B., D.W., D.M.H., C.S. Analysis and interpretation: M.T., A.N. Manuscript drafting: M.T., C.S. Manuscript revision: All authors. Supervision: D.M.H., C.S.

## Funding

## Competing interests

The authors declare the following financial interests/personal relationships which may be considered as potential competing interests: M.T. and J. Schulz reports financial support was provided by the German Academic Scholarship Foundation ("Studienstiftung des deutschen Volkes"). C.S., A.M., and D.M.H. report financial support was provided by the German Research Foundation ("Deutsche Forschungsgemeinschaft"; DFG). P.B., D.W., and C.S. report financial support was provided by German Federal Ministry of Education and Science. D.W. and P.B. report financial support was provided by EU Horizon 2020. C.S., D.M.H., and B.S.-K. report financial support was provided by Commission for Clinical Research, Technical University of Munich. Data collection for the Bavarian Longitudinal Study from birth to 26 years was supported by grants from the German Federal Ministry of Education and Science (BMBF). D.W. and data collection of BLS at age 38 years are supported by the UK Research and Innovation (UKRI) Research Frontier Grant under the UK governments Horizon Europe funding guarantee. J. Seidlitz, R.A.I.B., and A.A.-B. hold equity in and J. Seidlitz and R.A.I.B. are directors of Centile Bioscience. Other authors have no known competing financial interests or personal relationships that could have appeared to influence the work reported in this paper.

## Additional information

[1]Technical University of Munich, School of Medicine and Health, Department of Diagnostic and Interventional Neuroradiology, TUM Klinikum Rechts der Isar, Munich, Germany. [2]Technical University of Munich, School of Medicine and Health, TUM-NIC Neuroimaging Center, TUM Klinikum Rechts der Isar, Munich, Germany. [3]Department of Child and Adolescent Psychiatry and Behavioral Science, The Children's Hospital of Philadelphia, Philadelphia, PA, USA. [4]Lifespan Brain Institute, The Children's Hospital of Philadelphia and Penn Medicine, Philadelphia, PA, USA. [5]Department of Psychiatry, University of Pennsylvania, Philadelphia, PA, USA. [6]Institute for Translational Medicine and Therapeutics, University of Pennsylvania, Philadelphia, PA, USA. [7]Center for Neuropathology and Prion Research, Ludwig-Maximilians-Universität Munich, Munich, Germany. [8]Departments of Psychiatry and Anatomy & Physiology, The University of Melbourne, Melbourne, VIC, Australia. [9]Department of Psychiatry, Brigham and Women's Hospital, Harvard Medical School, Boston, MA, USA. [10]Department of Psychology, University of Cambridge, Cambridge, UK. [11]Developmental Imaging, Murdoch Children's Research Institute, Melbourne, VIC, Australia. [12]Department of Pediatrics, University of Melbourne, Melbourne, VIC, Australia. [13]Division of Development and Growth, Department of Women's, Children's, and Adolescent Health, University Hospitals of Geneva, Geneva, Switzerland. [14]Department of Diagnostic and Interventional Radiology, University Hospital Bonn, Bonn, Germany. [15]Department for Nuclear Medicine, University Hospital Bonn, Clinical Functional Imaging Group, Bonn, Germany. [16]Department of Neonatology and Pediatric Intensive Care, University Hospital Bonn, Bonn, Germany. [17]Department of Psychology, University of Warwick, Coventry, UK. [18]Warwick Medical School, University of Warwick, Coventry, UK. [19]Technical University of Munich, School of Medicine and Health, Department of Psychiatry, Munich, Germany. ✉e-mail: melissa.thalhammer@tum.de

