## [Transparent Peer Review file · Nature Communications]

Heterogeneous, temporally consistent, and plastic brain development after preterm birth

Corresponding Author: Ms Melissa Thalhammer

Version 0:

Reviewer comments:

Reviewer #1

(Remarks to the Author)

In this manuscript Thalhammer and colleagues characterise the development of regional structural brain patterns after preterm birth, using publicly available cohorts of neonates (dHCP), children (ABCD-10/ABCD-12) and adults (BLS-26/BLS-38). They mostly focus on cortical thickness development, but also include supplementary analyses of surface area (SA), white matter, grey matter, and subcortical grey matter volumes. Based on their results, they suggest heterogeneous abnormal patterns of development after preterm birth, but consistent commonalities between subjects, which they attribute to the same underlying mechanisms of injury and plasticity.

In my view this is a nice addition to the preterm literature, making good use of publicly available data and extending previous work characterising the normative development of the brain. The findings presented are not particularly controversial, mostly reinforcing the prevalent understanding of the effects of preterm birth. I think that the methods are overall technically sound, though there are a few elements that the authors should clarify. Similarly, I am a bit unsure about some of the research design decisions and their justification. In my opinion, the authors could improve the impact of the manuscript by clarifying a few methodological aspects and using the available data to perform additional analysis filling the analytical gaps in their study, i.e., they present longitudinal data only for SA (but not CTh), and they analyse association of IBAPs with SES and cognitive outcomes only in adults (but not in neonates or children). Please see detailed comments/suggestions below.

Main suggestions:

-Results L96-98: Were there any ROIs with increased CTh in adults? It would also be interesting to show these results in neonates and children in the main body of the manuscript. These results (i.e. average differences) are the cornerstone for the rest of the manuscript, and I suggest to incorporate in a single figure in the main body of the manuscript, for all the 5 samples assessed, allowing direct comparison of preterm effects from neonatal to adulthood. I was also surprised that SA average differences are not presented together with CTh. I recommend to present CTh and SA results together, or change the introduction so it is not implied as one of the main aims of the study (e.g. L69: "First, we employed normative modeling to describe regional CTh and SA developmental trajectories"). In any case, I would also recommend including a table with specific effect size and statistics for each ROI in supplementary material.

-Results L108: The authors justify the validity of their normative model because more than 85% of full-term individuals resided within the normative range for any given region. However, if I understand correctly, the original model (Bethlehem et al 2022) was trained with dHCP term controls. If so, it wouldn't be surprising that the model fits the data that was used to train it. I wonder if this has repercussions in the way the preterm deviations are characterised. Is it fair to compare unseen data (preterm) to the data used to train the model (term)?

-Results L176-178: "We focused on SA rather than CTh due to previous studies demonstrating that FreeSurfer estimates of "raw" regional CTh have substantially higher overall variability compared to SA" -> I am not sure I am convinced with this justification. The authors have focused mostly on CTh, so it seems unfair that they characterise individual longitudinal stability only in SA.

-Results L214: Characterisation of SES effects is very relevant, and I really appreciate the inclusion of this analysis. However, it would be even more interesting to show the (potentially different?) effect of SES in IBAPs through the lifespan. I

suggest the authors consider including SES variables available from the dHCP and ABCD data and assess whether IBAP-SES associations are similar in infancy and childhood.

-Results L248 (and Introduction L77): why not characterise the association of IBAPs with outcomes earlier on? As far as I understand both the dHCP and the ABCD provide outcome data (e.g. Bayley cognitive outcomes at 18-month follow up for dHCP). I think this would be a great addition to the study, and I encourage the authors to include this additional analysis.

-Methods L380: I wonder why the authors didn't use the 3rd (or 4th) data release for the dHCP. Sample size is increased by hundreds from 2nd to 3rd releases (up to 783 subjects available in the latest release).

Other questions/suggestions:

-Introduction: I think it could be clearer why going beyond the "group average" paradigm is so important, and what could it bring conceptually and clinically to identify within preterm group developmental patterns - in particular for readers not familiar with preterm literature.

-Introduction L40-41: could you clarify why accelerated WM maturation could be an issue?

-Introduction L46: clarify what are "average dysmaturation outcomes". Perhaps this whole sentence could be clearer.

-Introduction L68: SA is introduced for the first time. Consider introducing it with CTh earlier, and why is it relevant to study in relation to preterm birth?

-Results L92: is a t-test the most appropriate statistical approach here? Don't the authors need to include relevant covariates (e.g. corrected age at scan)? Does the data fulfil t-test assumptions of homoscedasticity / normal distributions? Methods L483-486 suggest linear regression models were used.

-Figure 2d (and Figure S1b_2, etc): what is bankssts? Please include actual ROI names instead of variable names or define in the legend (same for S1c, etc)

-Fig 2b / L118: Term controls seem to be far from the median in the model presented. Is this an issue? Figure S2 seems to be missing the fitting of term participants, which is relevant to assess the goodness of fit of the model.

-Results L130: "The overall prevalence of at least one extranormal deviation varied with age from only less than 65 % in preterm neonates to over 85 % of preterm children and adults, suggesting developmental and/or environmental influences on brain structure." -> I am not sure that this last sentence is adding much

-Results L132: "27 % of preterm individuals sharing deviations in any given region". A bit unclear if this is the same result presented in L115.

-Results L160: "indicates that earlier birth increases the likelihood of structural alterations" -> as far as I understand the authors are not looking at likelihood of alterations but rather to what extent they share locations.

-Results L170: "Results demonstrate that GA has a substantial impact on the extent" -> I would be careful with the causal inference here. Isn't it possible that other factors such as BW and perinatal clinical complications are also related to the extent of atypical development?

-Results L253: Is it justified to present one-sided p-values? Please clarify whether p-values are corrected for multiple comparisons.

-Discussion L268: is it really "influenced" by – or mediated / attenuated?

-Discussion L285: "suggesting individual developmental trajectories of the brain after preterm birth." -> Clarify these are trajectories of brain structural features

-Methods (overall): Perhaps improve consistency of "preterm" / "very preterm", slightly confusing which group is included. E.g. L416; "Since this estimation is rather vague, we only included subjects born more than 7 weeks premature (i.e., gestational age \leq 32 weeks) as preterm". $GA \leq 32$ is very preterm, but in introduction it is suggested that the study focus on preterm birth (including late-onset), e.g. L29: "preterm birth is defined as birth before 37 weeks of gestation". Indeed most of preterm neonates part of the dHCP are late preterm ($32 < GA < 37$).

-Methods L481: I am not sure I understand the justification for not harmonising the dHCP and BLS-38. Difference in size, sequences (T1 vs T2?), and preprocessing pipelines seems enough justification for site harmonisation.

-Methods L510: the choice of 30 weeks could be better justified. Usually 28 weeks (extreme preterm) or 32 weeks (very preterm) are clinically used as thresholds.

(Remarks on code availability)

Code available seems to be complete and well documented.

Reviewer #2

(Remarks to the Author)

(Remarks on code availability)

Reviewer #3

(Remarks to the Author)

This work presents a comprehensive investigation into the effects of preterm birth on individual deviations in brain structure from a normative growth pattern. By leveraging three brain MR scan datasets covering preterm neonates, children, and adults, the authors reveal significant variability in deviation maps between individuals, yet demonstrate relatively consistent deviations within individuals over time, particularly in cortical thickness and surface area. They also explore the neurobiological basis and cognitive relevance of preterm-related individual brain deviations by incorporating a brain cellular atlas, social environmental factors, and cognitive assessments. The topic is highly engaging, and the authors offer insights into understanding the atypical developmental patterns of preterm infants, which may facilitate the identification of stable, individualized neuroimaging biomarkers for cognitive and behavioral outcomes related to preterm birth. This work employs diverse brain MRI datasets, strict scan harmonization approaches, and is presented with clear and effective writing, reflecting a high level of quality. However, I have some suggestions that may help further improve the manuscript before publication.

Major suggestions:

- 1) Some of the results are initially presented in a highly descriptive manner, relying heavily on percentage-based summaries, which limits the depth of the findings. More rigorous quantitative analyses should be emphasized. For example, the authors note that in the BLS-26 cohort, a maximum of 27% of adults share the same extreme abnormal region. This could be better captured using Spearman correlations by providing the average Spearman coefficient for each brain region across the population. While they do include a correlation matrix of overall coefficients in Figure 2, it does not show the similarity of profiles in each brain region. Similar percentage-based analyses are used for metrics like WMV and GMV, making cross-metric comparisons appear superficial. In the results on finding constant deviations within the same individual, similar percentage-based analyses are used again. Some measurements, such as ICC, could provide more quantitative findings. Additionally, when replicating individual deviation patterns with another normative model (Supplementary Material S2), they did not report the correlation coefficients between these patterns and the main text's results. However, I am not suggesting that all of their results are overly descriptive. Many quantitative analyses are very rigorous and interesting, such as the correlations between the number of abnormal regions and GA, and the correlations between the deviation-cell-type association and GA.
- 2) The authors are mainly concerned with binary brain areas and the number of deviations, but the intensity of deviations (i.e., weighted deviation profiles) is also important and should not be overlooked.
- 3) More strict FDR corrections may be needed. The authors control for family-wise error using FDR correction in the group difference analysis. However, all subsequent analyses for the deviation maps should also consider such corrections.

Minor suggestions:

- 1) I believe the results of all three cohorts are of equal importance. Some figures in the supplementary material could be condensed and placed in the main text (e.g., Figures S1, S2, S5, S6-7).
- 2) Figure 1 is too large and complex. Much of the information is relatively repetitive, such as the individual deviation maps and some result plots.
- 3) There are some typos in the manuscript. For example, in Line 72, what does "see 1" mean? In Line 87, where is (ii)?

(Remarks on code availability)

Reviewer #4

(Remarks to the Author)

The paper by Thalhammer and co author, titled "Heterogeneous, temporally consistent, and plastic brain development after preterm birth" and submitted to the journal Nature Communications, presents study on brain morphometry following preterm birth at three distinct time points: term equivalent age, school-age, and adulthood. While there are an impressive amount of

analyses, the paper falls short. I have several major comments listed below.

Stylistically the first paragraph contains multiple thoughts, compressed into a hard to follow paragraph. I would suggest breaking up this paragraph into several simpler paragraphs, each with a strong topic sentences, to guide the reader.

This extends throughout the paper. It is hard to read and follow

It would be helpful to better compare this work to Dimitrova R, et al 2021. The approaches are very similar—characterize individual deviations from a normative growth chart and move away from population averages. The conclusions from that paper are nearly identical to this paper, but are largely ignored in the current manuscript, which makes the current work appear much more novel.

Many would disagree with this statement: “This neurocognitive heterogeneity is widely assumed to be reflected in related heterogeneity of neurodevelopmental processes”. Preterm birth is known to be heterogeneous disorder due to the many different causes of it, where factors range from maternal to fetal. For example, induced preterm birth is a treatment for high risk of still birth. This may have very different brain based outcomes than spontaneous preterm birth due to an incompetent cervix. It may be worth discussing how the many difference causes of preterm birth may account for the observed results. Romero R, Dey SK, Fisher SJ. Preterm labor: one syndrome, many causes. *Science*. 2014 Aug 15;345(6198):760-5. doi: 10.1126/science.1251816. Epub 2014 Aug 14. PMID: 25124429; PMCID: PMC4191866.

Were the dHCP and ABCD data used in creating the reference brain charts? If so, how does this data leakage affect the results?

Can the authors justify split the groups into early and late birth at 30 weeks? Typically the breakdown would be extremely preterm birth (<28 weeks), very preterm birth (28-32 weeks), and moderate to late preterm birth (>32 weeks). The authors break down appears rather ad hoc.

Do the authors have detailed information about brain injuries in any of the cohorts? How many participants had intraventricular hemorrhage (and what grade), periventricular Leukomalacia, cerebellar hemorrhage, etc? Major brain injuries are common in preterms and associate with altered brain morphometry and cognitive outcomes.

The authors' thresholding approach to binarizing deviations likely increases the dissimilarity between individuals. For example, Taylor et al showed that thresholded results look more dissimilar to each other than using maps that retain all the information. This is because thresholding is a non linear operation that enforces discontinuities. How similar are the deviation maps if they are not binarized?

Taylor PA, Reynolds RC, Calhoun V, Gonzalez-Castillo J, Handwerker DA, Bandettini PA, Mejia AF, Chen G. Highlight results, don't hide them: Enhance interpretation, reduce biases and improve reproducibility. *Neuroimage*. 2023 Jul 1;274:120138. doi: 10.1016/j.neuroimage.2023.120138. Epub 2023 Apr 27. PMID: 37116766; PMCID: PMC10233921.

Given that the term group shows a large amount of large deviations, how similar are they to each other and the preterm group? In other words, is the amount of individual differences specific to the preterm group?

There has been many criticisms of the spin test. Exchangeability is violated in several ways. It assumes complete stationarity. Also there is a need to interpolate which also violates exchangeability.

I am not sure how the results support a theranostic approach in preterm birth or even integrating brain charts. Most preterm clinics would already recommend early cognitive and social interventions, given the well-known risk factors.

The lack of long term longitudinal data is more of a weakness the authors suggest. Certainly the ABCD and BLS allow to test for stability, but the follow up is short compared to the development period. For example, one would not expect a large amount of change between 26 and 38 years. Longitudinal changes over infancy or from infancy through childhood would be important to know to show the value of a growth chart type approach. The dHCP has longitudinal data for the preterm cohort.

(Remarks on code availability)

Version 1:

Reviewer comments:

Reviewer #1

(Remarks to the Author)

I commend the authors for the thorough revision of their manuscript - in particular the inclusion of the latest dHCP release data, and the integration of SA and CTh measurements in the body of the manuscript. Most of the technical limitations are now appropriately acknowledged in their discussion section.

(Remarks on code availability)

The authors provide the code to replicate their results and this is a useful tool for the community.

Reviewer #2

(Remarks to the Author)

(Remarks on code availability)

Reviewer #3

(Remarks to the Author)

The authors have addressed most of my original concerns. While a few points would benefit from following minor refinements.

1) In Figure S9 (SA metric, dHCP cohort), the reported Spearman correlation (mean $r = 0.5$, max $r = 0.75$) of each brain region for their deviation patterns across subjects is explained as “several regions not showing any extranormal deviations, which may increase the influence of a few deviating regions.” This is somewhat odd, as the correlations are performed on subject-wise deviation profiles—not region-wise, as in the main results. I would suggest that a plausible explanation is that although there is substantial heterogeneity in the specific regions showing deviations across individuals, certain brain regions show intrinsic covariation in their deviation patterns across the cohort. This does not undermine the main conclusion of their findings, but if the authors agree, a brief note in the discussion part may help readers better understand the nature of these heterogeneous deviations in preterm.

2) At line 154, the description of Figures S9–S10 is incorrect. These figures present results regarding the relationships between regional deviations across individuals, rather than results of non-binarized deviations.

3) In Figure 6, the presentation of location consistency for a few selected brain regions in the main text may give a somewhat selective impression. As shown in the supplementary materials, some regions such as the temporal lobe, do not display particularly stable deviations over time. This is also consistent with the results of ICC analysis. I highly appreciate that the authors included ICC analyses in the main text and emphasized the heterogeneity in the longitudinal stability of regional deviation. However, including the full distribution of regional ICC values in Figure 6 (such as the plot of cortical surface in Figure 7) would provide a more comprehensive perspective.

4) I did not find any description on Figure 2h. Please either include a description or remove it.

5) A few minor typos remain. For instance, line 124 reads “with. severe perinatal brain injury;”. A careful proofreading would help.

(Remarks on code availability)

Reviewer #4

(Remarks to the Author)

The authors has answered my question.

(Remarks on code availability)

Response to reviewers

Revision for the article

“Heterogeneous, temporally consistent, and plastic brain development after preterm birth”

by Thalhammer et al.

General points for all reviewers

Revised results for the dHCP cohort

We followed the suggestion by Reviewer 1 to use a more recent release of the dHCP dataset (i.e., 3rd release), resulting in the addition of more than 200 additional neonates to the study. However, this update to a later data release was accompanied by a major improvement in the M-CRIB-S parcellation, improving accuracy of brain segmentation and surface extraction¹. In particular, compared to the original software release, a whole brain labeling compatible with adult FreeSurfer was introduced using different registration tools and an improved surface extraction technique was applied. We refer to the original publications for further details^{1,2}. While both larger sample size and updated analysis pipeline resulted in different average dysmaturational outcomes compared to the original manuscript, the general finding of spatial heterogeneous cortical deviations was supported (see Fig. 3 and Fig. 4).

Adding surface area SA results in parallel to cortical thickness CTh results

Reviewer 1 suggested to use both CTh and SA as equally relevant outcomes for testing our hypotheses instead of focusing on CTh for most analyses. We followed this suggestion and added corresponding findings at each level of our analysis, which extended both results and the number of figures presented in the main manuscript.

Adding the results of preterm neonates and children in parallel to preterm adults

Lastly, we followed the recommendation of Reviewer 3 to report the results of all three investigated cohorts, i.e., dHCP, ABCD, and BLS, in the main text to treat them as equally important. This also resulted in an extension of results and figures.

Responses to Reviewer #1

Comment 1:

In this manuscript Thalhammer and colleagues characterise the development of regional structural brain patterns after preterm birth, using publicly available cohorts of neonates (dHCP), children (ABCD-10/ABCD-12) and adults (BLS-26/BLS-38). They mostly focus on cortical thickness development, but also include supplementary analyses of surface area (SA), white matter, grey matter, and subcortical grey matter volumes. Based on their results, they suggest heterogeneous abnormal patterns of development after preterm birth, but consistent commonalities between subjects, which they attribute to the same underlying mechanisms of injury and plasticity.

In my view this is a nice addition to the preterm literature, making good use of publicly available data and extending previous work characterising the normative development of the brain. The findings presented are not particularly controversial, mostly reinforcing the prevalent understanding of the effects of preterm birth. I think that the methods are overall technically sound, though there are a few elements that the authors should clarify. Similarly, I am a bit unsure about some of the research design decisions and their justification. In my opinion, the authors could improve the impact of the manuscript by clarifying a few methodological aspects and using the available data to perform additional analysis filling the analytical gaps in their study, i.e., they present longitudinal data only for SA (but not CTh), and they analyse association of IBAPs with SES and cognitive outcomes only in adults (but not in neonates or children). Please see detailed comments/suggestions below.

Response 1:

We thank the Reviewer for the thorough and benevolent revision of our work.

Comment 2:

Results L96-98: Were there any ROIs with increased CTh in adults? It would also be interesting to show these results in neonates and children in the main body of the manuscript. These results (i.e. average differences) are the cornerstone for the rest of the manuscript, and I suggest to incorporate in a single figure in the main body of the manuscript, for all the 5 samples assessed, allowing direct comparison of preterm effects from neonatal to adulthood. I was also surprised that SA average differences are not presented together with CTh. I recommend to present CTh and SA results together, or change the introduction so it is not implied as one of the main aims of the study (e.g. L69: "First, we employed normative modeling to describe regional CTh and SA developmental trajectories"). In any case, I would also recommend including a table with specific effect size and statistics for each ROI in supplementary material.

Response 2:

We thank the Reviewer for the helpful suggestions regarding the presentation of average dysmaturation outcomes.

First, the Reviewer asked whether there were any ROIs with increased CTh in adults. As shown in Fig. 2a, the isthmus cingulate, posterior cingulate, and medial orbitofrontal ROIs exhibited increased CTh in the BLS-26 preterm cohort compared to full-term controls. We agree that this may not have been sufficiently emphasized in the main text and thus we have revised the relevant parts to clarify this finding.

Second, we also agree that presenting average SA dysmaturation outcomes alongside CTh for all cohorts in the main text would strengthen the foundation for our subsequent focus on individual heterogeneity. To address this, we have moved the previous Supplementary Figures

S1 and S3 into the main body of the manuscript (now Figures 3 and 4) to provide a more complete view of average dysmaturation outcomes for both CTh and SA across development.

We have adapted the Results section to address the aforementioned aspects:

To replicate previous models of average dysmaturation, we used [...] to analyze mean CTh and SA differences between preterm and full-term individuals for each cohort for 34 cortical regions of the Desikan-Killiany parcellation³. Consistent with previous findings⁴⁻⁷, regional mean CTh was increased in most areas except for the occipital and inferior temporal lobes in preterm neonates. In contrast, while it was restrictedly decreased in frontal and temporal regions in children. In preterm adults, CTh was and widely decreased in lateral associative and primary cortices and increased in cingulate areas in preterm adults (Fig. 2a: BLS-26; Fig. 3: other cohorts, Supplementary Tables S4). SA decreases extended to frontal, temporal, and parietal areas across the lifespan. Only the temporal pole showed increased SA in preterm children and adults compared to controls (Fig. 4, Supplementary Tables S4).

Third, we followed the Reviewer's recommendation to include Supplementary Tables with specific effect sizes and statistics for each average dysmaturation outcome analysis (i.e., group comparison for each ROI between full-term and preterm subjects). Supplementary Tables S4a-o were added. For illustration purposes, the average dysmaturation outcomes for cortical thickness in the BLS-26 cohort are shown below (Table S4j). Significant differences ($p_{FDR} < 0.05$) are highlighted in bold.

Table S4j: Average dysmaturation outcomes of cortical thickness in the BLS-26 cohort

ROI	T-statistic	Cohen's d	p-value	p_{FDR}
CTh_bankssts	-2.969	0.428	0.003	0.013
CTh_caudalanteriorcingulate	1.010	-0.135	0.314	0.431
CTh_caudalmiddlefrontal	-1.041	0.155	0.299	0.431
CTh_cuneus	-0.565	0.094	0.572	0.671
CTh_entorhinal	-1.347	0.194	0.179	0.305
CTh_fusiform	-2.666	0.367	0.008	0.022
CTh_inferiorparietal	-4.632	0.665	<0.001	<0.001
CTh_inferiortemporal	-1.746	0.259	0.082	0.165
CTh_isthmuscingulate	3.071	-0.416	0.002	0.012
CTh_lateraloccipital	-0.054	0.033	0.957	0.975
CTh_lateralorbitofrontal	-0.458	0.076	0.648	0.710
CTh_lingual	-1.220	0.184	0.224	0.363
CTh_medialorbitofrontal	3.398	-0.466	0.001	0.005
CTh_middletemporal	-6.518	0.929	<0.001	<0.001
CTh parahippocampal	-0.478	0.082	0.633	0.710
CTh_paracentral	1.630	-0.241	0.105	0.198
CTh_parsopercularis	-1.900	0.275	0.059	0.125
CTh_parsorbitalis	-6.108	0.868	<0.001	<0.001
CTh_parstriangularis	-2.909	0.415	0.004	0.014
CTh_pericalcarine	0.877	-0.119	0.381	0.499
CTh_postcentral	-2.251	0.312	0.026	0.058
CTh_posteriorcingulate	2.393	-0.291	0.018	0.043

CTh_precentral	-2.752	0.395	0.006	0.018
CTh_precuneus	-1.003	0.152	0.317	0.431
CTh_rostralanteriorcingulate	0.823	-0.111	0.411	0.517
CTh_rostralmiddlefrontal	-3.565	0.514	<0.001	0.003
CTh_superiorfrontal	0.799	-0.111	0.425	0.517
CTh_superiorparietal	-2.858	0.409	0.005	0.015
CTh_superiortemporal	-1.029	0.152	0.305	0.431
CTh_supramarginal	-3.749	0.534	<0.001	0.002
CTh_frontalpole	-0.305	0.053	0.760	0.808
CTh_temporalpole	-1.443	0.195	0.151	0.270
CTh_transversetemporal	0.031	-0.003	0.975	0.975
CTh_insula	-2.969	0.422	0.003	0.013

Comment 3:

Results L108: The authors justify the validity of their normative model because more than 85% of full-term individuals resided within the normative range for any given region. However, if I understand correctly, the original model (Bethlehem et al 2022) was trained with dHCP term controls. If so, it wouldn't be surprising that the model fits the data that was used to train it. I wonder if this has repercussions in the way the preterm deviations are characterised. Is it fair to compare unseen data (preterm) to the data used to train the model (term)?

Response 3:

We thank the Reviewer for raising this important point regarding the validity of using overlapping datasets for both model training and application. We start with providing more details on training of the original model by Bethlehem et al. followed by clarifying how it was adapted in our current study.

The normative modeling framework by Bethlehem et al. was trained on a large, population-level reference dataset of more than 100,000 subjects that also included individuals from the dHCP and ABCD studies. Importantly, for original model training, participants were selected based on general health criteria (e.g., to never been diagnosed with a mental health disorder according to the parental response to the ABCD screening and risk questionnaire) and were **not stratified by gestational age**, i.e., both term- and preterm-born individuals were included to the reference cohort as “cognitively normal”. Furthermore, only chronological age (not gestational or postnatal age) was included as a covariate into the model, consistent with the authors' goal of creating developmental reference charts across the lifespan, rather than focusing specifically on prematurity.

In contrast, we aimed to assess how individuals born preterm may deviate from typical developmental trajectories. To do so, we adapted the pretrained model from Bethlehem et al. by refitting it to the dHCP and ABCD data, but with some key differences: we treated each dataset as a new site (i.e., “dHCP_new”, “ABCD_new”) and included only term-born individuals when estimating the site-specific random effects. This step ensures that any potential differences due to imaging site or preprocessing pipeline are accounted for **without including preterm data into the model fitting**. By this means, the application data of interest in our study (i.e., preterm individuals) are not part of the refitting process.

Since Bethlehem et al. demonstrated in their original work that leave-one-study-out models yielded nearly identical model parameters compared to the full model⁸, we argue that the overall model shape is only minimally influenced by inclusion or exclusion of individual cohorts and that refitting of the original model to term dHCP or ABCD data as well as its adaptation to

preterm data is not influenced by the fact that dHCP and ABCD were used in the original training process.

To clarify that, we have included the following description in the Methods section:

Normative reference charts for bilateral regional CTh, SA, as well as for global measures (i.e., GMV, sGMV, and WMV) were obtained from the BrainChart project⁸, in which GAMLSS-based normative models were computed over the human lifespan as a function of age and sex ~~from about 100,000 MRI scans~~. The original model was trained on a large, population-level reference dataset of more than 100,000 subjects that also included individuals from the dHCP and ABCD studies. Importantly, for original model training, participants were selected based on general health criteria and were not stratified by gestational age, i.e., both term- and preterm-born individuals were included to the reference cohort as “cognitively normal”. Furthermore, only chronological age (not gestational or postnatal age) was included as a covariate into the model.

To adapt the reference charts to the datasets used for this study, random effects of study were calculated only based on full-term individuals and by treating dHCP and ABCD as a new site (i.e., “dHCP_new”, “ABCD_new”). This step ensures that any potential differences due to imaging site or preprocessing pipeline are accounted for without including preterm data into the model fitting. By this means, the application data of interest in the present study (i.e., preterm individuals) are not part of the refitting process. Since Bethlehem et al. demonstrated in their original work that leave-one-study-out models yielded nearly identical model parameters compared to the full model⁸, we argue that the overall model shape is only minimally influenced by inclusion or exclusion of individual cohorts and that refitting of the original model to term dHCP or ABCD data as well as its adaptation to preterm data is not influenced by the fact that dHCP and ABCD were used in the original training process.

Comment 4:

Results L176-178: “We focused on SA rather than CTh due to previous studies demonstrating that FreeSurfer estimates of “raw” regional CTh have substantially higher overall variability compared to SA” -> I am not sure I am convinced with this justification. The authors have focused mostly on CTh, so it seems unfair that they characterize individual longitudinal stability only in SA.

Response 4:

We appreciate the Reviewer’s concern regarding our decision to characterize individual longitudinal stability only in SA IBAPs, despite the broader focus on CTh throughout the manuscript. As the Reviewer mentioned, our rationale for prioritizing SA in the main analysis in this case is based on prior evidence indicating that FreeSurfer-derived CTh estimates show substantially higher test-retest variability than SA estimates, which might explain why extranormal CTh deviations are longitudinally less stable compared to SA. Due to space limitations, we have decided to still present only SA longitudinal stability in the main text. However, we agree that it is important to present both metrics. To address this, we have now included the analysis of longitudinal CTh stability in the Supplement (Supplementary Fig. S14 and S15, Supplementary Tables S5c-d). These results are briefly described in the main text to provide a more balanced view across both cortical metrics:

Results for longitudinal consistency of CTh can be found in the Supplement (Supplementary Figures S14 and S15, Supplementary Tables S5c-d).

[...]

To quantitatively evaluate whether IBAPs of preterm individuals remain constant over time, we computed the intraclass correlation coefficient (ICC) for each brain region. For all regions except the frontal and temporal poles, which are known to be less reliably estimated by FreeSurfer^{3,9}, ICC was higher than 0.837 for children and higher than 0.803 for adults (Supplementary Tables S5a-b), indicating longitudinal stability of extranormal deviations. In contrast, CTh IBAPs were less stable across childhood and adulthood, with ICCs ranging from 0.394 to 0.876 in children and from 0.426 to 0.844 in adults (with the temporal pole being an extreme outlier, ICC = -0.093; Supplementary Tables S5c-d).

Comment 5:

Results L214: Characterisation of SES effects is very relevant, and I really appreciate the inclusion of this analysis. However, it would be even more interesting to show the (potentially different?) effect of SES in IBAPs through the lifespan. I suggest the authors consider including SES variables available from the dHCP and ABCD data and assess whether IBAP-SES associations are similar in infancy and childhood.

Response 5:

We appreciate that the Reviewer brings up this interesting question of how SES effects IBAPs throughout the lifespan.

Since the facets used for SES calculation tremendously impact correlations with brain measures¹⁰, we have focused on parental education as a proxy for SES to maintain consistency with the previous analysis in the BLS-26 dataset. However, we did not fully replicate the SES measure used in BLS, which also depends on occupation of the self-identified head of the family. As it was calculated in the 1980s, it relied heavily on now outdated occupational classifications (primarily based on the father's job) and lacked relevance and completeness for the current datasets. Therefore, we focused on parental education as an estimate for SES in the dHCP and ABCD datasets.

For the dHCP cohort, available variables were limited to questionnaires about the age at which the mother and father left full-time education. In a discussion with a member of the dHCP team we got to know that parental education data are inconsistently collected and often noisy, limiting interpretability. For the ABCD cohort, more reliable data for parent education are available (i.e., responses to the question "What is the highest grade or level of school you have completed or the highest degree you have received?"). Based on this, we estimated SES and included SES-IBAP correlations for preterm children in Fig. 8a. The Methods section "Perinatal, social environment, and behavioral variables" as well as the Results section "Impact of early social environment on IBAPs" were adapted accordingly.

Perinatal, social environmental, and behavioral variables

For the dHCP cohort, available variables for social environment were limited to questionnaires about the age at which the mother and father left full-time education, which does not automatically translate to the quality of education. [...]

For the ABCD cohort, parental education based on responses to questionnaires ("What is the highest grade or level of school you have completed or the highest degree you have received?") was used as an estimate for early social environment.

Impact of early social environment on adult IBAPs

In addition to consistency in injury-induced dysmaturation processes, we also expected a consistent plastic effect of the early environment on individual structural deviations after preterm birth (Fig. 1e). We addressed this question in children (ABCD-10) and adults (BLS-26). Data from the lifelong BLS cohort allowed us to examine whether an individual's social environment during childhood might contribute to heterogeneous

deviation profiles in adulthood. To summarize CTh and SA deviations across the cortex, we derived a first principal component (PC1) across all 34 regional CTh or SA deviation scores, respectively, for preterm individuals of each investigated cohort adults aged 26 years, capturing 37.2 % of the total variance in CTh deviation scores. Loadings for PC1 were positive across all 34 regions and both cohorts (Supplementary Figure S17), indicating that this component represents a global trend in CTh deviations across the cortex, with lower PC1 scores reflecting lower deviation scores relative to population norms. As representatives of the early developmental environment, we used family socio-economic status (SES) as a general, widely used measure of the economic and sociological environment components of a developing child¹¹, and the Parent-Infant-Relationship Index (PIRI) as a measure of parental social environment specifically (only available for BLS-26)¹².

In children of the ABCD-10 cohort, SES was estimated based on parental education level based on parental interviews. In adults of the lifelong BLS study, SES was assessed neonatally as a composite score of family profession and parental education based on parent interviews and was classified into low, middle, and high¹³. PIRI describes attachment-related parental concerns and the mother's current and anticipated relationship problems¹². A sum score from 0 (good parent-infant relationship) to 8 (poor parent-infant relationship) was calculated based on nurse observations neonatally and an interview at five months after birth. This allowed us to examine whether an individual's social environment during childhood might contribute to heterogeneous deviation profiles in adulthood.

To analyze the impact of the whether a worse early social environment negatively impacts on CTh and SA heterogeneity after preterm birth, we correlated the respective variable with CTh and SA PC1 scores, respectively. In children, SES was significantly associated with SA PC1 (Spearman rho = 0.269, $p = 0.001$, $p_{FDR} = 0.006$), but not with CTh PC1 (Spearman rho = -0.002, $p = 0.977$, $p_{FDR} = 0.977$; Fig. 8a). In adults, CTh PC1 scores and both SES (Spearman rho = -0.217, $p = 0.033$, $p_{FDR} = 0.088$) and PIRI (Spearman rho = -0.211, $p = 0.044$, $p_{FDR} = 0.088$) were significantly associated (Fig 8b), indicating that a more favorable social and economic environment links to higher CTh deviations across the cortex. However, SA PC1 scores were not (SES: Spearman rho = -0.006, $p = 0.952$, $p_{FDR} = 0.977$; PIRI: Spearman rho = -0.076, $p = 0.474$, $p_{FDR} = 0.711$). As control measures for PCA-based integrated individual CTh deviations across the cortex, we used the number of infranormal regions as well as the deviation scores of mean CTh across the whole cortex (Supplementary Fig. S18). In preterm adults, we observed that a lower SES was correlated with a lower deviation score of mean CTh (Spearman rho = -0.223, $p = 0.029$, $p_{FDR} = 0.051$) and a higher number of infranormal regions in CTh per subject (Spearman rho = 0.200, $p = 0.051$, $p_{FDR} = 0.051$), substantiating the association between early social environment and CTh deviations. Based on previous literature suggesting that a lower SES modifies the relationship between preterm birth and adverse neurodevelopmental outcomes¹⁴⁻¹⁶ in later life, we tested whether a lower SES also adversely modifies the association between preterm birth and adult IBAPs. Moderation analysis revealed that the association between GA and PC1 scores ($r = 0.264$, $p = 0.009$; Fig. 8c) was significantly weakened among individuals with middle ($p_{one-sided} = 0.001$) and low ($p_{one-sided} < 0.001$) SES, indicating that an adverse early social environment modifies IBAPs after preterm birth.

Comment 6:

Results L248 (and Introduction L77): why not characterise the association of IBAPs with outcomes earlier on? As far as I understand both the dHCP and the ABCD provide outcome data

(e.g. Bayley cognitive outcomes at 18-month follow up for dHCP). I think this would be a great addition to the study, and I encourage the authors to include this additional analysis.

Response 6:

We thank the Reviewer for the suggestion to examine the association between IBAPs and cognitive outcomes in the dHCP and ABCD cohorts as well. We have conducted the requested analyses and completely revised the Methods section “Perinatal, social environmental, and behavioral variables”, the Results section “Impacts on cognitive outcome variability”, and Figure 9 (former Figure 5).

Perinatal, social environmental, and behavioral variables

For the dHCP cohort, [...]. Neurocognitive assessments were conducted at the age of about 18 months using the Bayley Scales of Infant and Toddler Development, Third Edition (Bayley-III)^{17,18}. We used the age-normed composite scores for cognitive development (mean = 100, SD = 15)¹⁷.

For the ABCD cohort, [...]. Neurocognitive development was assessed with the NIH Toolbox Cognition Battery at the age of 10 years¹⁹, which is highly ($r = 0.89$) correlated with IQ scores measured with the Wechsler Adult Intelligence Scale, fourth edition (WAIS-IV)²⁰. For this study, the fully corrected T-scores of the Total Score Composite (considering certain demographic characteristics such as gender, education, and race/ethnicity; mean = 50, SD = 10) were used¹⁹.

Impacts on cognitive outcome variability

Finally, to test whether heterogeneity in IBAPs underlies cognitive performance variability among preterm individuals across the lifespan, we associated CTh and SA PC1 scores with measures of cognitive outcomes in the dHCP, ABCD-10, and BLS-26 cohorts. In the BLS-26 cohort, we used the full-scale IQ score. Due to the link of a more favorable social environment with higher CTh PC1 scores, we expected lower PC1 scores to be associated with lower individual IQ in preterm adults.

In the dHCP cohort, neurocognitive testing was performed at 18 months using the Bayley Scales of Infant and Toddler Development, Third Edition (Bayley-III)^{17,18}. There was no significant correlation between IBAPs at term-equivalent age and cognitive abilities at 18 months (CTh PC1: Spearman $\rho = -0.153$, $p = 0.127$, $p_{FDR} = 0.152$; SA PC1: Spearman $\rho = 0.157$, $p = 0.118$, $p_{FDR} = 0.152$; Fig. 9a), presumably because of the age gap between imaging and neurocognitive assessments as previously reported by others⁵. In the ABCD cohort, the NIH Toolbox Cognition Battery composite scores were used to estimate intelligence¹⁹, which has previously been validated to provide a reliable measure of IQ²⁰. Cognition total composite scores were significantly associated with PC1 of SA IBAPs (Spearman $\rho = 0.234$, $p = 0.006$, $p_{FDR} = 0.027$; Fig. 9b) but not with PC1 of CTh IBAPs (Spearman $\rho = -0.053$, $p = 0.484$, $p_{FDR} = 0.484$) in 10-year-olds. Finally, in the BLS-26 cohort, we observed a significant correlation between full-scale IQ and SA PC1 (Spearman $\rho = 0.270$, $p = 0.009$, $p_{FDR} = 0.027$) but not with CTh PC1 (Spearman $\rho = 0.198$, $p = 0.059$, $p_{FDR} = 0.118$; Fig. 9c). The control analyses associating the number of infranormal CTh SA regions with IQ confirmed this relationship in children (Spearman $\rho = -0.232$, $p = 0.006$, $p_{FDR} = 0.041$) and adults (Spearman $\rho = -0.214$, $p = 0.041$, $p_{FDR} = 0.041$; Supplementary Fig. S19). Likewise, SA PC1 scores was significantly linked to IQ in preterm adults (Spearman $\rho = 0.270$, $p = 0.009$; Fig. 5b). This relationship was specific to preterm individuals, with full-term individuals not showing a significant association (CTh: Spearman $\rho = -0.030$, $p = 0.760$; SA: Spearman $\rho = 0.041$, $p = 0.679$). Control analyses of correlations between IQ with the deviation score of mean CTh (Spearman $\rho = 0.194$, $p = 0.063$) as well as with the number of infranormal regions of CTh (Spearman $\rho = -0.227$, $p = 0.030$) and SA (Spearman $\rho = -0.214$, $p = 0.041$) reinforced the relationship between a more severely affected cortex with lower IQ (Supplementary

Fig. S11). In brief, results indicate that individual structural deviations in SA contribute, at least partly, to the variability of general cognitive performance in adulthood after preterm birth.

We want to highlight that for the dHCP cohort, MRI data were acquired at term-equivalent age, while cognitive outcomes were assessed at 18 months. Therefore, the research question that can be addressed with the available data differs compared to the ones investigated for the other two cohorts, where MRI and cognitive assessments were conducted at the same time.

Comment 7

Methods L380: I wonder why the authors didn't use the 3rd (or 4th) data release for the dHCP. Sample size is increased by hundreds from 2nd to 3rd releases (up to 783 subjects available in the latest release).

Response 7:

We thank the Reviewer for the suggestion to use a more recent release from the dHCP project. In response, we have included data from the 3rd release and thereby increased our sample size by nearly 250 additional subjects. We note that this update also involved substantial changes in the processing pipeline and the M-CRIB-S parcellation¹. Since this decision resulted in some differences in the reported results compared to the original manuscript, we decided to stress these changes at the beginning of this letter to be accessible for all reviewers. To ensure transparency of the revised results, we have marked and thoroughly adapted the publicly available analysis code.

Comment 8:

Introduction: I think it could be clearer why going beyond the “group average” paradigm is so important, and what could it bring conceptually and clinically to identify within preterm group developmental patterns – in particular for readers not familiar with preterm literature.

Response 8:

We thank the Reviewer for pointing out that the motivation to go beyond the group average paradigm should be stressed and explained further. We therefore adjusted the Introduction in the following way:

Whereas this model adequately describes the **central tendency of average** brain aberrations of prematurity, it disregards, however, potential individual heterogeneity of altered brain development. Heterogeneity following preterm birth is evident for several outcomes^{21–26}, including postmortem neonatal brain aberrations as well as neurocognitive functioning. In particular, the average IQ after very preterm birth (i.e., before 32 weeks of gestation) is about 11 points lower than that of full-term peers in childhood and adulthood, but individual IQ varies considerably between subjects^{22,27–29}. This neurocognitive heterogeneity is widely assumed to be reflected in related heterogeneity of neurodevelopmental processes^{30–32}. [...] **Moving beyond the group-average paradigm to characterize individualized patterns of neurodevelopment may enable substantial conceptual as well as clinical progress in understanding and ultimately treating the consequences of preterm birth. Conceptually, the heterogeneity of the impact of preterm birth on the brain might be recognized as an essential feature of preterm birth that needs to be considered in our understanding of prematurity**

effects. If so, the diagnosis of such individual brain abnormality patterns might indeed allow for individualized prognosis and treatment of risks for aberrant neurocognitive development.

Comment 9:

Introduction L40-41: could you clarify why accelerated WM maturation could be an issue?

Response 9:

We thank the Reviewer for suggesting to comment on the potential impact of accelerated WM maturation on CTh after preterm birth. Based on neuropathological methods, Marín-Padilla suggested that prematurity-induced damage of cortico-cortical connections may lead to changes in pyramidal neurons with less dendritic and axon branching and length³³. Using both DWI-imaging and histological methods, Dean and colleagues observed less dendritic arborization and synaptic density in the cortex of a preterm large animal model (i.e., sheep), which is detectable by macroscopic MRI³⁴. Finally, using multi-modal MRI in preterm born adults, Rimol and colleagues observed a relationship between the microstructure of tracts and the increased CTh of connected regions, in principle suggesting a prematurity-induced link between aberrant white matter fiber maturation and CTh detectable by in-vivo MRI³⁵. Since we notice that the last finding is about increased CTh instead of decreased ones as we mentioned in the paper, we adapted the sentence as follows:

For example, cortical thickness (CTh) alterations reductions might reflect various microscopic processes affected by prematurity, such as reduced synaptic density, dendritic arborization, and axon length, potentially linked to aberrant accelerated white matter fiber maturation³⁵⁻³⁷.

Comment 10:

Introduction L46: clarify what are “average dysmaturation outcomes”. Perhaps this whole sentence could be clearer.

Response 10:

We agree with the Reviewer to explain the term “average dysmaturation outcomes” in a more clarifying way. Therefore, we adapted the Introduction:

Group average focused brain magnetic resonance imaging (MRI) studies suggest that CTh is persistently altered in preterm-born subjects compared to full-term controls, with widespread CTh increases in infancy, a faster thinning rate in adolescence, and widespread decreases in adulthood^{4,5,38-40}. These findings contribute to a model of widespread altered brain development following preterm birth largely shared between individuals, explaining injury-induced patterns of average dysmaturation outcomes at different stages of development^{7,36,41}. These findings set the basis for a model in which preterm birth induces mean aberrations from typical maturation at different stages of development that are largely shared between individuals. We term these widespread and consistent aberrations suggested by group average MRI studies “average dysmaturation outcomes”.

Comment 11:

Introduction L68: SA is introduced for the first time. Consider introducing it with CTh earlier, and why is it relevant to study in relation to preterm birth?

Response 11:

We thank the Reviewer for the suggestion to introduce SA earlier and to elaborate on the motivation why it should be studied in relation to preterm birth. We added the following to the Introduction:

Group average focused brain magnetic resonance imaging (MRI) studies suggest that CTh is persistently altered in preterm-born subjects compared to full-term controls, with widespread CTh increases in infancy, a faster thinning rate in adolescence, and widespread decreases in adulthood^{4,5,38-40}. Similarly, surface area (SA), thought to reflect the number of radial cortical columns⁴², shows little or no alterations at term-equivalent age after birth^{5,40}, followed by a more pronounced and widespread decrease in childhood and adulthood^{40,43,44}. [...]

To investigate these hypotheses, we focused on CTh and surface area (SA) as paradigmatic outcomes of brain development. Since CTh and SA capture distinct aspects of cortical development⁴⁵, both cortical measures implicated after preterm birth were investigated in parallel.

Comment 12:

Results L92: is a t-test the most appropriate statistical approach here? Don't the authors need to include relevant covariates (e.g. corrected age at scan)? Does the data fulfil t-test assumptions of homoscedasticity / normal distributions? Methods L483-486 suggest linear regression models were used.

Response 12:

We thank the Reviewer for pointing out the mismatch in the description of the statistical method applied to identify average dysmaturation outcomes. Indeed, linear regression models including age and sex as regressors were used for all group comparisons as described in the Methods section. To illustrate an example, in order to compare CTh in the medial temporal gyrus between groups, the following model was used:

$$CTh_{medialtemporal} \sim \beta_1 * group + \beta_2 * age + \beta_3 * sex$$

We have used the term "t-tests" in the Results section since our approach of using linear regression models is statistically equivalent to comparing residuals with a t-test after regressing out covariates. We have revised the Results section to avoid confusion and to ensure consistent language between Methods and Results:

To replicate previous models of average dysmaturation, we used linear regression models correcting for age and sex ~~two-sample t-tests~~ to analyze mean CTh differences between preterm and full-term individuals for each cohort for 34 cortical regions of the Desikan-Killiany parcellation³.

Similarly, the Figure legend of Figure 2a was adapted accordingly:

Cortical thickness (CTh) average dysmaturation outcome for 26-year-old adults after preterm birth estimated by linear regression models correcting for age and sex ~~two-~~

sample t-tests of between-group differences ($p_{FDR} < 0.05$) suggests abnormalities shared between preterm subjects.

Comment 13:

Figure 2d (and Figure S1b_2, etc): what is bankssts? Please include actual ROI names instead of variable names or define in the legend (same for S1c, etc)

Response 13:

We appreciate the Reviewer’s comment to provide the actual ROI names of the Desikan Killiany atlas instead of the shortened variable names as output by FreeSurfer. As some ROI names are quite long (e.g., the full name of “bankssts” is “banks of the superior temporal sulcus”), we have decided to add a Supplementary Table with the full ROI names to maintain readability of axis descriptions:

Table S2: Regions of the Desikan-Killiany atlas³

Region	Abbreviation used
Banks superior temporal sulcus	bankssts
Caudal anterior-cingulate cortex	caudalanteriorcingulate
Caudal middle frontal gyrus	caudalmiddlefrontal
Cuneus cortex	cuneus
Entorhinal cortex	entorhinal
Fusiform gyrus	fusiform
Inferior parietal cortex	inferiorparietal
Inferior temporal gyrus	inferiortemporal
Isthmus– cingulate cortex	isthmuscingulate
Lateral occipital cortex	lateraloccipital
Lateral orbital frontal cortex	lateralorbitofrontal
Lingual gyrus	lingual
Medial orbital frontal cortex	medialorbitofrontal
Middle temporal gyrus	middletemporal
Parahippocampal gyrus	parahippocampal
Paracentral lobule	paracentral
Pars opercularis	parsopercularis
Pars orbitalis	parsorbitalis
Pars triangularis	parstriangularis
Pericalcarine cortex	pericalcarine
Postcentral gyrus	postcentral
Posterior-cingulate cortex	posteriorcingulate
Precentral gyrus	precentral
Precuneus cortex	precuneus
Rostral anterior cingulate cortex	rostralanteriorcingulate
Rostral middle frontal gyrus	rostralmiddlefrontal
Superior frontal gyrus	superiorfrontal
Superior parietal cortex	superiorparietal
Superior temporal gyrus	superiortemporal

Supramarginal gyrus	supramarginal
Frontal pole	frontalpole
Temporal pole	temporalpole
Transverse temporal cortex	transversetemporal
Insular cortex	insula

Comment 14:

Fig 2b / L118: Term controls seem to be far from the median in the model presented. Is this an issue? Figure S2 seems to be missing the fitting of term participants, which is relevant to assess the goodness of fit of the model.

Response 14:

We thank the Reviewer for highlighting these two important points regarding model accuracy. Concerning Figure 2b, we agree that the term-born controls appear to be distributed around a value that is slightly lower than the population median. The observation is likely due to the insular CTh distribution of the term-born female subgroup in the BLS-26 cohort, which appears to be lower than the median of the reference population for their age. The reason for this might be the relatively small size ($n=45$) of the subgroup, which is probably not fully representative of the general population. As a result, their median CTh could deviate slightly from the reference. Nevertheless, when considering both male and female term-born individuals together, fewer than 5% exhibit an extranormal deviation in insular CTh, as shown in Figures 2d, 3b, and 4b. This indicates that the model generalizes well to the BLS cohort overall. Since the decision to show the female insular CTh in the plot was not driven by any specific reason and we want to avoid confusion, we have exchanged the depicted trajectory to the male rostral anterior cingulate CTh one. This is now indicated in the figure description:

Figure 2: Brain development after preterm birth is individually heterogeneous. [...] b, [...] The lifespan trajectory of the rostral anterior cingulate cortical thickness (CTh) in males is shown as an example region, with the insular CTh of male preterm (red) and full-term (black) subjects plotted on top.

Furthermore, we have added the fitting of full-term participants to Supplementary Figure S1 (former S2) to illustrate that also in the control analysis, the model generalized well to unseen data.

Comment 15:

Results L130: “The overall prevalence of at least one extranormal deviation varied with age from only less than 65 % in preterm neonates to over 85 % of preterm children and adults, suggesting developmental and/or environmental influences on brain structure.” -> I am not sure that this last sentence is adding much

Response 15:

We agree with the Reviewer that our interpretation is not very specific and useful for further interpretation. Therefore, we have adapted the sentence as follows:

The overall prevalence of at least one extranormal deviation varied with age from only less than 65 % in preterm neonates to over 85 % of preterm children and adults, suggesting developmental and/or environmental influences on brain structure.

Comment 16:

Results L132: “27 % of preterm individuals sharing deviations in any given region”. A bit unclear if this is the same result presented in L115.

Response 16:

We appreciate the Reviewer’s careful investigation of the manuscript. The Reviewer is right that the Results presented show the same value (i.e., up to 27 % of preterm individuals share an infranormal deviation in any given region). However, former line 115 presents findings in the BLS cohort at age 26 years (i.e., the BLS-26 cohort), whereas former line 132 refers to the results at age 38 years (i.e., the BLS-38 cohort). They are very similar as extranormal deviations seem to be stable over time (see Supplementary Figure S15 and Supplementary Table S5d). To avoid confusion, we have added the cohort name additionally:

*No more than 27 % of preterm adults **of the BLS-26 cohort** shared extranormal deviations in the same cortical region (Fig. 2d).*

[...]

*Location patterns remained heterogeneous across all developmental stages, with no more than 27 % of preterm individuals sharing deviations in any given region **in the BLS-38 cohort as well**, [...].*

Comment 17:

Results L160: “indicates that earlier birth increases the likelihood of structural alterations” -> as far as I understand the authors are not looking at likelihood of alterations but rather to what extent they share locations.

Response 17:

We thank the Reviewer for pointing out the misleading interpretation of findings here. We have rephrased the Results section in the following way:

*This indicates that earlier birth increases the **likelihood percentage of individuals that share extranormal of** structural alterations **in overlapping regions**.*

Comment 18:

Results L170: “Results demonstrate that GA has a substantial impact on the extent” -> I would be careful with the causal inference here. Isn’t it possible that other factors such as BW and perinatal clinical complications are also related to the extent of atypical development?

Response 18:

We thank the Reviewer for this important and thoughtful comment. We modified the Results section to address the Reviewer’s concerns:

*These results demonstrate that GA **might contribute substantially to has a substantial impact on** the extent of CTh deviations, with earlier birth leading to more widespread patterns of deviations, also later in life.*

Comment 19:

Results L253: Is it justified to present one-sided p-values? Please clarify whether p-values are corrected for multiple comparisons.

Response 19:

We appreciate the Reviewer's criticism to use one-sided p-values when assessing significance of the correlation between IBAPs and cognitive outcome measures. We agree that our rationale for using one-sided tests was not sufficiently explained in the initial manuscript. Initially, we have used one-sided p-values as we hypothesized a clear directionality of the association (i.e., that lower PC1 reflecting more infranormal deviations would be associated with poorer cognitive performance) in our main analysis. However, we agree that despite this directional expectation, the opposite direction is possible. To provide a more conservative and transparent assessment, we now present two-sided p-values in the revised manuscript. Additionally, we explicitly report both uncorrected and FDR-corrected p-values to account for multiple comparisons. As a consequent of this stricter approach and the inclusion of the ABCD cohort in this analysis framework as a response to previous comments (i.e., resulting in a higher number of statistical tests), the previously significant association between CTh PC1 and full-scale IQ is no longer significant. We have updated the relevant section of the Results ("Impacts of cognitive outcome variability", see response to comment 6) and Figure 9 (former Figure 5) accordingly.

Comment 20:

Discussion L268: is it really "influenced" by – or mediated / attenuated?

Response 20:

We appreciate the Reviewer's suggestion to adapt the phrasing to summarize the effect of the social environment on IBAPs. Therefore, the Discussion was changed in the following manner:

*This study provides evidence for lasting and heterogeneous brain aberrations after preterm birth, shaped by consistent effects of variable initial injury-induced dysmaturation at macroscopic (i.e., extent and location) and microscopic (i.e., cellular) scales, and **mediated influenced** by the early social environment.*

Comment 21:

Discussion L285: "suggesting individual developmental trajectories of the brain after preterm birth." -> Clarify these are trajectories of brain structural features

Response 21:

We thank the Reviewer for the recommendation to clarify the Discussion further. We agree with the Reviewer and have changed the formulation:

*This heterogeneity persisted across all examined cohorts, extended beyond CTh to other structural measures, and was robust across different normative reference models (Supplementary Fig. S1), suggesting individual developmental trajectories of **the brain structural features** after preterm birth.*

Comment 22:

Methods (overall): Perhaps improve consistency of “preterm” / “very preterm”, slightly confusing which group is included. E.g. L416; “Since this estimation is rather vague, we only included subjects born more than 7 weeks premature (i.e., gestational age ≤ 32 weeks) as preterm”. $GA \leq 32$ is very preterm, but in introduction it is suggested that the study focus on preterm birth (including late-onset), e.g. L29: “preterm birth is defined as birth before 37 weeks of gestation”. Indeed most of preterm neonates part of the dHCP are late preterm ($32 < GA < 37$).

Response 22:

We thank the Reviewer for pointing out inconsistencies in the definition of preterm and very preterm birth. Therefore, we have more precisely described inclusion criteria for each cohort in the Methods section:

Neonatal brain measure estimation from dHCP data

[...] Fetal age was estimated from the mother’s last menstrual period and confirmed by ultrasound where possible. Gestational age at birth ranged from 24.29 to 36.86 weeks, i.e., the cohort includes preterm as well as very preterm born neonates.

Child brain measure estimation from ABCD data

Gestational age was derived from a parent questionnaire administered approximately 10 years after childbirth, asking “About how many weeks premature was the child when they were born?” (included in file `abcd_devhxxss01.txt` of the 4th data release). Given the retrospective and potentially imprecise nature of this estimate, we applied a conservative approach to define our cohorts. We determined gestational age as 40 weeks minus the reported number of weeks premature the baby was said to be premature and included only those participants whose estimated GA was ≤ 32 weeks in the “preterm” group. This threshold corresponds to very preterm birth and was chosen to minimize misclassification due to vague recall of GA at birth. Therefore, subjects born between gestational weeks 32 and 37 were excluded. Full-term birth was defined as $GA \geq 37$ weeks. Since this estimation is rather vague, we only included subjects born more than 7 weeks premature (i.e., gestational age ≤ 32 weeks) as preterm and born at or after week 37 as full-term.

Adult brain measure estimation from BLS data

The Bavarian Longitudinal Study (BLS) is a geographically defined, whole-population sample of individuals born very preterm (VP; < 32 weeks of gestation) and/or with very low birth weight (VLBW; $< 1,500$ g) and full-term (FT; > 37 weeks of gestation) controls that were followed from birth between January 1985 and March 1986 until adulthood^{46,47}. Therefore, the cohort included subjects born very preterm but also subjects born later in the preterm period that had VLBW.

To avoid confusion and to simplify readability, we will refer to all subjects born before the 37th week of gestation as “preterm” in the manuscript. This clarification was added at the beginning of the Results section:

Subjects born at different stages of the preterm period, namely extremely preterm ($GA \leq 28$ weeks), very preterm ($28 < GA \leq 32$ weeks), and moderate to late preterm ($32 < GA < 37$ weeks) were included in this study and are collectively referred to as “preterm” throughout this work (see Methods for the detailed inclusion criteria for each cohort).

Furthermore, we have highlighted the inclusion of several stages of prematurity as a limitation of our study:

Clearly, the present work should be considered alongside some important methodological limitations. First, the study combined data from three cohorts with differing inclusion criteria for preterm birth, resulting in the investigation of individuals ranging from extreme preterm (GA \leq 28 weeks) to late preterm (32 < GA < 37 weeks). Importantly, the consequences of preterm birth are influenced by the degree of prematurity, and findings from very preterm individuals may not generalize to those born moderately or late preterm.

Comment 23:

Methods L481: I am not sure I understand the justification for not harmonising the dHCP and BLS-38. Difference in size, sequences (T1 vs T2?), and preprocessing pipelines seems enough justification for site harmonisation.

Response 23:

We thank the Reviewer for this important comment regarding MRI data harmonization. In our study, several key factors guided our harmonization strategy.

Importantly, variability in multisite data is two-fold: first, **between-dataset variability** (i.e., effects of “study”) refers to the variability between different studies (e.g., between ABCD and BLS). This includes technical aspects (e.g., scanner vendor, T1-weighted vs. T2-weighted, sequence parameters, preprocessing pipelines etc.) and aspects of sample characteristics (e.g., inclusion/exclusion criteria, sample size etc.)⁴⁸. Second, **within-dataset variability** (effects of “(scanning) site”) refers to variability within one study that arises from the use of different hardware or data being acquired at multiple sites for the same study, while following the same general framework (i.e., inclusion/exclusion, sequence parameters, processing pipelines).

Between-dataset variability is inherently accounted for in our GAMLSS-based normative modeling framework by including random effects of study (see Fig. S5.2.3. and Supplementary Analyses of the original publication by Bethlehem and colleagues⁸). However, within-dataset variability is beyond the resolution of our normative modeling framework due to two reasons: (i) “site” was not included as a further covariate in the original models, and (ii) GAMLSS-based between-site or between-study harmonization may not be applicable for studies with small ($n < 100$) numbers of healthy control participants per site⁸.

Study	Dataset	Number of different scanner hardware and/or sites for data acquisition
dHCP	dHCP	1
ABCD	ABCD-10	21
	ABCD-12	21 (the same as used for ABCD-10)
BLS	BLS-26	4
	BLS-38	1 (not the same as used for BLS-26)

For the dHCP study, the same scanner at the same site was used to acquire all images. Therefore, no within-dataset harmonization was required.

For the ABCD study, different scanners were used at different sites were used, which requires within-dataset harmonization. The use of longCombat was possible since scanners and sites were equivalent in longitudinal acquisition.

For the BLS study, at the first MRI acquisition (i.e., BLS-26), two sites (i.e., Munich and Bonn) acquired MRI data. Furthermore, the scanner hardware was changed during the acquisition

period at both sites (from Philips Achieva 3 T TX systems to Philips Ingenia 3 T systems), resulting in four different acquisition combinations (i.e., Bonn-Achieva, Bonn-Ingenia, Munich-Achieva, Munich-Ingenia). For the longitudinal data acquisition (i.e., BLS-38), only Munich acquires data at one scanner that has not been used at timepoint 1 (i.e., Philips Ingenia Elition X 3 T). Therefore, BLS-26 and BLS-38 had to be treated as two different studies in normative modeling, which resulted in the calculation of random effects for study. This is not optimal but the only way to account for variability with the present available information.

Comment 24:

Methods L510: the choice of 30 weeks could be better justified. Usually 28 weeks (extreme preterm) or 32 weeks (very preterm) are clinically used as thresholds.

Response 24:

We thank the Reviewer for this insightful comment and agree that gestational age thresholds of 28 weeks (extreme preterm) and 32 weeks (very preterm) are more commonly used in clinical practice. The cutoff at 30 weeks GA was chosen to balance the number of participants in the earlier preterm (GA of ≤ 30) and later preterm (GA of > 30) groups and to thereby reducing the likelihood that the percentage of subjects in the groups with extranormal deviations are driven by outliers. Using a cutoff of 30 weeks in the original analysis led to the following group sizes:

Dataset	earlier preterm (GA ≤ 30 weeks)	later preterm (GA > 30 weeks)
BLS-26	52	43
dHCP	27	65

To evaluate the robustness of our findings, we conducted an additional control analysis using standard clinical thresholds, dividing preterm subjects into the following three groups: (i) extreme preterm (GA ≤ 28 weeks), (ii) very preterm ($28 < GA \leq 32$ weeks), and (iii) late preterm (GA > 32 weeks). Results are included in the Supplementary Material. Briefly, for the dHCP cohort, the group sizes were $n = 14, 30,$ and $48,$ with up to 57 %, 20 % and 19 % of participants respectively sharing a supranormal deviation in any given region. For the BLS-26 cohort, the group sizes were $n = 14, 67,$ and $15,$ with corresponding maxima of 57%, 27%, and 20% of subjects showing shared infranormal deviations.

These results are consistent with our main analysis and support the conclusion that the likelihood of shared extranormal deviations increases with the severity of prematurity.

In the Results section, the Supplementary analysis was referred to as follows:

Repeating this analysis with partitioning subjects into groups more commonly used in clinical practice, i.e., extreme preterm (GA of ≤ 28 weeks), very preterm ($28 < GA \leq 32$ weeks), and late preterm (GA > 32 weeks) resulted in similar results (Supplementary Fig. S11) but led to less balanced group sizes. This in general, these results indicate that earlier birth increases the percentage of individuals that share extranormal structural alterations in overlapping regions.

Comment 25: Remarks on code availability

Code available seems to be complete and well documented.

Response 25:

We thank the Reviewer for appreciating our efforts in sharing and documenting the analysis code.

Responses to Reviewer #2

Comment 1:

Response 1:

We thank the Reviewer for the effort to thoroughly co-review this manuscript and take part in this important initiative.

Responses to Reviewer #3

Comment 1:

This work presents a comprehensive investigation into the effects of preterm birth on individual deviations in brain structure from a normative growth pattern. By leveraging three brain MR scan datasets covering preterm neonates, children, and adults, the authors reveal significant variability in deviation maps between individuals, yet demonstrate relatively consistent deviations within individuals over time, particularly in cortical thickness and surface area. They also explore the neurobiological basis and cognitive relevance of preterm-related individual brain deviations by incorporating a brain cellular atlas, social environmental factors, and cognitive assessments. The topic is highly engaging, and the authors offer insights into understanding the atypical developmental patterns of preterm infants, which may facilitate the identification of stable, individualized neuroimaging biomarkers for cognitive and behavioral outcomes related to preterm birth. This work employs diverse brain MRI datasets, strict scan harmonization approaches, and is presented with clear and effective writing, reflecting a high level of quality. However, I have some suggestions that may help further improve the manuscript before publication.

Response 1:

We thank the Reviewer for the positive evaluation of the paper and the constructive feedback.

Comment 2:

Some of the results are initially presented in a highly descriptive manner, relying heavily on percentage-based summaries, which limits the depth of the findings. More rigorous quantitative analyses should be emphasized. For example, the authors note that in the BLS-26 cohort, a maximum of 27% of adults share the same extreme abnormal region. This could be better captured using Spearman correlations by providing the average Spearman coefficient for each brain region across the population. While they do include a correlation matrix of overall coefficients in Figure 2, it does not show the similarity of profiles in each brain region.

Similar percentage-based analyses are used for metrics like WMV and GMV, making cross-metric comparisons appear superficial.

In the results on finding constant deviations within the same individual, similar percentage-based analyses are used again. Some measurements, such as ICC, could provide more quantitative findings.

Additionally, when replicating individual deviation patterns with another normative model (Supplementary Material S2), they did not report the correlation coefficients between these patterns and the main text's results.

However, I am not suggesting that all of their results are overly descriptive. Many quantitative analyses are very rigorous and interesting, such as the correlations between the number of abnormal regions and GA, and the correlations between the deviation-cell-type association and GA.

Response 2:

We appreciate the Reviewer's suggestion to also provide quantitative analyses to better illustrate the results on brain development heterogeneity after preterm birth.

First, since percentage-based summaries are commonly used in the normative modeling literature to illustrate the prevalence of deviations within a population^{5,49-51}, we decided to stick to this percentage-based presentation for our main results. However, we agree that complementary, more quantitative analyses may help to grasp the concept of interregional heterogeneity. Therefore, we have followed the Reviewer's recommendation to provide the

average Spearman correlation coefficient for each brain region across the preterm population, which captures the degree of similarity in deviation patterns across individuals. We have added Supplementary Figures S8 and S9, showing (i) the correlation between each possible region pair across subjects, and (ii) the distribution of the average correlation of one region with all others. This control analysis again suggests that there are no spatial patterns of deviation scores across individuals, neither for CTh nor for SA. Moreover, we have described the procedure and results in the Supplement:

Heterogeneity of regions across preterm subjects

To examine the similarity of inter-regional deviation patterns of CTh and SA across individuals, we computed the average Spearman correlation coefficient for each brain region across the preterm population in each cohort. Specifically, we cross-correlated binarized IBAPs across subjects (see Methods for a detailed description on binarization). The resulting average correlation coefficient per brain region indicate that patterns of extranormal deviations were generally not shared among individuals, with an average Spearman correlation of $r < 0.3$ for most region pairs, both for CTh and SA (Supplementary Fig. S8 and S9). An exception can be observed in the dHCP cohort, where average coefficients reach up to $r = 0.75$. This likely reflects that several regions do not show any extranormal deviations, which may increase the influence of a few deviating regions within this cohort. In general, this control analysis supports our main finding that there is no to little consistent inter-regional pattern of extranormal deviations across preterm individuals.

For measures of cerebral tissue volumes like WMV, we stuck to the percentage-based presentation of results more common in normative modeling literature.

Second, regarding the results on IBAP location consistency within the same individual, we have also followed the Reviewer's helpful suggestion and have included the intraclass correlation coefficient (ICC) as a quantitative measure. We have added the following to the Results section "Constant IBAP locations in individual development" as well as to the Methods section. Furthermore, complete results are listed in Supplementary Tables S5.

To quantitatively evaluate whether IBAPs of preterm individuals remain constant over time, we computed the intraclass correlation coefficient (ICC) for each brain region. For all regions except the frontal and temporal poles, which are known to be less reliably estimated by FreeSurfer^{3,9}, ICC was higher than 0.837 for children and higher than 0.803 for adults (Supplementary Tables S5a-b), indicating longitudinal stability of extranormal deviations.

[...]

We compared whether extranormal deviations remained extranormal over time. To quantify the consistency of brain region deviation scores across timepoints, we calculated the Intraclass Correlation Coefficient (ICC) for each brain region using Python's pingouin package. Specifically, the ICC(3,1) model was used, which is appropriate for repeated measures where the timepoints are treated as fixed effects and subjects as random effects.

Third, regarding the supplementary analysis using another population reference normative model by Rutherford and colleagues⁵², we agree that correlating the deviation score estimates from the two frameworks provides a meaningful comparison. To illustrate this, we added Supplementary Figures S3 and S4, along with Supplementary Tables S3a and S3b, which quantitatively compare the results from the two frameworks. Furthermore, we described the procedure and results in the Supplementary methods and results section:

Next, to compare deviation score estimates between the two different frameworks, “Rutherford-framework” Z-scores of each Desikan-Killiany ROI were averaged across hemispheres and transformed into centile scores using the cumulative distribution function of the standard normal distribution. We computed the Spearman correlation coefficient between the “Bethlehem-framework” centile score estimate and the “Rutherford-framework” centile score estimate for each ROI across full-term and preterm subjects of the BLS-26 cohort (Supplementary Fig. S3 and S4). Centile score estimates from the two frameworks showed strong correlation across ROIs for both CTh and SA, with Spearman correlation coefficients ranging from 0.660 to 0.886 for SA and from 0.881 to 0.995 for CTh (see Supplementary Tables S3). Notably, the largest differences between frameworks occurred within the normal centile range, while most individuals with extranormal scores were consistently identified as such in both frameworks, suggesting that the presented deviation scores are largely independent on the population reference normative model used for their estimation.

Comment 3:

The authors are mainly concerned with binary brain areas and the number of deviations, but the intensity of deviations (i.e., weighted deviation profiles) is also important and should not be overlooked.

Response 3:

We appreciate the Reviewer’s recommendation to also consider the intensity of deviations, not just their presence or absence. Again, we have followed previous normative modeling studies when using the classification into infra- and supranormal deviations, without taking into account the intensity deviations further population^{5,49-51}. In Supplementary Fig. S6 and S7, we show unthresholded deviations scores and their correlation across preterm subjects for CTh and SA, respectively. This provides a more nuanced view of the similarity between individual deviation profiles.

Comment 4:

More strict FDR corrections may be needed. The authors control for family-wise error using FDR correction in the group difference analysis. However, all subsequent analyses for the deviation maps should also consider such corrections.

Response 4:

We appreciate the Reviewer’s suggestion to conduct more strict FDR corrections. For the analyses investigating the extent and location consistency of IBAPs as well as the association with early social environment and cognitive outcomes, we have applied FDR corrections for each experiment and reported both uncorrected and corrected two-sided p-values in the manuscript. Since the analysis testing the association between cell type-specific gene expression with IBAPs was exploratory, we did not correct for multiple hypothesis testing.

Comment 5:

I believe the results of all three cohorts are of equal importance. Some figures in the supplementary material could be condensed and placed in the main text (e.g., Figures S1, S2, S5, S6-7).

Response 5:

We thank the Reviewer for the recommendation to visualize results from all cohorts in the main text. We have thoroughly updated the figure story accordingly. Due to space restrictions, we have included only a subset of the suggested figures to the main text (i.e., former Figures S1 and S3, now Figures 3 and 4).

Comment 6:

Figure 1 is too large and complex. Much of the information is relatively repetitive, such as the individual deviation maps and some result plots.

Response 6:

We appreciate the Reviewer's helpful suggestion to simplify Figure 1. In response, we have revised the figure to reduce redundancy where possible, for example by removing individual deviation maps and some results plots. However, after careful discussion with the co-authors, the feedback was that the figure helps to grasp the workflow and overall structure and narrative of the study. Therefore, we decided to retain its general layout while refining specific elements to improve clarity and reduce complexity.

Comment 7:

There are some typos in the manuscript. For example, in Line 72, what does "see 1" mean? In Line 87, where is (ii)?

Response 7:

We thank the Reviewer for pointing out that the manuscript contains typos. We have thoroughly revised the manuscript for typos.

Responses to Reviewer #4

Comment 1:

The paper by Thalhammer and co author, titled “Heterogeneous, temporally consistent, and plastic brain development after preterm birth” and submitted to the journal Nature Communications, presents study on brain morphometry following preterm birth at three distinct time points: term equivalent age, school-age, and adulthood. While there are an impressive amount of analyses, the paper falls short. I have several major comments listed below.

Response 1:

We thank the Reviewer for the thorough revision of our work.

Comment 2:

Stylistically the first paragraph contains multiple thoughts, compressed into a hard to follow paragraph. I would suggest breaking up this paragraph into several simpler paragraphs, each with a strong topic sentences, to guide the reader. This extends throughout the paper. It is hard to read and follow.

Response 2:

We thank the Reviewer for this valuable feedback regarding the clarity and structure of the manuscript. We have rephrased the Introduction as well as other dense sections throughout the paper to improve readability by breaking them into shorter, more focused paragraphs. With this, we hope to have sufficiently addressed the Reviewer’s remarks. For example, we have cited the rephrased introduction below:

*In humans, preterm birth is defined as birth before 37 weeks of gestation¹. With a worldwide prevalence of about 11 %², it is the leading cause of perinatal mortality and long-term motor, cognitive, and behavioral impairments³⁻⁵. Impairments are typically more severe with decreased gestational age (GA) and lower quality of postnatal care^{6,7}. Preterm birth impacts brain development through initial brain injuries that result in dysmaturation patterns affecting multiple tissue types and brain regions^{8,9}. Initial injuries like hypoxic-ischemic events damage vulnerable cell populations, mainly the pre-oligodendrocyte (OL) cell line and the subplate, with subsequent local inflammation mediated by reactive astrocytes and activated microglia⁹⁻¹⁵. Resulting **brain aberrations after dysmaturation patterns of** preterm birth have been shown to affect most parts of the brain, i.e., widespread gray matter areas, including cortical regions as well as basal ganglia, amygdala, thalamus, and hypothalamus nuclei¹⁶⁻²⁴, and white matter areas²⁵⁻³⁰, not only in infants and children, but also in adolescents and adults. For example, cortical thickness (CTh) **reductions alterations** might reflect various microscopic processes affected by prematurity, such as **reduced synaptic density, dendritic arborization, and axon length, potentially linked to aberrant accelerated** white matter maturation^{28,31,32}. Group average focused brain magnetic resonance imaging (MRI) studies suggest that CTh is persistently altered in preterm-born subjects compared to full-term controls, with widespread CTh increases in infancy, a faster thinning rate in adolescence, and widespread decreases in adulthood^{30,33-36}. **Similarly, surface area (SA), thought to reflect the number of radial cortical columns³⁷, shows little or no alterations at term-equivalent age after birth^{30,36}, followed by a more pronounced and widespread decrease in childhood and adulthood^{36,38,39}. These findings set the basis for a model in which preterm birth induces mean aberrations from typical maturation at different stages of development that are largely shared between individuals^{8,40,41}. We term these***

widespread and consistent aberrations suggested by group average MRI studies “average dysmaturation outcomes”.

Whereas this model adequately describes the central tendency of average brain aberrations of prematurity, it disregards, however, potential individual heterogeneity of altered brain development. Heterogeneity following preterm birth is evident for several outcomes^{5,42-46}, including postmortem neonatal brain aberrations as well as neurocognitive functioning. In particular, the average IQ after very preterm birth (i.e., before 32 weeks of gestation) is about 11 points lower than that of full-term peers in childhood and adulthood, but individual IQ varies considerably between subjects^{42,47-49}. Furthermore, preterm birth is caused by a wide range of events, from spontaneous to induced, with various underlying pathogenic processes, each potentially associated with heterogeneous developmental trajectories. This neurocognitive and etiological heterogeneity is widely assumed to be reflected in related heterogeneity of neurodevelopmental processes^{8,30,50,51}. In preterm neonates, Dimitrova and colleagues have already presented tremendous heterogeneity of cortical outcomes with little spatial overlap³⁰. Moving beyond the group-average paradigm to characterize individualized patterns of neurodevelopment may enable substantial conceptual as well as clinical progress in understanding and ultimately treating the consequences of preterm birth. Conceptually, the heterogeneity of the impact of preterm birth on the brain might be recognized as an essential feature of preterm birth that needs to be considered in our understanding of prematurity effects. If so, the diagnosis of such individual brain abnormality patterns might indeed allow for individualized prognosis and treatment of risks for aberrant neurocognitive development.

Thus, considering vast heterogeneity in neurocognitive-behavioral outcomes alongside persistent and widespread average brain aberrations, we hypothesized substantial spatial heterogeneity of individual brain abnormality patterns (IBAPs) after preterm birth across development. Despite suspected heterogeneity, we expected consistency in specific aspects of IBAPs, particularly concerning features related to injury-induced dysmaturation and developmental plasticity. Regarding injury-induced dysmaturation, we hypothesized a consistent impact of initial injury on IBAPs in three domains: extent, anatomical location, and cellular underpinnings. Specifically, we hypothesized that (i) IBAP extent depends on GA, with earlier birth associated with larger IBAPs, (ii) anatomical locations of IBAPs remain temporally constant along individual development, and (iii) cellular underpinnings of IBAPs involve glial cells. Regarding developmental plasticity, we additionally expected IBAPs to be modified by the child's early social environment, supported by previous findings linking social environment to neurocognitive outcomes after preterm birth⁵²⁻⁵⁴. Finally, inspired by neurocognitive heterogeneity, we proposed that IBAP heterogeneity would underpin the cognitive performance variability among preterm born individuals.

To investigate these hypotheses, we focused on CTh and SA as paradigmatic outcomes of brain development. Since CTh and SA capture distinct aspects of cortical development⁵⁵, both cortical measures implicated after preterm birth were investigated in parallel. First, we employed normative modeling to describe regional CTh and SA developmental trajectories⁵⁶ and to assess spatial heterogeneity of IBAPs across three cohorts, encompassing preterm and full-term neonates, children aged 10 and 12 years, and adults aged 26 and 38 years (for an overview of the study hypotheses, data, and methods, see Fig. 1). Second, we tested the temporal consistency of initial injury using longitudinal IBAPs of preterm children and adults and integrated cell density maps derived from the Allen Human Brain Atlas (AHBA) to trace cellular underpinnings of adult IBAPs back to preterm birth. Third, we examined plasticity of IBAPs in response to the environment by linking adult IBAPs with social environmental features of early development. Finally, we tested whether adult IBAPs were associated with IQ after preterm birth. Taken together, we propose a temporally consistent effect of variable

initial injury-induced dysmaturation at the macroscopic (i.e., extent and location) and microscopic (i.e., cellular) scale, which is, however, sensitive to early social-environmental influences, leading to individually heterogeneous abnormality patterns after preterm birth.

Comment 3:

It would be helpful to better compare this work to Dimitrova R, et al 2021. The approaches are very similar—characterize individual deviations from a normative growth chart and move away from population averages. The conclusions from that paper are nearly identical to this paper, but are largely ignored in the current manuscript, which makes the current work appear much more novel.

Response 3:

We thank the reviewer for drawing attention to the study by Dimitrova et al, which indeed can be considered an important milestone in shifting the focus of neurodevelopment after preterm birth towards individualized investigation. Therefore, we have cited the work by Dimitrova and colleagues in the original manuscript to highlight the results as well as to acknowledge their work. We have adapted the Introduction to particularly stress the study:

*This neurocognitive **and etiological** heterogeneity is widely assumed to be reflected in related heterogeneity of neurodevelopmental processes^{8,30,50,51}. **In preterm neonates, Dimitrova and colleagues have already presented tremendous heterogeneity of cortical outcomes with little spatial overlap⁵.***

Although we agree that the approaches are very similar, we respectfully disagree that the conclusions of our manuscript are nearly identical to those of Dimitrova et al. While both studies share a normative modeling framework to characterize individual-level deviations, Dimitrova et al. focused exclusively on the neonatal period and primarily demonstrated heterogeneity in structural brain development at term-equivalent age. In contrast, our study extends this approach across the life span and demonstrates for the first time that individual variability in brain structure among preterm-born individuals persists well beyond the neonatal period. Importantly, we also examined potential biological and environmental contributors to this heterogeneity, such as cellular underpinnings and socio-demographic factors, which were not considered in the study by Dimitrova et al.

In summary, while methodologically similar in terms of applying normative modelling to preterm born cohorts, our study provides further novel insights into the long-term developmental consequences as well as underpinnings of structural heterogeneity, which we believe is a significant extension to the field beyond the neonatal focus of previous work.

To better acknowledge the relevance of Dimitrova et al., we have revised the manuscript to more explicitly compare our work to theirs:

*Consistent with our results, previous normative modeling studies have also presented substantial heterogeneity of brain measures in preterm neonates, including CTh and SA^{5,31,32}. **The work by Dimitrova and colleagues suggests that no more than 10% of preterm neonates share extranormal deviations in several structural brain features, which is similar to our findings in the same cohort of a maximum of about 20% spatial overlap of deviations. Small discrepancies might stem from methodological differences, especially different normative models and cortical parcellations used. Although overlapping for neonates, our study substantially advances the field by additionally investigating lifelong heterogeneity as well as cellular underpinnings. but without investigating possible manifestations in childhood or adulthood. Our results fill this critical gap by presenting heterogeneous structural***

aberrations in preterm neonates, children, and adults; Thereby, we fundamentally extend our the understanding of brain development after preterm birth.

Comment 4:

Many would disagree with this statement: “This neurocognitive heterogeneity is widely assumed to be reflected in related heterogeneity of neurodevelopmental processes”. Preterm birth is known to be heterogeneous disorder due to the many different causes of it, where factors range from maternal to fetal. For example, induced preterm birth is a treatment for high risk of still birth. This may have very different brain based outcomes than spontaneous preterm birth due to an incompetent cervix. It may be worth discussing how the many difference causes of preterm birth may account for the observed results.

Romero R, Dey SK, Fisher SJ. Preterm labor: one syndrome, many causes. *Science*. 2014 Aug 15;345(6198):760-5. doi: 10.1126/science.1251816. Epub 2014 Aug 14. PMID: 25124429; PMCID: PMC4191866.

Response 4:

We thank the Reviewer for raising this important point. We fully agree that various causes of preterm birth exist - from spontaneous to medically induced, from maternal to fetal – each potentially associated with heterogeneous developmental trajectories, as emphasized by Romero and colleagues⁵³. We also agree that these heterogeneous causes likely contribute to diverse neurodevelopmental trajectories.

However, although a few recent neuroimaging studies have started to investigate interindividual heterogeneity in brain development among preterm neonates^{5,31,32,54}, the vast majority of the neuroimaging literature still focuses on group average effects (i.e., average dysmaturation after preterm birth). Most studies not even differentiate between spontaneous and induced preterm birth – often because these data are not even available in datasets collected for neuroimaging studies. So as described in our introduction, heterogeneity in brain structure is suggested by the heterogeneity of causes and consequences of preterm birth. Yet, no study currently exists that has investigated this suspected heterogeneity in brain structure across the lifespan. Therefore, this manuscript addresses this gap to raise awareness, especially in the neuroimaging community, that preterm birth is indeed a heterogeneous disorder, also when it comes to brain structure.

We have revised the manuscript to clarify this point further:

Whereas this model adequately describes the central tendency of brain aberrations of prematurity, it disregards, however, potential individual heterogeneity of altered brain development. Heterogeneity following preterm birth is evident for several outcomes^{21–26}, including postmortem neonatal brain aberrations as well as neurocognitive functioning. In particular, the average IQ after very preterm birth (i.e., before 32 weeks of gestation) is about 11 points lower than that of full-term peers in childhood and adulthood, but individual IQ varies considerably between subjects^{22,27–29}. Furthermore, preterm birth is caused by a wide range of events, from spontaneous to induced, with various underlying pathogenic processes, each potentially associated with heterogeneous developmental trajectories. This neurocognitive and etiological heterogeneity is widely assumed to be reflected in related heterogeneity of neurodevelopmental processes^{30–32}.

Comment 5:

Were the dHCP and ABCD data used in creating the reference brain charts? If so, how does this data leakage affect the results?

Response 5:

We thank the Reviewer for raising this important point regarding the validity of using overlapping datasets for both model training and application. We start with providing more details on training of the original model by Bethlehem et al. followed by clarifying how it was adapted in our current study.

The normative modeling framework by Bethlehem et al. was trained on a large, population-level reference dataset of more than 100,000 subjects that also included individuals from the dHCP and ABCD studies. Importantly, for original model training, participants were selected based on general health criteria (e.g., to never been diagnosed with a mental health disorder according to the parental response to the ABCD screening and risk questionnaire) and were **not stratified by gestational age**, i.e., both term- and preterm-born individuals were included to the reference cohort as “cognitively normal”. Furthermore, only chronological age (not gestational or postnatal age) was included as a covariate into the model, consistent with the authors' goal of creating developmental reference charts across the lifespan, rather than focusing specifically on prematurity.

In contrast, we aimed to assess how individuals born preterm may deviate from typical developmental trajectories. To do so, we adapted the pretrained model from Bethlehem et al. by refitting it to the dHCP and ABCD data, but with some key differences: we treated each dataset as a new site (i.e., “dHCP_new”, “ABCD_new”) and included only term-born individuals when estimating the site-specific random effects. This step ensures that any potential differences due to imaging site or preprocessing pipeline are accounted for **without including preterm data into the model fitting**. By this means, the application data of interest in our study (i.e., preterm individuals) are not part of the refitting process.

Since Bethlehem et al. demonstrated in their original work that leave-one-study-out models yielded nearly identical model parameters compared to the full model⁸, we argue that the overall model shape is only minimally influenced by inclusion or exclusion of individual cohorts and that refitting of the original model to term dHCP or ABCD data as well as its adaptation to preterm data is not influenced by the fact that dHCP and ABCD were used in the original training process.

To clarify that, we have included the following description in the Methods section:

Normative reference charts for bilateral regional CTh, SA, as well as for global measures (i.e., GMV, sGMV, and WMV) were obtained from the BrainChart project⁸, in which GAMLSS-based normative models were computed over the human lifespan as a function of age and sex from about 100,000 MRI scans. The original model was trained on a large, population-level reference dataset of more than 100,000 subjects that also included individuals from the dHCP and ABCD studies. Importantly, for original model training, participants were selected based on general health criteria and were not stratified by gestational age, i.e., both term- and preterm-born individuals were included to the reference cohort as “cognitively normal”. Furthermore, only chronological age (not gestational or postnatal age) was included as a covariate into the model.

To adapt the reference charts to the datasets used for this study, random effects of study were calculated only based on full-term individuals and by treating dHCP and ABCD as a new site (i.e., “dHCP_new”, “ABCD_new”). This step ensures that any potential differences due to imaging site or preprocessing pipeline are accounted for without including preterm data into the model fitting. By this means, the application data of interest in the present study (i.e., preterm individuals) are not part of the refitting process. Since Bethlehem et al. demonstrated in their original work that leave-one-study-out models yielded nearly identical model parameters compared to the full model⁸, we argue that the overall model shape is only minimally influenced by inclusion or exclusion of individual cohorts and that refitting of the original model to term dHCP or

ABCD data as well as its adaptation to preterm data is not influenced by the fact that dHCP and ABCD were used in the original training process.

Comment 6:

Can the authors justify split the groups into early and late birth at 30 weeks? Typically the breakdown would be extremely preterm birth (<28 weeks), very preterm birth (28-32 weeks), and moderate to late preterm birth (>32 weeks). The authors break down appears rather ad hoc.

Response 6:

We thank the Reviewer for this insightful comment and agree that gestational age thresholds of 28 weeks (extreme preterm) and 32 weeks (very preterm) are more commonly used in clinical practice. The cutoff at 30 weeks GA was chosen to balance the number of participants in the earlier preterm (GA of ≤ 30) and later preterm (GA of > 30) groups and to thereby reducing the likelihood that the percentage of subjects in the groups with extranormal deviations are driven by outliers. Using a cutoff of 30 weeks in the original analysis led to the following group sizes:

Dataset	earlier preterm (GA ≤ 30 weeks)	later preterm (GA > 30 weeks)
BLS-26	52	43
dHCP	27	65

To evaluate the robustness of our findings, we conducted an additional control analysis using standard clinical thresholds, dividing preterm subjects into the following three groups: (i) extreme preterm (GA ≤ 28 weeks), (ii) very preterm ($28 < \text{GA} \leq 32$ weeks), and (iii) late preterm (GA > 32 weeks). Results are included in the Supplementary Material. Briefly, for the dHCP cohort, the group sizes were $n = 14, 30,$ and $48,$ with up to 57%, 20% and 19% of participants respectively sharing a supranormal deviation in any given region. For the BLS-26 cohort, the group sizes were $n = 14, 67,$ and $15,$ with corresponding maxima of 57%, 27%, and 20% of subjects showing shared infranormal deviations.

These results are consistent with our main analysis and support the conclusion that the likelihood of shared extranormal deviations increases with the severity of prematurity.

In the Results section, the Supplementary analysis was referred to as follows:

Repeating this analysis with partitioning subjects into groups more commonly used in clinical practice, i.e., extreme preterm (GA of ≤ 28 weeks), very preterm ($28 < \text{GA} \leq 32$ weeks), and late preterm (GA > 32 weeks) resulted in similar results (Supplementary Fig. S9) but led to less balanced group sizes. This in general, these results indicate that earlier birth increases the percentage of individuals that share extranormal structural alterations in overlapping regions.

Comment 7:

Do the authors have detailed information about brain injuries in any of the cohorts? How many participants had intraventricular hemorrhage (and what grade), periventricular Leukomalacia, cerebellar hemorrhage, etc? Major brain injuries are common in preterms and associate with altered brain morphometry and cognitive outcomes.

Response 7:

We greatly appreciate that the Reviewer posed the very interesting question of whether the preterm participants of any cohort had brain injuries. We have collected and summarized the available information about brain injuries for each cohort in the following and them to the Supplementary Methods and Results section. Furthermore, we have conducted control

analyses for average dysmaturational outcomes, interindividual heterogeneity, and the correlation of IBAPs with cognitive outcomes when excluding subjects with neonatal brain injuries for the dHCP and BLS-26 cohorts, where data on neonatal brain injuries were available.

Heterogeneity of cortical aberrations and association with cognition after preterm birth when excluding cases of severe perinatal brain injury

Since perinatal brain injuries such as intracranial hemorrhage or focal white matter lesions, which are often caused by preterm birth, might impact brain morphometry and cognitive outcomes, we performed a control analysis of the main results excluding subjects with perinatal brain injuries.

For the dHCP cohort, the variable “radiology_score” was used, which indicates the presence of incidental findings that might be of clinical significance and/or might affect the image reconstruction. Radiology score was rated on a 1–5-point scale with the following meanings: (1) normal appearance for age, (2) incidental findings with unlikely significance for clinical outcome or analysis (e.g., subdural hemorrhage, isolated subependymal cysts, mild inferior vermis rotation), (3) incidental findings with unlikely clinical significance but possible analysis significance (e.g. several punctate lesions or other focal white matter/ cortical lesions not thought to be of clinical significance), (4) incidental findings with possible clinical significance, unlikely analysis significance (e.g. isolated non brain anomaly, e.g., in pituitary / on tongue), (5) incidental findings with possible/likely significance for both clinical and imaging analysis (e.g., major lesions within white / matter, cortex, cerebellum and or basal ganglia, small head/brain < 1st centile) (file “nnsi01_definitions.csv” downloaded from https://nda.nih.gov/edit_collection.html?id=3955). Subjects with a radiology score of > 2 were excluded for the control analysis ($n_{\text{preterm}} = 64$, $n_{\text{full-term}} = 78$).

For the ABCD cohort, no information on perinatal brain injury was available.

For the BLS cohort, since no MRI scan was performed at birth, information on neonatal brain injury was limited. Only a few preterm subjects had an intracranial hemorrhage (ICH) according to ultrasound examinations graded on a 1-4-point scale. As previous work by our group has shown, the presence but not the grading of ICH is a significant predictor of ventricular enlargement in adulthood⁵⁵, suggesting that the presence of neonatal brain injury impacts brain structure into adulthood. Therefore, in this control analysis, we excluded all subjects with neonatal ICH of any grading ($n_{\text{preterm}} = 15$).

After exclusion of subjects with perinatal brain injury, analyses investigating average dysmaturational, interindividual heterogeneity, as well as correlation with cognitive outcomes were conducted as described in the main methods section.

Whereas average dysmaturational outcomes differed compared to the main analyses (see Fig. 2-4), individual heterogeneity results remained fairly stable, with no more than 30% of preterm subjects sharing extranormal deviations in any given region (Supplementary Fig. S4). The association of the first principal component of IBAPs across regions with cognitive outcome measures were not significant, similar to the original findings (see Fig. 9). On the other hand, the association between PC1 of SA IBAPs with full-scale IQ was still significant in preterm adults, also when excluding subjects with intracranial hemorrhage, but only before FDR-correction (Spearman rho = 0.274, p-value = 0.016, $p_{\text{FDR}} = 0.064$).

Comment 8:

The authors’ thresholding approach to binarizing deviations likely increases the dissimilarity between individuals. For example, Taylor et al showed that thresholded results look more dissimilar to each other than using maps that retain all the information. This is because

thresholding is a non linear operation that enforces discontinuities. How similar are the deviation maps if they are not binarized?

Taylor PA, Reynolds RC, Calhoun V, Gonzalez-Castillo J, Handwerker DA, Bandettini PA, Mejia AF, Chen G. Highlight results, don't hide them: Enhance interpretation, reduce biases and improve reproducibility. *Neuroimage*. 2023 Jul 1;274:120138. doi: 10.1016/j.neuroimage.2023.120138. Epub 2023 Apr 27. PMID: 37116766; PMCID: PMC10233921.

Response 8:

We thank the Reviewer for raising this important and well-founded concern that binarization of maps can artificially inflate perceived dissimilarities between individuals. However, we would like to clarify the rationale for the use of binarized deviation maps and provide additional context about the interpretation of centile-based deviation scores.

Importantly, centile scores cannot be interpreted on a linear scale in their biological meaning. The distance between two centile values does not necessarily reflect a proportional biological or clinical difference. For example, two individuals with deviation scores of 0.05 and 0.10 may be biologically more dissimilar than those with scores of 0.45 and 0.50, despite the same absolute difference in units. This is due to the increasing rarity and extremity of deviations as one moves toward the tails of the distribution. In this context, centiles near the median reflect typical variation, whereas values beyond the 5th or 95th percentile reflect deviations that are more likely to be clinically meaningful or biologically atypical⁵⁶. Therefore, our binarization strategy (i.e., binarizing deviations at <0.05 or >0.95) was motivated by this interpretive difficulty. The binarized maps were not intended to capture the full spectrum of interindividual heterogeneity, but rather to highlight that the locations of extranormal deviations, which are likely to be clinically or biologically relevant, tend to be very different between most preterm individuals. Even if a similar event occurs (i.e., preterm birth), the locations where this event manifests itself in the brain are very different.

Nevertheless, we acknowledge the Reviewer's request to report the similarity of unthresholded deviation maps between preterm individuals. To address this, we calculated the correlation coefficients between the non-binarized IBAP of one subject with the IBAPs of all others, thereby preserving the full distributional information. In summary, correlation coefficients are higher between IBAPs of preterm individuals when IBAPs are not binarized but still low ($r < 0.25$) for most subjects. To ensure transparency, we included this analysis to the Supplement (Supplementary Figures S6 and S7).

Comment 9:

Given that the term group shows a large amount of large deviations, how similar are they to each other and the preterm group? In other words, is the amount of individual differences specific to the preterm group?

Response 9:

We thank the Reviewer for raising this important point. We would like to clarify that the term-born groups of each cohort show the expected level of variability for a healthy population. As illustrated in Fig. 2d, Fig. 3b, and Fig. 4b (as well as Supplementary Fig. S1 for a different normative model), the majority of term-born individuals do not show an extranormal deviation in any region since only about 5 % of subjects show deviations that are below the 5th or above the 95th percentile of the reference population. This is consistent with the 5 % cutoff we defined as supra- or infranormality: a deviation is infranormal (supranormal) if its value is lower than the 5th percentile (higher than the 95th percentile) of the general healthy population. This distribution reflects normal interindividual variation, not unusually high levels of deviation. In line, also previous normative modeling studies investigating various brain measures have reported

extranormal deviations for small fractions of healthy participants (roughly 5 % infra- and supranormal deviations)^{49,52,57–59}. In contrast, a substantially higher proportion of preterm individuals show extranormal deviations across the cortex, supporting our main conclusion that individual-level structural alterations are more prevalent in the preterm group.

Concerning Figure 2b, we agree that the term-born controls appear to be distributed around a value that is slightly lower than the population median. The observation is likely due to the insular CTh distribution of the term-born female subgroup in the BLS-26 cohort, which appears to be lower than the median of the reference population for their age. The reason for this might be the relatively small size ($n=45$) of the subgroup, which is probably not fully representative of the general population. As a result, their median CTh could deviate slightly from the reference. Nevertheless, when considering both male and female term-born individuals together, fewer than 5% exhibit an extranormal deviation in insular CTh, as shown in Figures 2d, 3b, and 4b. This indicates that the model generalizes well to the BLS cohort overall.

Since the decision to show the female insular CTh in the plot was not driven by any specific reason and we want to avoid confusion, we have exchanged the depicted trajectory to the male insular CTh one. This is now indicated in the figure description:

Figure 2: Brain development after preterm birth is individually heterogeneous. [...] b, [...] The lifespan trajectory of the insula cortical thickness (CTh) in males is shown as an example region, with the insular CTh of male preterm (red) and full-term (black) subjects plotted on top.

We hope this clarifies that the pattern observed in the term-born groups is expected and that the elevated rate and spatial clustering of deviations in the preterm group point to a specific effect of early developmental disruption.

Comment 10:

There has been many criticisms of the spin test. Exchangeability is violated in several ways. It assumes complete stationarity. Also there is a need to interpolate which also violates exchangeability.

Response 10:

We thank the Reviewer for their valuable comment regarding the use of the spin test in our analysis. While widely used, we agree that the method has theoretical limitations. Therefore, we would like to provide more information about the procedure, which we admit was not sufficiently described in the original manuscript, and to clarify our rationale for its application.

First, we want to highlight that the theoretical basis of the spin test differs from that of traditional permutation tests⁶⁰, which would imply exchangeable elements. Unfortunately, the procedure was described as “spatial permutation test” in the original publication⁶⁰ and has since been referred to as such^{61–63}. As we agree that this term is misleading, we will refer to the method as “spatial null model” or “spin test” from now on to avoid confusion.

Regarding exchangeability, we fully agree that this assumption is typically violated in spatially structured systems such as the brain since it exhibits spatial autocorrelation and therefore implies that regions are not statistically independent. In line, prior work has shown that both conventional parametric and non-parametric tests lead to inflated false-positive rates when assessing the correspondence between two brain maps^{60,64}. The spin test addresses this problem by treating the overall spatial structure of the brain map as fixed and resampling the alignment of two maps via rotation on the spherical surface, rather than resampling the data points themselves. Thus, the null distribution is generated by randomizing the relative alignment, not by assuming exchangeable vertices. Therefore, as vertex-wise exchangeability is

not assumed, the spin test provides a more appropriate framework for testing map-to-map correspondence^{60,64}.

Concerning stationarity, we agree that the assumption of uniform spatial autocorrelation across the cortex is a simplification that is unlikely to be true in the brain. However, in the context of the spin test, this assumption is looser compared to other spatial models⁶⁴. Because the rotations preserve the spatial structure of the input data, the method maintains regional non-stationarities in the maps being compared. Importantly, the alignment itself between the two maps is being resampled (as opposed to the maps being resampled). Still, as more recent work shows, the projection to the surface⁶⁵ and the rotation of the medial wall onto the cortex⁶⁶, in some cases leads to the disruption of spatial autocorrelation when generating spatial surrogates with the spin test. While imperfect, spatial null models like the spin test have been shown to better control for false-positive rates compared to parametric or non-parametric tests when assessing statistical significance of map-to-map correspondence^{60,64}.

We have adapted the Methods and Results sections to better describe the applied procedure and refer to it as “spatial null models” now:

To bridge macro- and microscale by mapping cell-type marker gene expression patterns to regional CTh deviation scores across the cortical sheet, we conducted regional Spearman correlation analysis across the 34 brain regions for each subject. Thereby, significance was assessed using spatial null models a permutation test that preserves spatial autocorrelation (“spin test”, see Methods)^{60,64}.

[...]

To explore the cellular correlates of individual CTh heterogeneity in preterm 26-year-old adults, we spatially aligned assessed the spatial correspondence between cell type-specific gene expression maps with and individual deviation score patterns using spatial Spearman correlation across 34 cortical regions. Whereas correlation coefficients for analyses of two spatial maps are meaningful, standard parametric or non-parametric p-values-methods for statistical inference are not as they are influenced by the spatial autocorrelation in spatial systems such as the brain.; This potentially leads to inflated p-values and subsequently to increased familywise error rates across analyses⁶⁴. To circumvent this problem, we applied spatial autocorrelation preserving permutation tests-spatial null models (“spin tests”) to assess the statistical significance of these correlations map-to-map correspondences^{60,64}. In brief, one of the investigated brain maps is projected to a sphere, random angular rotations are applied to this spherical projections of the brain, and the rotated map is then projected back to the brain surface. By repeating this procedure 1000 times and calculating the correspondence between the rotated maps and the second brain map of interest, to calculate a null distribution of correlation coefficients between the two brain maps is generated. The original correlation coefficient is then compared to this null distribution. This method maintains the existing spatial autocorrelation by applying the same random rotations to each parcel, thereby preserving the spatial structure of the brain map of a brain map by substituting the original values with those from the nearest rotated coordinate^{60,64}. The analyses were implemented in Python using the ENIGMA toolbox⁶⁷ (<https://github.com/MICA-MNI/ENIGMA>).

Importantly, we want to highlight that the statistical significance derived from the spin test did not play a central role in the interpretation of macroscopic underpinnings of adult IBAPs. Following the correlation between cell type-specific gene expression maps and IBAPs, Spearman correlation coefficients of all preterm individuals from the BLS-26 dataset were linked to gestational age, regardless of whether the correlations reached significance under the spin

test. We followed this procedure because we have tested the hypothesis whether the spatial profiles of IBAP tends to be more similar to the regional distribution of glial cells if the subject was born earlier in the preterm period. Thus, the main conclusions of our study are not dependent on the specific outcomes of the spin-based significance testing.

Still, to address the concerns about methodological robustness, we also report the correlation between adult IBAPs and cell type-specific gene expression patterns using an alternative spatial null model to assess significance between the association between cell type-specific gene expression maps and IBAPs after preterm birth, namely Brain Surrogate Maps with Autocorrelated Spatial Heterogeneity (BrainSMASH; Supplementary Methods and Results, Supplementary Fig. S16). The method generates surrogate brain maps with similar spatial autocorrelation than the original one via variogram estimation^{64,68} and therefore is not based on applying random angular rotations to the brain maps:

Spatial association between adult CTh IBAPs and cell type-specific gene expression using an alternative spatial null model

In the main analysis, the spin test was used to assess statistical significance of the spatial association between adult CTh IBAPs and cell type-specific gene expression. While widely used, we agree that this method has theoretical limitations. For example, the projection to the surface⁶⁵ and the rotation of the medial wall onto the cortex⁶⁶, in some cases leads to the disruption of spatial autocorrelation when generating spatial surrogates with the spin test. While imperfect, spatial null models like the spin test have been shown to better control for false-positive rates compared to parametric or non-parametric tests when assessing statistical significance of map-to-map correspondence^{60,64}.

As a control analysis, we provide the significance of correlations between adult CTh IBAPs and regional cell type-specific gene expression as assessed with an alternative spatial null model, namely Brain Surrogate Maps with Autocorrelated Spatial Heterogeneity (BrainSMASH). The method generates surrogate brain maps with similar spatial autocorrelation than the original one via variogram estimation^{64,68}. Similarly to the main results, the association of some subjects' CTh IBAPs with some regional profiles of cellular density were significantly similar (Supplementary Methods and Results, Supplementary Fig. S16).

Comment 11:

I am not sure how the results support a therapeutic approach in preterm birth or even integrating brain charts. Most preterm clinics would already recommend early cognitive and social interventions, given the well-known risk factors.

Response 11:

We appreciate the Reviewer's comment and agree that early interventions are already standard in many preterm clinics due to known group-level risk. However, our aim is to go beyond general risk factor estimation by incorporating *individual* brain abnormality patterns as potential predictors for later outcomes. This might help differentiating the preterm population into higher or lower risk based on actual brain deviations of each individual subject instead of general risk factors, allowing for more tailored monitoring and intervention strategies as well as individual recommendations - a key step toward a more therapeutic approach. While preliminary, our findings suggest that integrating such individual markers of brain aberrations may support more personalized care in the future.

Comment 12:

The lack of long term longitudinal data is more of a weakness the authors suggest. Certainly the ABCD and BLS allow to test for stability, but the follow up is short compared to the development period. For example, one would not expect a large amount of change between 26 and 38 years. Longitudinal changes over infancy or from infancy through childhood would be important to know to show the value of a growth chart type approach. The dHCP has longitudinal data for the preterm cohort.

Response 12:

We appreciate the Reviewer's comment highlighting the importance of long-term longitudinal data, especially across early developmental periods. We fully agree that monitoring individual trajectories from infancy through childhood would offer a stronger basis for growth chart-like approaches and for assessing longitudinal consistency of extranormal deviations. However, as the Reviewer may be aware, such longitudinal datasets in the preterm populations are rare, very challenging to acquire, and typically not publicly available.

The dHCP dataset is the only dataset of this kind. However, as the Reviewer correctly mentioned, longitudinal scans were only acquired for preterm subjects, with the first time-point being acquired before term-equivalent age (i.e., fetal scans). Because refitting the normative models relies on term-born reference data at each timepoint to disentangle effects of age from diagnosis, incorporating these fetal scans into the analysis was not possible since model refitting could not be conducted only based on preterm data. To illustrate these considerations, we have adjusted the Limitations section to emphasize this constraint more explicitly:

Second, for preterm neonates, we rely on cross-sectional data only. Although fetal scans were available for a small group of participants, normative modeling relies on reference data from term-born subjects at each timepoint to disentangle effects of age from diagnosis. Due to lack of these data, longitudinal analysis in the dHCP cohort was not possible. Furthermore, apart from the BrainChart reference charts⁸, robust models for early age ranges between birth and 2 years of age are lacking, hindering validation of neonatal results. Since substantial heterogeneity of regional CTh in preterm neonates has been shown previously⁵, however, there is further evidence based on different methodologies for individual heterogeneity of structural brain aberrations after preterm birth also in infancy. Future studies would benefit from longitudinal data, both for reference chart estimations⁶⁹ and for assessing individual preterm brain developmental trajectories.

References

1. Adamson, C. L. *et al.* Updates to the Melbourne Children’s Regional Infant Brain Software Package (M-CRIB-S). *Neuroinformatics* **22**, 207–223 (2024).
2. Adamson, C. L. *et al.* Parcellation of the neonatal cortex using Surface-based Melbourne Children’s Regional Infant Brain atlases (M-CRIB-S). *Sci Rep* **10**, 4359 (2020).
3. Desikan, R. S. *et al.* An automated labeling system for subdividing the human cerebral cortex on MRI scans into gyral based regions of interest. *Neuroimage* **31**, 968–980 (2006).
4. Schmitz-Koep, B. *et al.* Decreased cortical thickness mediates the relationship between premature birth and cognitive performance in adulthood. *Hum Brain Mapp* **41**, 4952–4963 (2020).
5. Dimitrova, R. *et al.* Preterm birth alters the development of cortical microstructure and morphology at term-equivalent age. *Neuroimage* **243**, 118488 (2021).
6. Zubiaurre-Elorza, L. *et al.* Cortical Thickness and Behavior Abnormalities in Children Born Preterm. *PLoS One* **7**, e42148 (2012).
7. Kelly, C. E. *et al.* Long-lasting effects of very preterm birth on brain structure in adulthood: A systematic review and meta-analysis. *Neurosci Biobehav Rev* **147**, 105082 (2023).
8. Bethlehem, R. A. I. *et al.* Brain charts for the human lifespan. *Nature* **604**, 525–533 (2022).
9. McCarthy, C. S. *et al.* A comparison of FreeSurfer-generated data with and without manual intervention. *Front Neurosci* **9**, (2015).
10. Rakesh, D., Zalesky, A. & Whittle, S. Similar but distinct – Effects of different socioeconomic indicators on resting state functional connectivity: Findings from the Adolescent Brain Cognitive Development (ABCD) Study®. *Dev Cogn Neurosci* **51**, 101005 (2021).
11. Bradley, R. H. & Corwyn, R. F. Socioeconomic Status and Child Development. *Annu Rev Psychol* **53**, 371–399 (2002).
12. Riegel, K., Ohrt, B. & Wolke, D. *Die Entwicklung Gefährdeter Geborener Kinder Bis Zum Fünften Lebensjahr : Die Arvo Ylppö-Neugeborenen-Nachfolgestudie in Südbayern Und Südfinnland [The Development of Children Born at Risk until Their Fifth Year of Life]*. (Ferdinand Enke Verlag, Stuttgart, Germany, 1995).
13. Bauer, A. *Ein Verfahren Zur Messung Des Für Das Bildungsverhalten Relevanten Sozial Status (BRSS)—Überarbeitete Fassung*. (Deutsches Institut für Internationale Pädagogische Forschung, Frankfurt, Germany, 1988).
14. Joseph, R. M., O’Shea, T. M., Allred, E. N., Heeren, T. & Kuban, K. K. Maternal educational status at birth, maternal educational advancement, and neurocognitive outcomes at age 10 years among children born extremely preterm. *Pediatr Res* **83**, 767–777 (2018).
15. Wong, H. S. & Edwards, P. Nature or Nurture: A Systematic Review of the Effect of Socio-economic Status on the Developmental and Cognitive Outcomes of Children Born Preterm. *Matern Child Health J* **17**, 1689–1700 (2013).
16. Fernández de Gamarra-Oca, L. *et al.* Preterm birth and early life environmental factors: neuropsychological profiles at adolescence and young adulthood. *Journal of Perinatology* **43**, 1429–1436 (2023).
17. Edwards, A. D. *et al.* The Developing Human Connectome Project Neonatal Data Release. *Front Neurosci* **16**, (2022).
18. Albers, C. A. & Grieve, A. J. Test Review: Bayley, N. (2006). Bayley Scales of Infant and Toddler Development– Third Edition. San Antonio, TX: Harcourt Assessment. *J Psychoeduc Assess* **25**, 180–190 (2007).
19. Luciana, M. *et al.* Adolescent neurocognitive development and impacts of substance use: Overview of the adolescent brain cognitive development (ABCD) baseline neurocognition battery. *Dev Cogn Neurosci* **32**, 67–79 (2018).

20. Heaton, R. K. *et al.* Reliability and Validity of Composite Scores from the NIH Toolbox Cognition Battery in Adults. *Journal of the International Neuropsychological Society* **20**, 588–598 (2014).
21. Wolke, D., Johnson, S. & Mendonça, M. The Life Course Consequences of Very Preterm Birth. *Annu Rev Dev Psychol* **1**, 69–92 (2019).
22. Twilhaar, E. S. *et al.* Cognitive Outcomes of Children Born Extremely or Very Preterm Since the 1990s and Associated Risk Factors. *JAMA Pediatr* **172**, 361 (2018).
23. Mendonça, M., Bilgin, A. & Wolke, D. Association of Preterm Birth and Low Birth Weight With Romantic Partnership, Sexual Intercourse, and Parenthood in Adulthood A Systematic Review and Meta-analysis. *JAMA Netw Open* **2**, E196961 (2019).
24. Dassios, T., Williams, E. E., Harris, C. & Greenough, A. Using cluster analysis to describe phenotypical heterogeneity in extremely preterm infants: a retrospective whole-population study. *BMJ Open* **12**, e056567 (2022).
25. Lammertink, F., Vinkers, C. H., Tataranno, M. L. & Benders, M. J. N. L. Premature Birth and Developmental Programming: Mechanisms of Resilience and Vulnerability. *Front Psychiatry* **11**, (2021).
26. Haynes, R. L., Sleeper, L. A., Volpe, J. J. & Kinney, H. C. Neuropathologic Studies of the Encephalopathy of Prematurity in the Late Preterm Infant. *Clin Perinatol* **40**, 707–722 (2013).
27. Eves, R. *et al.* Association of Very Preterm Birth or Very Low Birth Weight With Intelligence in Adulthood. *JAMA Pediatr* **175**, e211058 (2021).
28. Lacalle, L., Martínez-Shaw, M. L., Marín, Y. & Sánchez-Sandoval, Y. Intelligence Quotient (IQ) in school-aged preterm infants: A systematic review. *Front Psychol* **14**, 1216825 (2023).
29. Baranowska-Rataj, A., Barclay, K., Costa-Font, J., Myrskylä, M. & Özcan, B. Preterm birth and educational disadvantage: Heterogeneous effects. *Popul Stud (NY)* **77**, 459–474 (2023).
30. Inder, T. E., Volpe, J. J. & Anderson, P. J. Defining the Neurologic Consequences of Preterm Birth. *New England Journal of Medicine* **389**, 441–453 (2023).
31. Dimitrova, R. *et al.* Heterogeneity in Brain Microstructural Development Following Preterm Birth. *Cerebral Cortex (New York, NY)* **30**, 4800–4810 (2020).
32. Dimitrova, R. *et al.* Phenotyping the Preterm Brain: Characterizing Individual Deviations From Normative Volumetric Development in Two Large Infant Cohorts. *Cerebral Cortex* **31**, 3665–3677 (2021).
33. Marín-Padilla, M. Developmental Neuropathology and Impact of Perinatal Brain Damage. II. *J Neuropathol Exp Neurol* **56**, 219–235 (1997).
34. Dean, J. M. *et al.* Prenatal cerebral ischemia disrupts MRI-defined cortical microstructure through disturbances in neuronal arborization. *Sci Transl Med* **5**, 168ra7 (2013).
35. Rimol, L. M. *et al.* Reduced white matter fractional anisotropy mediates cortical thickening in adults born preterm with very low birthweight. *Neuroimage* **188**, 217–227 (2019).
36. Croteau-Chonka, E. C. *et al.* Examining the relationships between cortical maturation and white matter myelination throughout early childhood. *Neuroimage* **125**, 413–421 (2016).
37. Natu, V. S. *et al.* Apparent thinning of human visual cortex during childhood is associated with myelination. *Proc Natl Acad Sci U S A* **116**, 20750–20759 (2019).
38. Choi, U.-S., Shim, S.-Y., Cho, H. J. & Jeong, H. Association between cortical thickness and cognitive ability in very preterm school-age children. *Sci Rep* **14**, 2424 (2024).
39. Bjuland, K. J., Løhaugen, G. C. C., Martinussen, M. & Skranes, J. Cortical thickness and cognition in very-low-birth-weight late teenagers. *Early Hum Dev* **89**, 371–380 (2013).
40. Kelly, C. E. *et al.* Cortical growth from infancy to adolescence in preterm and term-born children. *Brain* **147**, 1526–1538 (2023).

41. Volpe, J. J. Placental assessment provides insight into mechanisms and timing of neonatal hypoxic-ischemic encephalopathy. *J Neonatal Perinatal Med* **12**, 113–116 (2019).
42. Rakic, P. A small step for the cell, a giant leap for mankind: a hypothesis of neocortical expansion during evolution. *Trends Neurosci* **18**, 383–388 (1995).
43. Sripada, K. *et al.* Trajectories of brain development in school-age children born preterm with very low birth weight. *Sci Rep* **8**, 15553 (2018).
44. Rimol, L. M. *et al.* Atypical brain structure mediates reduced IQ in young adults born preterm with very low birth weight. *Neuroimage* **266**, 119816 (2023).
45. Rakic, P. Specification of Cerebral Cortical Areas. *Science* (1979) **241**, 170–176 (1988).
46. Wolke, D. & Meyer, R. Cognitive status, language attainment, and prereading skills of 6-year-old very preterm children and their peers: the Bavarian Longitudinal Study. *Dev Med Child Neurol* **41**, 94–109 (1999).
47. Eryigit Madzwamuse, S., Baumann, N., Jaekel, J., Bartmann, P. & Wolke, D. Neuro-cognitive performance of very preterm or very low birth weight adults at 26 years. *J Child Psychol Psychiatry* **56**, 857–864 (2015).
48. Bayer, J. M. M. *et al.* Site effects how-to and when: An overview of retrospective techniques to accommodate site effects in multi-site neuroimaging analyses. *Front Neurol* **13**, 923988 (2022).
49. Di Biase, M. A. *et al.* Cell type-specific manifestations of cortical thickness heterogeneity in schizophrenia. *Mol Psychiatry* **27**, 2052–2060 (2022).
50. Zabihi, M. *et al.* Dissecting the Heterogeneous Cortical Anatomy of Autism Spectrum Disorder Using Normative Models. *Biol Psychiatry Cogn Neurosci Neuroimaging* **4**, 567–578 (2019).
51. Worker, A. *et al.* Extreme deviations from the normative model reveal cortical heterogeneity and associations with negative symptom severity in first-episode psychosis from the OPTiMiSE and GAP studies. *Transl Psychiatry* **13**, 373 (2023).
52. Rutherford, S. *et al.* Charting brain growth and aging at high spatial precision. *Elife* **11**, (2022).
53. Romero, R., Dey, S. K. & Fisher, S. J. Preterm labor: One syndrome, many causes. *Science* (1979) **345**, 760–765 (2014).
54. O’Muircheartaigh, J. *et al.* Modelling brain development to detect white matter injury in term and preterm born neonates. *Brain* **143**, 467–479 (2020).
55. Hedderich, D. M. *et al.* Sequelae of Premature Birth in Young Adults: Incidental Findings on Routine Brain MRI. *Clin Neuroradiol* **31**, 325–333 (2021).
56. Marquand, A. F., Rezek, I., Buitelaar, J. & Beckmann, C. F. Understanding Heterogeneity in Clinical Cohorts Using Normative Models: Beyond Case-Control Studies. *Biol Psychiatry* **80**, 552–561 (2016).
57. Huang, A. S. *et al.* Lifespan development of thalamic nuclei and characterizing thalamic nuclei abnormalities in schizophrenia using normative modeling. *Neuropsychopharmacology* **49**, 1518–1527 (2024).
58. Allen, P. *et al.* Normative Modeling of Brain Morphometry in Clinical High Risk for Psychosis. *JAMA Psychiatry* **81**, 77 (2024).
59. Sun, X. *et al.* Mapping Neurophysiological Subtypes of Major Depressive Disorder Using Normative Models of the Functional Connectome. *Biol Psychiatry* **94**, 936–947 (2023).
60. Alexander-Bloch, A. F. *et al.* On testing for spatial correspondence between maps of human brain structure and function. *Neuroimage* **178**, 540–551 (2018).
61. Larivière, S. *et al.* Structural network alterations in focal and generalized epilepsy assessed in a worldwide ENIGMA study follow axes of epilepsy risk gene expression. *Nat Commun* **13**, 4320 (2022).
62. Li, J. *et al.* Cortical structural differences in major depressive disorder correlate with cell type-specific transcriptional signatures. *Nat Commun* **12**, 1647 (2021).

63. Dear, R. *et al.* Cortical gene expression architecture links healthy neurodevelopment to the imaging, transcriptomics and genetics of autism and schizophrenia. *Nat Neurosci* (2024) doi:10.1038/s41593-024-01624-4.
64. Markello, R. D. & Misic, B. Comparing spatial null models for brain maps. *Neuroimage* **236**, 118052 (2021).
65. Bazinet, V., Liu, Z.-Q. & Misic, B. The effect of spherical projection on spin tests for brain maps. Preprint at <https://doi.org/10.1101/2024.12.15.628553> (2024).
66. Váša, F. *et al.* Adolescent Tuning of Association Cortex in Human Structural Brain Networks. *Cerebral Cortex (New York, NY)* **28**, 281–294 (2018).
67. Larivière, S. *et al.* The ENIGMA Toolbox: multiscale neural contextualization of multisite neuroimaging datasets. *Nat Methods* **18**, 698–700 (2021).
68. Burt, J. B., Helmer, M., Shinn, M., Anticevic, A. & Murray, J. D. Generative modeling of brain maps with spatial autocorrelation. *Neuroimage* **220**, 117038 (2020).
69. Di Biase, M. A. *et al.* Mapping human brain charts cross-sectionally and longitudinally. *Proceedings of the National Academy of Sciences* **120**, (2023).

Response to reviewers

Revision for the article

“Heterogeneous, temporally consistent, and plastic brain development after preterm birth”

by Thalhammer et al.

Responses to Reviewer #1

Comment 1:

I commend the authors for the thorough revision of their manuscript - in particular the inclusion of the latest dHCP release data, and the integration of SA and CTh measurements in the body of the manuscript. Most of the technical limitations are now appropriately acknowledged in their discussion section.

Response 1:

We thank the Reviewer for the effort to thoroughly review this manuscript.

Comment 2:

The authors provide the code to replicate their results and this is a useful tool for the community.

Response 2:

We thank the Reviewer for the appreciation to share our analysis code.

Responses to Reviewer #2

Comment 1:

Response 1:

We thank the Reviewer for the positive evaluation of the paper and to take part in this initiative.

Responses to Reviewer #3

Comment 1:

The authors have addressed most of my original concerns. While a few points would benefit from following minor refinements.

Response 1:

We thank the Reviewer for the thorough evaluation of the manuscript.

Comment 2:

In Figure S9 (SA metric, dHCP cohort), the reported Spearman correlation (mean $r = 0.5$, max $r = 0.75$) of each brain region for their deviation patterns across subjects is explained as “several regions not showing any extranormal deviations, which may increase the influence of a few deviating regions.” This is somewhat odd, as the correlations are performed on subject-wise deviation profiles—not region-wise, as in the main results. I would suggest that a plausible explanation is that although there is substantial heterogeneity in the specific regions showing deviations across individuals, certain brain regions show intrinsic covariation in their deviation patterns across the cohort. This does not undermine the main conclusion of their findings, but if the authors agree, a brief note in the discussion part may help readers better understand the nature of these heterogeneous deviations in preterm.

Response 2:

We agree with the Reviewer that our original interpretation of correlations between regional deviations across subjects was unclear. Therefore, we have revised the corresponding section in the Supplement:

An exception can be observed in the dHCP cohort, where average coefficients reach up to $r = 0.75$. This likely reflects that, although there is substantial heterogeneity among regions that show an extranormal deviation between individuals, certain brain regions may exhibit some covariation in their deviation patterns within the cohort, i.e., some deviation patterns between subgroups of preterm individuals may exist.

We opted not to include this point in the main Discussion, as these findings are based on supplementary, exploratory analyses and do not alter the main conclusions of the study.

Comment 3:

At line 154, the description of Figures S9–S10 is incorrect. These figures present results regarding the relationships between regional deviations across individuals, rather than results of non-binarized deviations.

Response 3:

We thank the Reviewer's for pointing out the incorrect description of Figures S7-S8 (former Figures S9-S10). Thus, we have revised the text to accurately reflect the content of these figures and to provide more specific references to the Supplementary Information:

Control analyses for non-binarized deviations (Supplementary Fig. S5-S6) as well as the similarity of regional deviation patterns across individuals (Supplementary methods and results, Supplementary Fig. S7-S8) are provided in the Supplement.

Comment 4:

In Figure 6, the presentation of location consistency for a few selected brain regions in the main text may give a somewhat selective impression. As shown in the supplementary materials, some regions such as the temporal lobe, do not display particularly stable deviations over time. This is also consistent with the results of ICC analysis. I highly appreciate that the authors included ICC analyses in the main text and emphasized the heterogeneity in the longitudinal stability of regional deviation. However, including the full distribution of regional ICC values in Figure 6 (such as the plot of cortical surface in Figure 7) would provide a more comprehensive perspective.

Response 4:

We thank the Reviewer for the constructive feedback and the helpful suggestion to include a full distribution of regional ICC values in Figure 6 on cortical surfaces. We have followed this suggestion and included two further panels, showing regional longitudinal ICCs of surface area on cortical surfaces. Furthermore, an additional Supplementary Figure was generated to show the regional ICC distribution for cortical thickness (i.e., Supplementary Fig. S16).

Comment 5:

I did not find any description on Figure 2h. Please either include a description or remove it.

Response 5:

We thank the Reviewer for pointing out that the legend to Figure 2h is missing. We have added the following to the Figure legend:

Figure 2: Brain development after preterm birth is individually heterogeneous. [...] h, Further example individual deviation score profiles. Left and right columns as in panel c.

Comment 6:

A few minor typos remain. For instance, line 124 reads “with. severe perinatal brain injury,”. A careful proofreading would help.

Response 6:

We have thoroughly revised the manuscript for typos.

Responses to Reviewer #4

Comment 1:

The authors has answered my question.

Response 1:

We thank the Reviewer for the positive evaluation of the manuscript.